# Consistency Conditions for Differentiable Surrogate Losses

**Drona Khurana**
University of Colorado Boulder
drona.khurana@colorado.edu

**Anish Thilagar**
University of Colorado Boulder
anish@colorado.edu

**Dhamma Kimpara**$^*$
NSF National Center for Atmospheric Research
Boulder, Colorado
dkimpara@ucar.edu

**Rafael Frongillo**
University of Colorado Boulder
raf@colorado.edu

## Abstract

The statistical consistency of surrogate losses for discrete prediction tasks is often checked via the condition of calibration. However, directly verifying calibration can be arduous. Recent work shows that for polyhedral surrogates, a less arduous condition, indirect elicitation (IE), is still equivalent to calibration. We give the first results of this type for non-polyhedral surrogates, specifically the class of convex differentiable losses. We first prove that under mild conditions, IE and calibration are equivalent for one-dimensional losses in this class. We construct a counter-example that shows that this equivalence fails in higher dimensions. This motivates the introduction of strong IE, a strengthened form of IE that is equally easy to verify. We establish that strong IE implies calibration for differentiable surrogates and is both necessary and sufficient for strongly convex, differentiable surrogates. Finally, we apply these results to a range of problems to demonstrate the power of IE and strong IE for designing and analyzing consistent differentiable surrogates.

## 1 Introduction

In supervised learning problems, the goal of the learner is to output a model that accurately predicts labels on unseen feature vectors. These problems are specified by *target losses*, metrics intended to reflect model error. Natural choices for target losses arise in discrete prediction tasks like classification, ranking, and structured prediction. As minimizing discrete target losses directly is generally NP-hard, a convex *surrogate loss* is typically used instead. A link function maps surrogate reports (predictions) to target reports. Beyond ease of optimization, the surrogate and link must be *statistically consistent* with respect to the target loss, meaning that minimizing the surrogate should closely approximate minimizing the target given sufficient training data. In the finite-outcome setting, consistency turns out to be equivalent to a simpler condition called *calibration* [Bartlett et al., 2006, Tewari and Bartlett, 2007, Ramaswamy and Agarwal, 2016]. In particular, calibration has been central to the design of new consistent surrogates, serving as the key condition which must be satisfied.

While simpler than consistency, directly verifying calibration is often cumbersome. In particular, calibration requires that *all sequences of reports* converging to the surrogate minimizers (i.e., minimizers of the expected surrogate loss), eventually link to target minimizers (Figure 1). This complexity of verifying calibration in turn impedes the design of consistent surrogates. *What easily*

---

$^*$Work done while at University of Colorado Boulder.

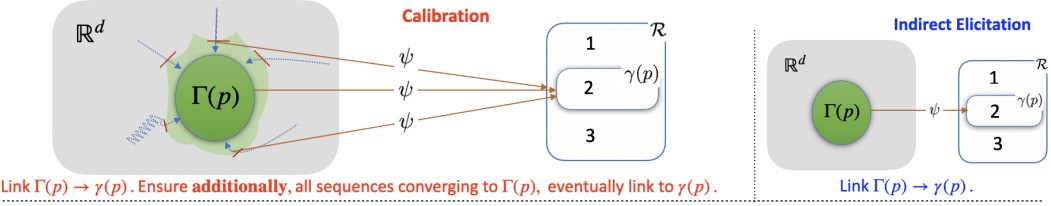

Notation: $\mathbb{R}^d$- surrogate reports; $\Gamma(p)$- surrogate minimizers; $\mathcal{R}$ - target reports; $\gamma(p) = \{2\}$- target minimizers; $\psi$ - link

Figure 1: **Calibration vs. Indirect Elicitation.** Calibration requires surrogate minimizers, as well as all sequences converging to surrogate minimizers link to target minimizers. In general, it is not trivial to choose a universal threshold past which the sequences link as desired. Determining such a threshold requires careful reasoning about the relative positions of surrogate minimizers across different outcome distributions, i.e., $\Gamma(p)$ relative to $\Gamma(q)$ for $q \neq p$. IE is analytically easier to verify, as it only requires that surrogate minimizers link to target minimizers. Moreover, IE can be thought of as a geometric condition on the probability simplex, which can directly lead to design insights (§4).

*verifiable conditions still imply calibration for important classes of surrogate losses?* One promising candidate is *indirect elicitation* (IE), which only requires that surrogate minimizers be linked to target minimizers (Figure 1). Finocchiaro et al. [2019] established an equivalence between calibration and IE for polyhedral surrogates, which paved the way for the design of novel, consistent surrogates for several open target losses of interest [Wang and Scott, 2020, Thilagar et al., 2022, Finocchiaro et al., 2022b]. Whether or not this equivalence extends to other classes of surrogates has remained open. More generally, beyond the polyhedral case, we still lack simpler conditions for calibration that support the design of new surrogates. We give the first such results—IE-like conditions which are easier to verify than calibration—for the broad and practically relevant class of convex, differentiable losses. We then demonstrate the power of these conditions to streamline the design process.

**Theoretical Contributions.** We first show that IE and calibration are equivalent for 1-d convex, differentiable losses (§ 3).[2] In higher dimensions, however, IE no longer implies calibration, even for strongly convex surrogates (Example 2). To address this disparity, we propose a novel strengthening of IE we call *strong indirect elicitation* (strong IE; see Definition 6 in § 3.4). Strong IE is as easy to verify as IE, as it only depends on surrogate minimizers and not on the surrounding sequences (§ 4.1). We prove that under mild technical assumptions, strong IE implies calibration for differentiable surrogates (Theorem 2). Moreover, for the important class of strongly convex, differentiable surrogates, we show that strong IE is both necessary and sufficient for calibration (Theorem 3). All our calibration proofs are constructive, providing explicit link functions as part of the argument. Taken together, our results deepen our understanding of the conditions required for statistical consistency of surrogate losses.

**Significance for Design.** We illustrate with two examples that proving IE or strong IE is strictly simpler than directly proving calibration, thus drastically shortening the pathway to establishing consistency (§4). We then demonstrate how the geometric insights from these simpler conditions enable the construction of consistent, 1-d differentiable surrogates for any orderable target loss (Theorem 4). As an application of Theorem 4, we construct a novel 1-d surrogate that is convex, differentiable and consistent with respect to the ordinal regression loss. Together, these applications offer instructive proofs of concept and highlight how IE and strong IE can guide efficient surrogate design. We conclude with important future directions (§5).

**Related Work.** Previous works have also studied easier-to-verify conditions that imply calibration for certain classes of surrogate losses; our work is unique in proposing conditions for arbitrary discrete targets that are broadly applicable to the class of differentiable surrogates. The first conditions for calibration were studied for the important case of multi-class classification, where the target loss is the 0-1 loss. For binary classification, Bartlett et al. [2006] study 1-d surrogates and show that these are calibrated if and only if the loss, in margin form $\phi(uy)$, is differentiable at 0 and $\phi'(0) < 0$. In multi-class classification, Tewari and Bartlett [2007] study higher dimensional surrogates with symmetric superprediction sets [Williamson and Cranko, 2023]. They establish a technical condition on these sets that allows for easier conditions for calibration such as ensuring that the optimizer sets are singletons. The first to formally study the relationship between IE and calibration were Agarwal and Agarwal [2015]. However, their results do not apply to general discrete targets. For polyhedral

---

[2]The statement does not hold if one relaxes differentiability; see Example 1.

losses, Ramaswamy and Agarwal [2016] showed that an IE-like condition is sufficient for calibration and Finocchiaro et al. [2024] showed that IE is *equivalent* to calibration for arbitrary discrete targets. Specific applications where IE influenced the design and analysis of surrogates/calibration include Finocchiaro et al. [2022b,a], Wang and Scott [2020], Nueve et al. [2024]. Our study of 1-d surrogates is heavily informed by the structure and elicitation of the target properties elucidated in Lambert et al. [2008], Lambert [2011], Steinwart et al. [2014], Finocchiaro and Frongillo [2018].

## 2 Background and Preliminaries

| Symbol | Description | Symbol | Description |
|---|---|---|---|
| $\mathcal{Y} = [n]$ | Set of labels | $\mathcal{R} = [k]$ | Set of target reports |
| $\ell : \mathcal{R} \to \mathbb{R}^n$ | Discrete target loss function | $L : \mathbb{R}^d \to \mathbb{R}^n$ | Surrogate loss function |
| $\psi : \mathbb{R}^d \to \mathcal{R}$ | Link function | $\Delta_n$ | Probability simplex |
| $\gamma : \Delta_n \rightrightarrows \mathcal{R}$ | Property elicited by $\ell$ | $\Gamma : \Delta_n \rightrightarrows \mathbb{R}^d$ | Property elicited by $L$ |
| $\gamma_r$ | Level-set for $\gamma$ at $r$ | $\Gamma_u$ | Level-set for $\Gamma$ at $u$ |

Table 1: Summary of key notation

### 2.1 Targets, Surrogates and Link Functions

Given a finite **label space** $\mathcal{Y}$ and a finite **report space** $\mathcal{R}$, let $\ell : \mathcal{R} \to \mathbb{R}^{|\mathcal{Y}|}$ be a **discrete target loss** associated with some prediction task. $\ell(\cdot)_y$ represents the loss when the label is $y \in \mathcal{Y}$. Unless specified otherwise, we assume $\mathcal{Y} = [n]$ and $\mathcal{R} = [k]$, for $n, k \geq 2$. We denote the probability simplex over $\mathcal{Y}$ by $\Delta_n := \{p \in \mathbb{R}^n | p_i \geq 0, \forall i \in [n], p^\top \mathbf{1}_n = 1\}$. As in Finocchiaro et al. [2020, 2024], we assume that the target loss under consideration is *non-redundant*, i.e., every report $r \in \mathcal{R}$ uniquely minimizes the expected loss for some distribution, i.e., $\forall r \in \mathcal{R}, \exists p \in \Delta_n$ such that $\operatorname{argmin}_{r' \in \mathcal{R}} \langle p, \ell(r') \rangle = \{r\}$.

Since $\ell$ is discrete and non-convex, it is hard to optimize. Our objective then, is to replace $\ell$ with a **surrogate loss**, defined over a **continuous prediction space**, say $\mathbb{R}^d$. Denote the surrogate by: $L : \mathbb{R}^d \to \mathbb{R}^n$. For $y \in [n]$, let $L(\cdot)_y : \mathbb{R}^d \to \mathbb{R}$ denote the $y^{th}$ component of $L$. To enable optimization of the surrogate, we assume that each component of the surrogate is *convex*, i.e., $L(\cdot)_y : \mathbb{R}^d \to \mathbb{R}$ is convex for each $y \in \mathcal{Y}$. Furthermore, since we are interested in analyzing differentiable surrogate losses, we will also assume that $L(\cdot)_y$ is *differentiable* for each $y \in \mathcal{Y}$. For all our theoretical results, we will make the following assumption:

**Assumption 1.** $\operatorname{argmin}_{u \in \mathbb{R}^d} L(u)_y$ *is non-empty and compact for each* $y \in [n]$.

Within the class of differentiable functions, Assumption 1 encompasses an important range of surrogates—including those with strongly convex components and strictly convex minimizable components. It also covers more nuanced cases, such as the surrogate described in Example 7 in Appendix A, which features two Huber-like components. Although each component is uniquely minimizable, not all their convex combinations are. We demonstrate in Section 4 that a one-dimensional instantiation of this loss is calibrated with respect to the ordinal regression target loss.

As the objective is to minimize $\ell$, we must systematically map predictions in $\mathbb{R}^d$ back to $\mathcal{R}$. To do so, we introduce a **link function** $\psi : \mathbb{R}^d \to \mathcal{R}$.

### 2.2 Property Elicitation, Calibration and Indirect Elicitation

Any set-valued function defined on $\Delta_n$ is called a *property*. We say a loss *elicits* a property, if it maps each distribution to the minimizer of the expected loss under said distribution. We work with two key properties, denoted $\gamma$ and $\Gamma$, which we define below:

**Definition 1** (Target Property, Elicits, Level Sets)**.** *The target loss* $\ell : \mathcal{R} \to \mathbb{R}^n$ *is said to elicit the property* $\gamma : \Delta_n \rightrightarrows \mathcal{R}$*, or in short-hand* $\gamma := prop[\ell]$ *if*

$$\gamma(p) := \operatorname*{arg\,min}_{r \in \mathcal{R}} \langle p, \ell(r) \rangle .$$

*For any $r \in \mathcal{R}$, denote $\gamma_r \subseteq \Delta_n$ as the* level-set *for $\gamma$ at $r$, i.e., $\gamma_r := \{p \in \Delta_n | r \in \gamma(p)\}$. Since $\mathcal{R}$ is a finite set, we say $\gamma$ is a finite property.*

**Definition 2** (Surrogate Property, Elicits, Level Sets). *The surrogate loss $L : \mathbb{R}^d \to \mathbb{R}^n$ is said to elicit the property $\Gamma : \Delta_n \rightrightarrows \mathbb{R}^d$, or in short-hand $\Gamma := prop[L]$ if*

$$\Gamma(p) := \underset{u \in \mathbb{R}^d}{\arg \min} \langle p, L(u) \rangle \ .$$

*For any $u \in \mathbb{R}^d$, denote $\Gamma_u \subseteq \Delta_n$ as the* level-set *for $\Gamma$ at $u$, i.e., $\Gamma_u := \{p \in \Delta_n | u \in \Gamma(p)\}$.*

In order to ensure that a surrogate-link pair is actually solving the target problem, we need to ensure that statistical consistency holds. In the finite outcome setting, it is well known that consistency reduces to the simpler notion of calibration [Bartlett et al., 2006, Tewari and Bartlett, 2007, Ramaswamy and Agarwal, 2016]. We thus focus on calibration throughout this paper. Given some distribution $p \in \Delta_n$, and the corresponding surrogate minimizer(s) $\Gamma(p)$, calibration roughly requires that all sequences of approximate minimizers link to the optimal target report.

**Definition 3** (Calibration). *Given a discrete target $\ell$, a surrogate-link pair $(L, \psi)$ is calibrated if $\forall p \in \Delta_n$:*

$$\underset{u \in \mathbb{R}^d : \psi(u) \notin \gamma(p)}{\inf} \langle p, L(u) \rangle > \underset{u \in \mathbb{R}^d}{\inf} \langle p, L(u) \rangle \ .$$

*We also say $L$ is calibrated, if there exists a link $\psi$, such that $(L, \psi)$ is calibrated.*

We next define indirect elicitation, a condition even weaker than calibration (see Theorem 6 in Appendix B for a proof).

**Definition 4** (Indirect Elicitation). *A surrogate-link pair, $(L, \psi)$ indirectly elicits a discrete target $\ell$, if $\forall u \in \mathbb{R}^d$, $\Gamma_u \subseteq \gamma_{\psi(u)}$. We also say $L$ indirectly elicits $\ell$, if $\forall u \in \mathbb{R}^d$, $\exists r \in \mathcal{R}$ such that $\Gamma_u \subseteq \gamma_r$.*

For polyhedral surrogates, Finocchiaro et al. [2024] established that indirect elicitation and calibration are equivalent. This result is striking, as indirect elicitation is significantly easier to verify than calibration. The latter requires ensuring that, for each distribution $p \in \Delta_n$, any sequence of reports minimizing $\langle p, L(\cdot) \rangle$ in the limit, eventually links to $\gamma(p)$. Equivalently, it demands that any sequence converging to $\Gamma(p)$ ultimately links to $\gamma(p)$ (see Lemma 17 in Appendix C). In contrast, verifying indirect elicitation only necessitates linking optimal surrogate reports to optimal target reports. Specifically, if $\Gamma_u = \emptyset$ for some report $u$ then IE holds trivially and there is nothing to check. Otherwise, if $p \in \Gamma(u)$, IE demands that $\psi(u) \in \gamma(p)$. Thus, IE is fully determined by the structure of the minimizing reports, i.e., $\Gamma(\Delta_n) := \{\Gamma(p) | p \in \Delta_n\}$, whereas calibration requires analyzing $\Gamma(p)$ along with the local behavior of reports around it.

## 3 Motivating Counterexamples and Main Results

Our primary aim is to identify simpler conditions that yield calibration for differentiable surrogates. IE seems to be an ideal candidate: it is substantially easier to verify, is known to be equivalent for polyhedral losses, and all previously studied calibrated convex surrogates satisfy IE. However, it is not quite strong enough in general. We present two novel counterexamples (Example 1 in 3.1, Example 2 in 3.3) that demonstrate why IE is insufficient for calibration. These examples are far from pathological, and thus demonstrate exactly why IE is too weak for this setting. Example 2 motivates a new condition, strong IE that we go onto show implies calibration for convex, differentiable surrogates, and is both necessary and sufficient if the surrogate has strongly convex components.

### 3.1 Indirect elicitation and calibration are not equivalent

All previously studied convex surrogates that are known to indirectly elicit a target loss are calibrated for *some* link function $\psi$.[3] The literature has therefore treated both conditions as roughly equivalent to each other. Yet, this is not generally true. We identify the first known example of a loss that satisfies IE but cannot be calibrated for *any* choice of link function.

---

[3]For example, consider the hinge surrogate for 0-1 loss, $\mathcal{Y} = \mathcal{R} = \{-1, 1\}$ and $L(u)_y = \max(0, 1 - uy)$. If the link boundary is at $u = 1$, $\psi(u) = 1 \iff u \geq 1$, $(L, \psi)$ satisfies IE but is not calibrated. However, if the link boundary is moved to $u = 0$, then $(L, \psi)$ is calibrated.

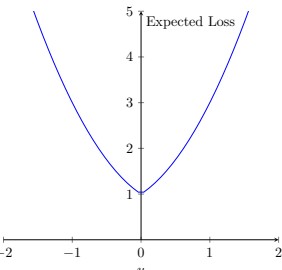 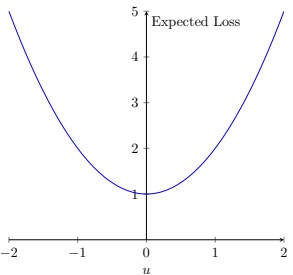

Figure 2: The expected loss for two surrogates for abstain loss. **Left:** $\mathbb{E}_p L_{\text{cusp}}$ at $p = (0.5, 0.5)$, it is clear that no link yields calibration for the abstain target (Example 1). **Right:** $\mathbb{E}_p[L_{\text{smooth}}]$, a smoothed version of $L_{\text{cusp}}$ that is calibrated, again depicted at $p = (0.5, 0.5)$ (Example 6, Appendix A).

**Example 1** (Cusp). *Let $\ell_{abs} : \{-1, \perp, 1\} \to \mathbb{R}^2$ be the target loss for binary classification with abstain level $\frac{1}{4}$ using the label space $\mathcal{Y} = \{-1, 1\}$ [Bartlett and Wegkamp, 2008].*

$$\ell_{abs}(r)_y = \begin{cases} 0 & r = y \\ 1/4 & r = \perp \\ 1 & r = -y \end{cases} .$$

*Let $L_{cusp} : \mathbb{R} \to \mathbb{R}^2$ be the surrogate loss with $L_{cusp}(u)_y = (1 - uy)^2 + |u|$, and $\psi(u) = \text{sign}(u)$ for $u \neq 0$ and $\psi(0) = \perp$ (abstain). The expected loss of $L_{cusp}$, $\mathbb{E}_p[L(u)]$ at $p = (0.5, 0.5)$ is plotted in Figure 2 (left).*

*It is target-optimal to abstain whenever the most likely outcome occurs with a probability of at most $3/4$, i.e., $\gamma_\perp := \{p \in \Delta_2 : \max\{p_1, p_2\} \leq 3/4\} = \{p \in \Delta_2 | p_1 \in [1/4, 3/4]\}$. Now, $(L_{cusp}, \psi)$ indirectly elicits $\ell$ as $\Gamma_0 = \gamma_\perp$. Note that for any link to satisfy IE, it must agree with $\psi$ in $[-0.5, 0.5]$. However, calibration is not satisfied for any $p_1 \in (1/4, 3/4)$, for example: set $p = (0.5, 0.5)$. Consider any positive sequence $\{u_t > 0\}_{t \geq 0}$ with $u_t \to 0$. Then, $\lim_{t \to \infty} \langle p, u_t \rangle = \inf_{u \in \mathbb{R}} \langle p, L(u) \rangle$. However, each $u_t$ links to $1$ and never the correct report, $\perp$. Indeed any link that satisfies IE exhibits this behavior (the sequence $\{u_t\}$ eventually links to $1$). Thus, there is no other choice that could yield calibration. To restore calibration, it suffices to "smooth out" the non-differentiable cusp at $u = 0$, to get a differentiable surrogate as in Figure 2. See Example 6 in Appendix A.*

$L_{\text{cusp}}$ is a remarkably simple non-polyhedral loss: it is strongly convex, one-dimensional, and differentiable everywhere except for a single cusp at $u = 0$. It is as 'nice' as a non-differentiable loss can be. Yet, despite indirectly eliciting $\ell_{\text{abs}}$, it still fails to satisfy calibration. Since smoothing out the cusp yields a calibrated loss, a natural question arises: *does differentiability, combined with indirect elicitation, always imply calibration?* Differentiable losses are well-structured and extensively studied in machine learning as they are optimization-friendly and enjoy fast convergence rates. This makes the question of understanding the connection between IE and calibration under differentiability all the more compelling.

## 3.2 Differentiability and IE imply calibration for $d = 1$

In 1-dimension, we answer the above question affirmatively: IE *does* imply calibration for differentiable real-valued surrogates. We provide a proof sketch of our theorem in this section.

To set the stage, we recall that a target loss $\ell$ that is indirectly elicited by a 1-d surrogate (differentiable or not) possesses special structure. In particular, Finocchiaro et al. [2020] showed that the property $\gamma := \text{prop}[\ell]$ corresponding to such a target satisfies a condition known as *orderability*, which roughly states that there exists a connected, 1-dimensional path that crosses each of the target level-sets.

**Definition 5** (Orderable [Lambert, 2011]). *A finite property $\gamma : \Delta_n \rightrightarrows \mathcal{R}$ is orderable, if there is an enumeration of $\mathcal{R} = \{r_1, r_2, ..., r_k\}$ such that for all $i \leq k - 1$, we have that $\gamma_{r_j} \cap \gamma_{r_{j+1}}$ is a hyperplane intersected with $\Delta_n$.*

**Theorem 1.** *Let $L : \mathbb{R} \to \mathbb{R}^n$ be a convex, differentiable surrogate that indirectly elicits $\ell$. Under Assumption 1, $L$ is calibrated with respect to $\ell$.*

*Proof sketch:* We first show that for each $j \in [k - 1]$, the boundary between adjacent target cells, i.e., $\gamma_{r_j} \cap \gamma_{r_{j+1}}$ overlaps completely with some surrogate level-set. So, for some $u^j \in \mathbb{R}$, $\Gamma_{u^j} = \gamma_{r_j} \cap \gamma_{r_{j+1}}$

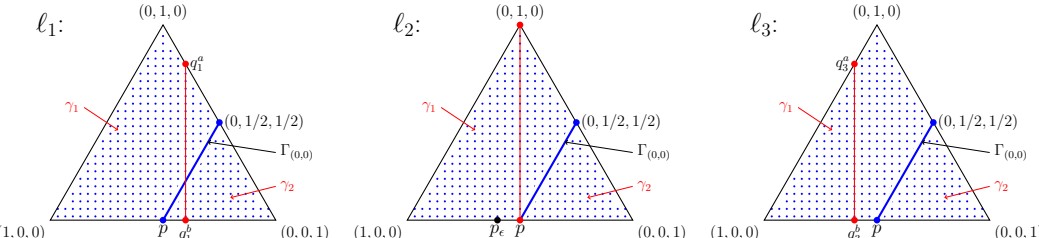

Figure 3: Let $\mathcal{Y} = \{1, 2, 3\}$, $\mathcal{R} = \{1, 2\}$. Three candidate target losses $\ell_1, \ell_2, \ell_3 : \mathcal{R} \to \mathbb{R}^3$, that $L_{\mathrm{CE}}$ (Example 2) could be a surrogate for. For each $i \in \{1, 2, 3\}$, $\ell_i(1) = (1, 1, 1)$. Whereas, $\ell_1(2) = (5/2, 5/4, 0)$, $\ell_2(2) = (2, 1, 0)$ and $\ell_3(2) = (5/3, 5/6, 0)$. The target boundary elicited by $\ell_1$ (resp. $\ell_3$) is the *red line segment* joining $q_1^a$ and $q_1^b$ (resp. $q_3^a$ and $q_3^b$). The target boundary elicited by $\ell_2$ is the *red line segment* joining $p$ and $(0, 1, 0)$. The level sets of $L_{\mathrm{CE}}$ are the *blue points*. All level sets of $L_{\mathrm{CE}}$ are single points, barring $\Gamma_{(0,0)}$, which is the entire segment spanning from $p = (1/2, 0, 1/2)$ to $(0, 1/2, 1/2)$ (*blue line segment*). **Left: no IE.** The segment level set *crosses* the target boundary, so $L_{\mathrm{CE}}$ cannot indirectly elicit $\ell_1$. **Center: IE.** The segment level set *does not cross, but just touches* the target boundary, so IE holds, however, strong IE does not hold. **Right: strong IE.** The segment level set *lies entirely within* the target cell, so strong IE holds. Note: $q_1^a = (0, 0.8, 0.2), q_1^b = (0.4, 0, 0.6), q_3^a = (0.2, 0.8, 0), q_3^b = (0.6, 0, 0.4)$.

(Lemma 20). We then establish that for any two distributions $p, q \in \Delta_n$ that lie on either side of the target boundary $\gamma_{r_j} \cap \gamma_{r_{j+1}}$, optimal reports $\Gamma(p), \Gamma(q)$ must lie on either side of $u^j$ (Lemma 21). Together, these results establish the existence of a connected, 1-dimensional path through surrogate minimizers that faithfully mirrors any connected 1-dimensional path traversing the target level sets. This naturally induces a link $\psi$ that tracks the paths by mapping $u^j$ to either of $\{r_j, r_{j+1}\}$, and mapping $\Gamma(p)$ and $\Gamma(q)$ to $r_j$ and $r_{j+1}$ (Theorem 10). $\qquad\square$

The full proof and constructive link $\psi$ are presented in Appendix D. It differs significantly from the polyhedral case, since barring convexity, differentiable and polyhedral losses have no commonality in their underlying structure.

### 3.3 Differentiability and IE do not imply calibration for $d > 1$

Unfortunately, there is no direct analogue of Theorem 1 in higher dimensions. In particular, the following 2-dimensional surrogate is differentiable and satisfies IE for a target, but is not calibrated.

**Example 2** (Counterexample: IE without calibration). *Let $\mathcal{Y} = \{1, 2, 3\}$, $\mathcal{R} = \{1, 2\}$ and consider $L_{CE} : \mathbb{R}^2 \to \mathbb{R}^{\mathcal{Y}}$, where*

$$L_{CE}(u) = \begin{pmatrix} u_1^2 + u_1 + u_2^2 + 2u_2 \\ 2u_1^2 + u_1 + 2u_2^2 + 2u_2 \\ 3u_1^2 - u_1 + u_2^2 - 2u_2 \end{pmatrix} .$$

*Each component of $L_{CE}$ is differentiable and strongly convex - and so $L_{CE}$ is minimizable. $L_{CE}$ indirectly elicits the target $\ell_2 : \mathcal{R} \to \mathbb{R}^3$ shown in Figure 3 (center, red). However, there is no link function $\psi : \mathbb{R}^2 \to \mathcal{R}$, such that the pair $(L_{CE}, \psi)$ is calibrated with respect to $\ell_2$. In particular, there exists a sequence of reports that uniformly link to 1, but converge to $(0, 0)$, which has to link to 2.*

*More formally, define for $0 < \epsilon \le 1$, $p_\epsilon := (1/2 + \epsilon/2, 0, 1/2 - \epsilon/2)$. Then $\Gamma(p_\epsilon) = \left\{ \left( \frac{-\epsilon}{5-\epsilon}, \frac{-2\epsilon}{3+\epsilon} \right) \right\}$. Notice that $\gamma(p_\epsilon) = \{1\} \implies \psi(\Gamma(p_\epsilon)) = 1$ necessarily. Simultaneously, $\Gamma_{(0,0)} \subseteq \gamma_2$ and $\Gamma_{(0,0)} \not\subseteq \gamma_1, \implies \psi((0, 0)) = 2$. Denote $p^* := (0, 1/2, 1/2) \in \Gamma_{(0,0)}$ and observe that $\gamma(p^*) = \{2\}$. Then, since $\Gamma(p_\epsilon) \to (0, 0)$ as $\epsilon \to 0$, it follows by continuity that $\langle p^*, L_{CE}(\Gamma(p_\epsilon)) \rangle \to \langle p^*, L_{CE}((0, 0)) \rangle = \inf_{u \in \mathbb{R}^2} \langle p^*, L(u) \rangle$. Thus, $L_{CE}$ violates calibration for any choice of link $\psi$.*

Similarly to $L_{\mathrm{cusp}}$, the surrogate $L_{\mathrm{CE}}$ is extremely well-behaved. Each of its components are differentiable, strongly convex, minimizable, and the minimizing reports are always compact sets. Yet, $L_{\mathrm{CE}}$ violates calibration despite satisfying IE with respect to $\ell_2$.

Turning our attention again to Figure 3 (center), where level-sets are depicted in blue: we see that geometrically, the violation stems from the location of $\Gamma_{(0,0)}$, where $\Gamma = \mathrm{prop}[L_{\mathrm{CE}}]$. Every surrogate

level-set is a singleton, except $\Gamma_{(0,0)}$ (blue line segment). Notice that $\Gamma_{(0,0)}$ just touches the (red) target boundary $\gamma_1 \cap \gamma_2$ at the distribution $(1/2, 0, 1/2)$. Shifting $\gamma_1 \cap \gamma_2$ to the right to get $\ell_1$ immediately violates IE, since $\Gamma_u \not\subseteq \gamma_r$ for any $r \in \{1, 2\}$, when $\gamma = \text{prop}[\ell_1]$ (Figure 3, left). On the other hand, shifting the boundary $\gamma_1 \cap \gamma_2$ by any amount to the left yields a target loss of form $\ell_3$, for which $L_{\text{CE}}$ is calibrated (Figure 3, right). So, while indirect elicitation requires that the segment level set be contained within $\gamma_2$, calibration is only achieved for $L_{\text{CE}}$ when $\Gamma_{(0,0)}$ is bounded away from the target boundary.

### 3.4 Strong indirect elicitation

Example 2 suggests that while calibration fails under IE, bounding the level set away from the target boundary resolves the problem. We formalize this idea with a new condition, *strong indirect elicitation*, which is a strengthening of indirect elicitation (see Theorem 5 in Appendix B).

**Definition 6** (Strong Indirect Elicitation). *Given a target loss $\ell$, let $\gamma_S^* = \{p : \gamma(p) = S\}$. A surrogate $L$ strongly indirectly elicits $\ell$ if $\forall u, \exists S \subseteq \mathcal{R}$ such that $\Gamma_u \subseteq \gamma_S^*$; equivalently, if for every $u \in \mathbb{R}^d$ and every $p, q \in \Gamma_u$, $\gamma(p) = \gamma(q)$.*

Revisiting Example 2: Notice that $L_{\text{CE}}$ does not satisfy strong IE with respect to $\ell_2$, since $\gamma(p) = \{1, 2\}$, while $\gamma((0, 1/2, 1/2)) = \{2\}$ and both $p, (0, 1/2, 1/2) \in \Gamma_{(0,0)}$. However, $\gamma(p) = \gamma((0, 1/2, 1/2)) = \{2\}$, when $\gamma = \text{prop}[\ell_3]$. Thus, $L_{\text{CE}}$ satisfies strong IE with respect to $\ell_3$.

Though close to IE in definition, strong IE turns out to be much more powerful for differentiable surrogates, in that it implies calibration.[4]

**Theorem 2.** *Let $L$ be a convex, differentiable surrogate that strongly indirectly elicits $\ell$. Under Assumption 1, $L$ is calibrated with respect to $\ell$.*

*Proof sketch:* Fix $p \in \Delta_n$. Key to our proof is establishing that the minimizers "surrounding" $\Gamma(p)$ link to $\gamma(p)$. Define the *level-set bundle at $p$* to be the collection of all level-sets passing through $p$, i.e., $\Gamma_{\Gamma(p)} := \cup_{u \in \Gamma(p)} \Gamma_u$. Repeated applications of strong IE establish the following: for a sufficiently small $\epsilon_p > 0$, the surrogate minimizers of distributions in '$\epsilon_p$-proximity' to the level-set bundle at $p$ link to $\gamma(p)$ (Lemma 29). For simplicity, denote this set of minimizers as the $\epsilon_p$-minimizers. We have thus far that any valid link $\psi$ must ensure that $\psi(\epsilon_p\text{-minimizers}) \in \gamma(p)$. By establishing upper-hemicontinuity of the set-valued map $\Gamma_{(\cdot)} : \mathbb{R}^d \rightrightarrows \mathcal{R}$ (Lemma 26), we show that for some $\delta_p > 0$ there exists a $\delta_p$-neighborhood around $\Gamma(p)$ wherein all minimizers are $\epsilon_p$-minimizers (Lemma 27). Thus, all minimizers surrounding $\Gamma(p)$ link to $\gamma(p)$. In effect, this means that surrogate minimizers that link to different target reports are well-separated in space which is imperative for calibration. We conclude via an explicit construction to extend this link $\psi$ to a calibrated link defined for all surrogate reports, including the nowhere-optimal ones (Theorem 11). The reader may refer to Figure 7 in Appendix F for visual intuition of the proof. $\square$

Finally, we show that restricting to surrogates with strongly convex components makes strong IE necessary for calibration, and thus strong IE and calibration are equivalent for these surrogates.

**Theorem 3.** *Let $L : \mathbb{R}^d \rightarrow \mathbb{R}^n$ be a surrogate, such that $L(\cdot)_y : \mathbb{R}^d \rightarrow \mathbb{R}$ is strongly convex and differentiable for each $y \in [n]$. Then, $L$ is calibrated with respect to $\ell$ if and only if it strongly indirectly elicits $\ell$.*

*Proof sketch:* Strong convexity and differentiability together imply Assumption 1. The sufficiency of strong IE thus follows by Theorem 2. For necessity, we show that violating strong IE implies violating calibration. If IE is violated, calibration is violated immediately. So let us assume we have IE but not strong IE. We show that under strong convexity, $\Gamma$ is continuous and single-valued (see Lemma 35 for a proof). Next, we show the existence of a report $u \in \mathbb{R}^d$ and a pair of distributions $p, q \in \Gamma_u$, such that $\gamma(p) \subset \gamma(q)$ (see Lemma 36). Thus, $\exists r \in \mathcal{R} : r \in \gamma(q)$, however, $r \notin \gamma(p)$. We then show that there exists a sequence of reports $q_t \rightarrow q$, such that $\gamma(q_t) = r$. As $\Gamma$ is single-valued there exists $u_t = \Gamma(q_t), \forall t$. By continuity of $\Gamma$, $q_t \rightarrow q \implies \Gamma(q_t) \rightarrow \Gamma(q) \iff u_t \rightarrow u$. Further, by the continuity of $\langle p, L(\cdot) \rangle$, $\langle p, L(u_t) \rangle \rightarrow \langle p, L(u) \rangle = \inf_{v \in \mathbb{R}^d} \langle p, L(v) \rangle$ since $p \in \Gamma_u$. However, since $\gamma(q_t) = \{r\}$, $\psi(u_t) = r$ necessarily. At the same time, $r \notin \gamma(p)$. Hence, calibration is violated at $p$. See Theorem 12 in Appendix G for a full proof. $\square$

---

[4] Interestingly, no polyhedral surrogate satisfies strong IE; see Theorem 7 in Appendix B

# 4  Applications

As IE and strong IE are easier to verify than calibration (Figure 1), our main results above lead to improved analytical methods to analyze and design consistent surrogates, which we now demonstrate.

## 4.1  Ease of verification

IE and strong IE are both completely characterized by the relation of optimal surrogate reports to optimal target reports. Importantly, neither condition requires analyzing sequences converging to optimal reports. Thus both conditions are strictly simpler to verify than calibration. While strong IE is a more stringent requirement than IE, checking strong IE is just as easy as checking IE at the individual-report level (see Proposition 1 in Appendix B).

We now present two examples illustrating how concluding calibration via IE or strong IE can significantly simplify the analysis: whereas direct calibration proofs require characterizing minimizers and analyzing nearby sequences, establishing IE or strong IE only requires reasoning about the minimizers themselves. (see also Figure 1 for visual intuition)

**Example 3** (Universally calibrated surrogate). *Lemma 11 of Ramaswamy and Agarwal [2016] proposes a $n - 1$-dimensional, strongly convex, differentiable surrogate that is calibrated for all discrete targets. After the first claim in their proof (see pages 29-30 Ramaswamy and Agarwal [2016]):*

> ### Proof via strong IE
>
> *Fix $p \in \Delta_n$. Minimizing $\langle p, L(u) \rangle = \sum_{j=1}^{n-1} \left( p_j (u_j - 1)^2 + (1 - p_j) u_j^2 \right)$ yields the unique minimizer $u^* = (p_1, \ldots, p_{n-1})^\top$. Hence $|\Gamma(p)| = 1$ and $\Gamma_u = \{p\}$. Immediately, L satisfies strong IE, and thus L is calibrated by Theorem 2.*

*Our approach shortens the proof from an entire page to a few lines. We also obviate the need for subtle arguments regarding the convergence of sequences that were required in the original proof.*

**Example 4** (Subset-ranking surrogates). *Theorem 3 of Ramaswamy et al. [2013] proposes a low-dimensional calibrated surrogate for subset-ranking targets common in information retrieval. Our results significantly shorten their calibration proof (see pages 3-4, Ramaswamy et al. [2013]):*

> ### Proof via strong IE
>
> *The surrogate is strongly convex and differentiable, so strong IE suffices for calibration. Pick any $\mathbf{u} \in \mathbb{R}^d$ and any $\mathbf{p}, \mathbf{q} \in \Gamma_{\mathbf{u}}$. To prove strong IE, it suffices to show that $\gamma(\mathbf{p}) = \gamma(\mathbf{q})$. $\mathbf{u}$ is the unique minimizer for $\langle \mathbf{p}, L(\cdot) \rangle$ and $\langle \mathbf{q}, L(\cdot) \rangle \implies (*) \underline{\mathbf{u}^{\mathbf{p}} = \mathbf{u}^{\mathbf{q}} = \mathbf{u}}$. By line 1 of page 4, $\mathbf{p}^T \boldsymbol{\ell}_t = (\mathbf{u}^{\mathbf{p}})^\top \boldsymbol{\beta}_t + c$. Similarly, $\mathbf{q}^\top \boldsymbol{\ell}_t = (\mathbf{u}^{\mathbf{q}})^\top \boldsymbol{\beta}_t + \underline{c}$. By $(*)$, $\mathbf{p}^\top \boldsymbol{\ell}_t = \mathbf{q}^\top \boldsymbol{\ell}_t$ for any $t \in \mathcal{T}$ (target reports). Thus, $argmin_{t \in \mathcal{T}} \mathbf{p}^\top \boldsymbol{\ell}_t = argmin_{t \in \mathcal{T}} \mathbf{q}^\top \boldsymbol{\ell}_t$. So, $\gamma(\mathbf{p}) = \gamma(\mathbf{q})$.*

*This bypasses all subsequent proof steps (25 lines) following the first line of page 4 wherein intricate reasoning to show all sequences converging to minimizer sets are appropriately linked.*

## 4.2  Design of 1-dimensional surrogates

Example 3 demonstrates the existence of an $n - 1$ dimensional surrogate that is calibrated for any target loss with $n$ outcomes. However, the complexity of several optimization algorithms is often linear, or even quadratic in the domain dimension. Thus, a major research goal of the surrogate loss literature is the design of dimension-efficient surrogates (ideally $d << n - 1$ for large $n$) when possible [Ramaswamy and Agarwal, 2012, Ramaswamy et al., 2013, 2015, Finocchiaro et al., 2019, Blondel, 2019, Finocchiaro et al., 2021]. Recall that Theorem 1 established the equivalence between IE and calibration for 1-d differentiable surrogates. This equivalence enables us to construct a 1-d surrogate that is convex and differentiable, for any orderable target loss. We formalize this statement below in Theorem 4 and provide a proof sketch that highlights the key ideas behind the construction.

**Theorem 4.** *Given an orderable target $\ell : \mathcal{R} \to \mathbb{R}^n$, there exists a convex, differentiable surrogate $L : \mathbb{R} \to \mathbb{R}^n$ satisfying Assumption 1, which is calibrated with respect to $\ell$.*

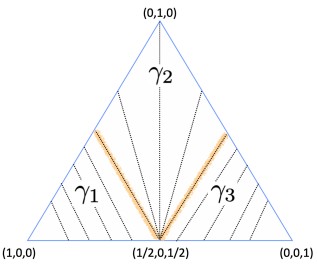

Figure 4: Let $\mathcal{Y} = \mathcal{R} = \{1, 2, 3\}$. The solid-peach colored lines depict the target boundaries elicited by the ordinal regression loss $\ell^{\mathrm{ord}} : \mathcal{R} \to \mathbb{R}^3$. The dotted-blue lines depict the level-sets of the surrogate $L_H : \mathbb{R} \to \mathbb{R}^3$ defined in Example 5. Since no level-set of $L_H : \mathbb{R} \to \mathbb{R}^3$ crosses from one target cell to another, IE holds. By Theorem 1 calibration follows.

*Proof sketch.* Since $\gamma := \mathrm{prop}[\ell]$ is orderable, there is an orderable enumeration $(r_1, \ldots, r_k)$ of reports (Def. 5). By Theorem 11 of Finocchiaro et al. [2020], there exist vectors $v_1, \ldots, v_{k-1} \in \mathbb{R}^n$ such that *(i)* for each $j \in [k-1]$, $\langle p, v_j \rangle = 0$ for all $p \in \gamma_{r_j} \cap \gamma_{r_{j+1}}$, and *(ii)* the coordinates are monotone, i.e., $v_{i,y} \leq v_{i+1,y}$ for every $i \in [k-2]$ and $y \in [n]$.

Let $V \in \mathbb{R}^{n \times (k-1)}$ have columns $v_1, \ldots, v_{k-1}$, and write $V^j$ for the $j$-th row. For each $j \in [n]$, define $L(\cdot)_j := \texttt{LinIntGrad}(V^j)$ (Subroutine 1, Appendix H). The subroutine first specifies a map $g_j : \mathbb{R} \to \mathbb{R}$ on $[1, k-1]$ by linear interpolation of the values $V^j[1], \ldots, V^j[k-1]$, so that $g_j(i) = V^j[i]$ for all integers $i \in \{1, \ldots, k-1\}$; hence $g_j$ is continuous and nondecreasing on $(1, k-1)$. It then extends $g_j$ outside $[1, k-1]$. In particular, for $x \leq 1, g_j(x) = X[1] + (x-1)$. And for $x \geq k-1$, $g_j(x) = X[k-1] + (x - (k-1))$. The construction ensures that continuity and monotonicity of $g_j$ are preserved across $\mathbb{R}$. Furthermore, $g_j$ crosses 0 either at a singleton, or at a compact interval. Finally, the subroutine sets $L(u)_j = \int_1^u g_j(t)\, dt$, so $(L(\cdot)_j)' = g_j$. Lemma 37 proves that each $L(\cdot)_j$ is convex, belongs to $C^1(\mathbb{R})$, has nonempty compact minimizers (the sets $\{g_j^{-1}(0) | j \in [n]\}$), and so $L$ satisfies Assumption 1. Moreover, at integers $i \in \{1, \ldots, k-1\}$ we have $\nabla L(i) = v_i$. In Theorem 13 of Appendix H, we show that these properties imply $L$ indirectly elicits $\ell$. Hence $L$ is calibrated with respect to $\ell$ by Theorem 1. $\qquad\square$

As an application of Theorem 4, we present a novel surrogate for the ordinal regression target loss in Example 5. While previous works have proposed surrogates for ordinal regression [Ramaswamy and Agarwal, 2016, Pedregosa et al., 2017, Finocchiaro et al., 2019], none of the surrogates therein are simultaneously convex, differentiable, minimizable and 1-dimensional.

**Example 5** (Huber-like surrogate for ordinal regression). *Here $\mathcal{Y} = \mathcal{R} = \{1, 2, 3\}$. Predictions farther away from the true outcome are more heavily penalized. The 3-class ordinal regression loss is $\ell^{\mathrm{ord}}(y, r) := |y - r|$, for $y, r \in \{1, 2, 3\}$. Then an application of Theorem 4 yields the surrogate*

$$L_H : \mathbb{R} \to \mathbb{R}^3; L_H(x) = [f(x-2), h(x), f(x+2)],$$

*where $h(x) = \frac{x^2}{2}$ and $f(x) = \frac{x^2}{2}$ for $-1 \leq x \leq 1$ and $f(x) = |x| - 0.5$ otherwise.*

*$L_H$ indirectly elicits $\ell^{\mathrm{ord}}$ and is therefore calibrated with respect to it. Figure 4 depicts the target (peach colored lines) and surrogate (blue dotted lines) level-sets for $\ell^{\mathrm{ord}}$ and $L_H$. The target $\ell^{\mathrm{ord}}$ poses a non-trivial challenge. In particular, for a 1-d convex, differentiable surrogate $L$ to indirectly elicit $\ell^{\mathrm{ord}}$, it must admit a non-unique minimizer at $(0, 1/2, 0)$ since the two target-boundaries intersect at this point. On the other hand, the minimizers elsewhere must be unique. $L_H$ is precisely such a minimizer. For $L_H$, $\Gamma((1/2, 0, 1/2)) = [-1, 1]$, whereas for every other $p \in \Delta_3$, $|\Gamma(p)| = 1$.*

## 5 Discussion and Future Directions

Our results are the first to establish general calibration conditions on the widely used class of convex differentiable surrogate losses in relation to arbitrary discrete target losses. We anticipate that the generality of our results will aid further advances in application and theory. Our conditions are inspired by the equivalence of IE and calibration for polyhedral surrogates. Like IE, strong IE is

substantially easier to verify than checking calibration directly. Hence, strong IE for differentiable losses could play a similar role to IE for polyhedral losses, where IE has been used to establish convex calibration dimension bounds [Ramaswamy and Agarwal, 2016] and to design and analyze numerous surrogates [Finocchiaro et al., 2022b,a, Wang and Scott, 2020, Nueve et al., 2024]. Indeed, we already make first steps in regards to design, by proposing a generalized construction for designing differentiable 1-dimensional surrogates for orderable targets.

**Lower bounds.** A promising direction for future work is to use strong IE to study prediction dimension. We believe it can establish lower bounds on the prediction dimension of calibrated surrogates for important target losses. Finocchiaro et al. [2021] leverage IE as a tool to establish such lower bounds. Recall that strong IE is necessary for calibration for the class of strongly convex, differentiable surrogate. At the same time, strong IE imposes more stringent constraints on surrogates than IE. We therefore believe strong IE offers promise to establish novel lower bounds in this setting.

**Relaxing Assumption 1.** Theorems 1 and 2 assume that $\arg\min_{u \in \mathbb{R}^d} L(\cdot)_y$ is non-empty and compact for each $y \in [n]$. Theorem 8 in Appendix C shows that Assumption 1 is equivalent to the condition that $\Gamma(p)$ is non-empty and compact for every $p \in \Delta_n$. The non-emptiness, i.e., minimizability of the functions $\{\langle p, L(\cdot)\rangle | p \in \Delta_n\}$ is mathematically well-motivated. Indeed, if minimizability fails for some distribution $p \in \Delta_n$, then $\Gamma(p)$ is empty. In this case, checking calibration at $p$ necessitates analyzing sequences of form $\{u_t\}_{t \in \mathbb{N}_+}$ such that $\lim_{t \to \infty} \langle p, L(u_t)\rangle = \inf_{u \in \mathbb{R}^d} \langle p, L(u)\rangle$. Thus, while understanding calibration for non-minimizable losses is an important and interesting direction in its own right, IE and strong IE are not the appropriate tools to do so. We speculate instead that the recently developed theory on astral spaces [Dudík et al., 2022] can be leveraged for this direction. Our assumption of compactness on $\Gamma(p)$ is technical and necessary for our proof approach, but may not be strictly necessary for strong IE to yield calibration. Differentiable surrogates with unbounded (and thus non-compact) minimizers are common in practice (for example: the squared hinge loss, the modified Huber loss, etc.) Relaxing this assumption is therefore a valuable direction for future research and could potentially enhance the practical appeal of strong IE.

## Acknowledgments and Disclosure of Funding

We thank Mabel Cluff for early discussions and insights on this project. We thank Stephen Becker for discussions on convergence properties of set-valued functions. We thank Jessie Finocchiaro and Enrique Nueve for discussions on 1d surrogate losses. This material is based upon work supported by the National Science Foundation under Grant No. IIS-2045347.

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

## A   Examples

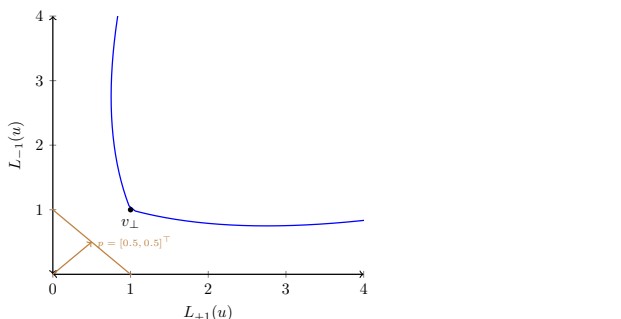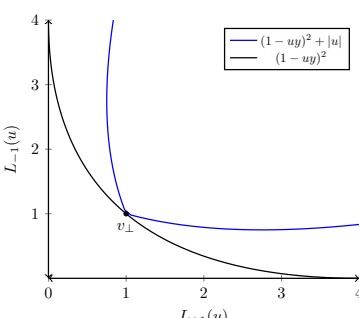

Figure 5: The figure on the left plots the superprediction set, $\{L(u) : u \in \mathbb{R}_+\}$, of a surrogate loss that IEs but is not calibrated. To see non-calibration, notice that there is only one possible link. Fix $p = [0.5, 0.5]^\top$. The optimal loss is achieved by $\arg\inf_{v \in \{L(u):u \in \mathbb{R}_+\}} \langle v, p \rangle$. Thus the loss is optimized by $v_\perp$ which must link to the abstain report $\perp$. However, consider the points to the left of $v_\perp$, which link to $+1$. The infimum of the loss over these points for $p$ is equal to the loss of $v_\perp$, thus violating calibration.

**Example 6.** *Let the target loss be binary classification with abstain level $\frac{1}{4}$, $L_{smooth} : \mathbb{R} \to \mathbb{R}^2$.*

$$L_{smooth}(u)_y := (1 - uy)^2. \tag{1}$$

*This is the smooth loss plotted on the right side of Figure 5. For any $p \in [0, 1]$, $\Gamma((p, 1-p)) = 2p-1$. Hence $\Gamma_u = \{(\frac{1+u}{2}, \frac{1-u}{2})\}$. Clearly, each component of $L_{smooth}$ is strongly convex and differentiable. Thus, if $L_{smooth}$ strongly indirectly elicits $\ell$, then $L_{smooth}$ is calibrated with respect to $\ell$ by Theorem 3. Since $\Gamma_u$ is a singleton, strong IE follows trivially by definition, and so $L_{smooth}$ is calibrated with respect to $\ell$. "Smoothening out" $L_{cusp}$ from Example 1 thus resolves calibration.*

**Example 7** (Convex Combinations of Huber losses). *For $u \in \mathbb{R}^d$, let*

$$f_H(u) = \begin{cases} \|u\|_2 - \frac{1}{2} & \|u\|_2 \geq 1 \\ \frac{1}{2}\|u\|_2^2 & \|u\|_2 \leq 1 \end{cases}$$

*be the Huber loss in $\mathbb{R}^d$. Define*

$$L_H(u) = \begin{pmatrix} f_H(u + (2, 0, \ldots, 0)) \\ f_H(u - (2, 0, \ldots, 0)) \end{pmatrix}$$

*be the sum of two Huber losses. Then, for $p = 1/2$, $\Gamma(p) = [-1, 1] \times \{0\}^{d-1}$. For $p > 1/2$, $\Gamma(p)$ is a single point in $(1, 2] \times \{0\}^{d-1}$. For $p < 1/2$, $\Gamma(p)$ is a single point in $[-2, -1) \times \{0\}^{d-1}$. We can see this visually for $d = 1$ in Figure 6.*

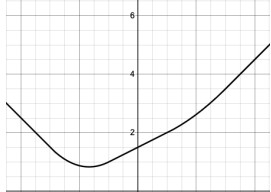 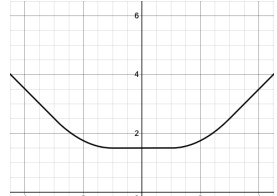 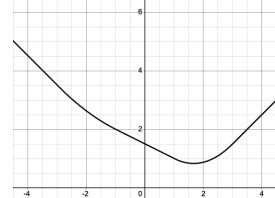

Figure 6: The plot of $\langle p, L_H(u) \rangle$ for 3 different values of $p$ with $d = 1$. In higher dimensions, the loss is minimized by exactly the same points, since by construction the minima will always lie on the $u_1$-axis. **Left:** $p = 1/4$, the $\langle p, L_H(u) \rangle$ has a unique minimum at $u = -5/3$. **Center:** $p = 1/2$, the $\langle p, L_H(u) \rangle$ is minimized by any choice of $u \in [-1, 1]$. **Left:** $p = 3/4$, the $\langle p, L_H(u) \rangle$ has a unique minimum at $u = 5/3$.

**Lemma 1.** $\mathrm{rank}(\nabla L_{CE}(u))$ *is 1 when* $u = (0, 0)$ *and 2 otherwise.*

*Proof.*

$$\nabla L_{\mathrm{CE}}(u) = \begin{pmatrix} 2u_1 + 1 & 2u_2 + 2 \\ 4u_1 + 1 & 4u_2 + 2 \\ 6u_1 - 1 & 2u_2 - 2 \end{pmatrix} .$$

Let $v_1$ and $v_2$ denote the first and second column of $\nabla L_{\mathrm{CE}}(u)$ respectively. $\mathrm{rank}(\nabla L_{\mathrm{CE}}(u)) = 0$ if and only if $v_1 = v_2 = 0$. However, there is no choice of $u_1$ or $u_2$ such that either $v_1$ or $v_2$ are 0, so $\mathrm{rank}(\nabla L_{\mathrm{CE}}(u)) \geq 1$ everywhere.

$\mathrm{rank}(\nabla L_{\mathrm{CE}}(u)) = 1$ if and only if $\lambda v_1 = v_2$ for some $\lambda \in \mathbb{R}$. For this to hold for the first row of $\nabla L_{\mathrm{CE}}(u)$ we must have $2\lambda u_1 + \lambda = 2u_2 + 2$, so $u_2 = \lambda u_1 + \frac{\lambda}{2} - 1$. Similarly, using the second row of $\nabla L_{\mathrm{CE}}(u)$ implies $u_2 = \lambda u_1 + \frac{\lambda}{4} - \frac{1}{2}$. This gives two equivalent expressions for $u_2$, so we must have $\frac{\lambda}{2} - 1 = \frac{\lambda}{4} - \frac{1}{2}$, so $\lambda = 2$. Plugging this into either of the expressions for $u_2$ yields $u_2 = 2u_1$. Finally, using these values of $\lambda$ and $u_2$ the last row of $\nabla L_{\mathrm{CE}}(u)$ becomes $12u_1 - 2 = 4u_1 - 2$, so $u_1 = 0$, and thus $u_2 = 2u_1 = 0$ as well. Therefore, $\mathrm{rank}(\nabla L_{\mathrm{CE}}(u)) = 1$ only when $u = (0, 0)$.

For any other value of $u$, we then have $\mathrm{rank}(\nabla L_{\mathrm{CE}}(u)) \geq 2$, but since $u \in \mathbb{R}^2$, $\mathrm{rank}(\nabla L_{\mathrm{CE}}(u)) \leq 2$ everywhere. Therefore, for all $u \neq (0, 0)$, $\mathrm{rank}(\nabla L_{\mathrm{CE}}(u)) = 2$. $\square$

# B    Property Elicitation, Level Sets and Minimizing Sets

**Lemma 2.** *Let* $A, B \subseteq \Delta_n : A \subset B$. *Then* $\Gamma(A) \subseteq \Gamma(B)$

*Proof.* Since $A \subseteq B$, we have that $\forall a \in A, a \in B$. So, $\forall a \in A, \Gamma(a) \subseteq \cup_{b \in B} \Gamma(b) = \Gamma(B)$. Thus, $\Gamma(A) = \cup_{a \in A} \Gamma(a) \subseteq \Gamma(B)$ $\square$

**Lemma 3.** *Consider any convex, differentiable* $L : \mathbb{R}^d \to \mathbb{R}^n$. *Let* $p \in \Delta_n$ *and* $u \in \mathbb{R}^d$. *Then,* $u \in \Gamma(p) \iff \nabla L(u)^\top p = \mathbf{0}_d$. *Equivalently,* $\Gamma_u = \{p \subseteq \Delta_n | \nabla L(u)^\top p = \mathbf{0}_d\}$.

*Proof.* Since $L_y$ is convex for every $y \in [n]$, it follows that the function $\langle p, L(\cdot) \rangle : \mathbb{R}^d \to \mathbb{R}$ is convex for any $p \in \Delta_n$. Now, since the domain of $\langle p, L(\cdot) \rangle$ is open and the minimum is attained at some $u \in \mathbb{R}^d$, it follows that $\nabla \langle p, L(u) \rangle = \mathbf{0}_d \implies \nabla L(u)^\top p = \mathbf{0}_d$. Conversely, if $\nabla \langle p, L(u) \rangle = \mathbf{0}_d$, then $u \in \Gamma(p)$ by convexity of $\langle p, L(\cdot) \rangle$. $\square$

**Lemma 4.** *Any convex, differentiable surrogate* $L : \mathbb{R}^d \to \mathbb{R}^n$ *that indirectly elicits a target* $\ell : \mathcal{R} \to \mathbb{R}^n$ *satisfies* $\mathrm{rank}(\nabla L(u)^\top) > 0$ *for every* $u \in \mathbb{R}^d$.

*Proof.* Assume to the contrary, i.e., $\exists u \in \mathbb{R}^d$, such that, $rank(\nabla L(u)^\top) = 0$. By the rank-nullity theorem, this means $null(\nabla L(u)^\top) = n$. So $\nabla L(u)^\top p = \mathbf{0}_d$ is satisfied by any $p \in \Delta_n$. From Lemma 3, we get that $\Gamma_u = \Delta_n$. We assume $k = |\mathcal{R}| \geq 2$. If not, the prediction problem is trivial, so $\{1, 2\} \subseteq \mathcal{R}$ for any problem of interest. First consider the pair of discrete reports $(1, 2)$. By the non-redundancy of discrete reports, $\exists p_1 \in \gamma_1, p_2 \in \gamma_2$, such that $\langle p_1, \ell(\cdot) \rangle$ is uniquely optimized by the discrete prediction 1 and $\langle p_2, \ell(\cdot) \rangle$ is uniquely optimized by the discrete prediction 2. Since

$\Gamma_u = \Delta_n$, it follows that $p_1, p_2 \in \Gamma_u$. This means $\Gamma_u \not\subseteq \gamma_1$, since $p_2 \notin \gamma_1$, and similarly $\Gamma_u \not\subseteq \gamma_2$. Similarly, for any $j \in \mathcal{R} : j \notin \{1, 2\}$, consider the reports $(1, j)$ and repeat the same rationale to establish that $\Gamma_u \not\subseteq \gamma_j$. Thus, $\exists u \in \mathbb{R}^d$ such that $\Gamma_u \not\subseteq \gamma_r, \forall r \in \mathcal{R}$, implying $L$ does not indirectly elicit $\ell$ by the definition of indirect elicitation. $\qquad\square$

**Lemma 5.** *Let $L : \mathbb{R}^d \to \mathbb{R}^n$ be a convex, but not necessarily differentiable surrogate. Then for any $u \in \mathbb{R}^d$, $\Gamma_u$ is compact.*

*Proof.* Pick any $u \in \mathbb{R}^d$. Since $\Gamma_u \subseteq \Delta_n$ and $\Delta_n$ is compact, it is clear that $\Gamma_u$ is bounded. So it suffices to show that $\Gamma_u$ is closed. Let $\{p_t\}_{t \in \mathbb{N}_+} \subseteq \Gamma_u$ and suppose that $p_t \to p$. We want to show that $p \in \Gamma_u$. First note that since $p_t \in \Delta_n$ for each $t$, it follows that $p \in \Delta_n$ by compactness of $\Delta_n$. Now, pick any $v \in \mathbb{R}^d$. It holds for each $t \in \mathbb{N}_+$ that $\langle p_t, L(u) \rangle \leq \langle p_t, L(v) \rangle$. Taking the limit as $t \to \infty$ on both sides, it holds that $\langle p, L(u) \rangle \leq \langle p, L(v) \rangle$ for any $v \in \mathbb{R}^d$. Thus, $u \in \Gamma(p) \implies p \in \Gamma_u$. Hence, $\Gamma_u$ is closed. $\qquad\square$

**Theorem 5.** *Let $L : \mathbb{R}^d \to \mathbb{R}^n$ be a surrogate that strongly indirectly elicits a target loss $\ell : \mathcal{R} \to \mathbb{R}^n$. Then $L$ indirectly elicits $\ell$.*

*Proof.* Pick any $u \in \mathbb{R}^d$. By definition, there exists some $S \subseteq \mathcal{R}$, such that $\gamma(p) = S$ for every $p \in \Gamma_u$. Equivalently, $p \in \cap_{r \in S} \gamma_r$ for every $p \in \Gamma_u$. Thus, $\Gamma_u \subseteq \cap_{r \in S} \gamma_r \implies \exists r \in \mathcal{R}$, such that $\Gamma_u \subseteq \gamma_r$. $\qquad\square$

**Theorem 6.** *Let $L : \mathbb{R}^d \to \mathbb{R}^n$ be a surrogate that is calibrated with respect to $\ell : \mathcal{R} \to \mathbb{R}^n$. Then $L$ indirectly elicits $\ell$.*

*Proof.* Since $L$ is calibrated with respect to $\ell$, there exists a link function $\psi : \mathbb{R}^d \to \mathcal{R}$, such that, $(L, \psi)$ is calibrated with respect to $\ell$. Suppose $u \in \mathbb{R}^d$. If $u \notin \Gamma(\Delta_n)$, then $\Gamma_u = \emptyset \implies \Gamma_u \subseteq \gamma_r$, $\forall r \in \mathcal{R}$ yielding indirect elicitation. Now, suppose $u \in \Gamma(\Delta_n)$. We show that $\Gamma_u \subseteq \gamma_{\psi(u)}$. Assume to the contrary. Then, there exists some $p \in \Gamma_u$, such that $p \notin \gamma_{\psi(u)} \implies \psi(u) \notin \gamma(p)$. Then, $\inf_{v \in \mathbb{R}^d : \psi(v) \notin \gamma(p)} \langle p, L(v) \rangle \leq \langle p, L(u) \rangle = \inf_{v \in \mathbb{R}^d} \langle p, L(v) \rangle$ since $p \in \Gamma_u$, hence violating calibration. Thus, $\Gamma_u \subseteq \gamma_{\psi(u)}$ and so $(L, \psi)$ indirectly elicit $\ell \implies L$ indirectly elicits $\ell$. $\qquad\square$

**Theorem 7.** *Let $L : \mathbb{R}^d \to \mathbb{R}^n$ be a polyhedral surrogate that indirectly elicits some target loss $\ell : \mathcal{R} \to \mathbb{R}^n$. Then $L$ does not strongly indirectly elicit $\ell$.*

*Proof.* We know from [Finocchiaro et al., 2024] that any polyhedral surrogate has a finite representative set $S$, i.e., $S \subset \mathbb{R}^d$, such that $S$ has a finite number of elements and that for any $p \in \Delta_n$, there exists some $u \in S$, such that $p \in \Gamma_u$. We leverage this fact to prove our claim. Assume by contradiction that $L$ strongly indirectly elicits $\ell$. Suppose WLOG that $S = \{u_1, u_2, ..., u_m\}$. Since $\cup_{i \in [m]} \Gamma_{u_i} = \Delta_n$, it follows that there exists some $S' \subseteq S$, such that $\text{relint}(\gamma_1) \subseteq \cup_{v \in S'} \Gamma_v$. In particular, $S' \subseteq S$ and the level-sets of the reports in $S'$ cover $\text{relint}(\gamma_1)$. Since $S$ is finite, there must exist a minimal subset of $S$, the level sets of whose elements cover $\text{relint}(\gamma_1)$. Assume $S'$ is such a minimal covering subset. First, we claim that for any $v \in S', p \in \Gamma_v \implies \gamma(p) \cap \{1\} \neq \emptyset$. Assume not. Then $\exists v \in S'$, such that $\gamma(p) \cap \{1\} = \emptyset$ for some $p \in \Gamma_v$. By strong IE, it holds that $\gamma(p) \cap \{1\} = \emptyset, \forall p \in \Gamma_v$. Thus, if $v \in S'$, $S'$ can't be minimal. Next, we claim that for any $v \in S', p \in \Gamma_v \implies \gamma(p) = \{1\}$. Assume not, then $\exists v \in S' : \{1\} \subset \gamma(p)$ for some $p \in \Gamma_v$. By strong IE, it follows that $\forall p \in \Gamma_v, \{1\} \subset \gamma(p) \implies \forall p \in \Gamma_v, p \notin \text{relint}(\gamma_1)$. Thus, if $v \in S'$, $S'$ can't be minimal. Hence, for every $v \in S', p \in \Gamma_v \implies \gamma(p) = \{1\}$. Therefore, $\text{relint}(\gamma_1) \not\subset \cup_{v \in S'} \Gamma_v \implies \text{relint}(\gamma_1) = \cup_{v \in S'} \Gamma_v$. However, $S'$ being a subset of finite $S$ is itself finite, and by Lemma 5, $\cup_{v \in S'} \Gamma_v$ is a finite union of closed sets implying that $\cup_{v \in S'} \Gamma_v$ itself must be closed. On the other hand, $\text{relint}(\gamma_1)$ is not closed by definition and so $\text{relint}(\gamma_1) \neq \cup_{v \in S'} \Gamma_v$. Hence, $L$ strongly indirectly eliciting $\ell$ yields a contradiction. $\qquad\square$

**Lemma 6.** *Consider any convex, differentiable $L : \mathbb{R}^d \to \mathbb{R}^n$. Suppose $u \in \mathbb{R}^d$. Then $\Gamma_u$ is a polytope. If $L : \mathbb{R}^d \to \mathbb{R}^n$ indirectly elicits any target $\ell : \mathcal{R} \to \mathbb{R}^n$, then affdim$(\Gamma_u) \leq n - 2$.*

*Proof.* Recall by Lemma 3 that $\Gamma_u = \{p \in \Delta_n | \nabla L(u)^\top p = \mathbf{0}_d\}$. In other words, $\Gamma_u = \Delta_n \cap \ker(\nabla L(u)^\top)$. So $\Gamma_u$ is the intersection of a polytope and a subspace, implying that $\Gamma_u$ is itself a polytope [Henk et al., 2017].

Next, suppose $L$ indirectly elicits some target $\ell$. Then by Lemma 4, it holds that

$d \geq \text{rank}(\nabla L(u)^\top) > 0$. Thus, $1 \leq \text{nullity}(L(u)^\top) < n$. So, $\Gamma_u$ is the intersection of a set of affine dimension $n-1$ (i.e., $\Delta_n$) and a subspace of dimension at least 1 (i.e., $\ker(\nabla L(u))^\top$). Thus, $\text{affdim}(\Gamma_u) \leq n - 2$. $\qquad\square$

**Lemma 7.** *Suppose $\ell : \mathcal{R} \to \mathbb{R}^n$ is an elicitable target loss, and that $\mathcal{Y}$ and $\mathcal{R}$ are finite sets. Suppose further that each $r \in \mathcal{R}$ is non-redundant. Then for any $r \in \mathcal{R}$, $\gamma_r$ is a convex polytope, such that $\text{affdim}(\gamma_r) = n - 1$.*

*Proof.* This can be observed directly from the fact that any finite target is elicitable if and only its cells $\gamma_r$ (where, $r \in \mathcal{R}$) form a power diagram [Lambert et al., 2008]. Power diagrams are essentially weighted Voronoi diagrams. For more details on power diagram, we refer the reader to [Aurenhammer, 1987]. $\qquad\square$

**Lemma 8.** *Let $L : \mathbb{R}^d \to \mathbb{R}^n$ be a convex, differentiable surrogate. Let $\ell : \mathcal{R} \to \mathbb{R}^n$. Let $u \in \mathbb{R}^d$. Suppose $p, p' \in \Gamma_u$ and that $\exists S \subseteq R$ such that $\gamma(p) \cap \gamma(p') = S \neq \emptyset$. Then, $\gamma(q) \subseteq S$, where $q := \frac{p+p'}{2}$.*

*Proof.* First note that by convexity of $\Gamma_u$, $q \in \Gamma_u \implies q \in \Delta_n$. For $i \in \mathcal{R}$, denote $\ell_i := (\ell(1,i), \ell(2,i), ..., \ell(n,i))$ as the loss vector corresponding to prediction $i$. Say $j \in \gamma(q) \implies q^\top \ell_j \leq q^\top \ell_i, \forall i \in \mathcal{R}$. Now, suppose $j \notin S$. Let $t \in S \implies t \in \gamma(p) \cap \gamma(p')$. So, $p'^\top \ell_t \leq p'^\top \ell_j$ and $p^\top \ell_t \leq p^\top \ell_j$, with at least one inequality strict (as if both were equalities, then $j \in S$). So, summing the strict inequality with the other inequality, we get that $(p' + p)^\top \ell_t < (p' + p)^\top \ell_j \implies \frac{(p'+p)}{2}^\top \ell_t < \frac{(p'+p)}{2}^\top \ell_j \implies q^\top \ell_t < q^\top \ell_j$, which contradicts our supposition that $j \in \gamma(q)$. Thus, $j \notin S \implies \gamma(q) \subseteq S$. $\qquad\square$

**Lemma 9.** *Let $L : \mathbb{R}^d \to \mathbb{R}^n$ be a convex, differentiable surrogate. Let $\ell : \mathcal{R} \to \mathbb{R}^n$. Let $u \in \mathbb{R}^d$, such that, $\gamma(p) \cap \gamma(p') \neq \emptyset$ for any $p, p' \in \Gamma_u$. Let $p_m \in \Gamma_u : |\gamma(p_m)| \leq |\gamma(p)|$ for every $p \in \Gamma_u$. Then, $\gamma(p_m) \subseteq \gamma(p), \forall p \in \Gamma_u$.*

*Proof.* Suppose not. We know that $\gamma(p_m) \cap \gamma(p) \neq \emptyset$, so $\exists S \subseteq R : \gamma(p_m) \cap \gamma(p) = S \neq \emptyset$. Clearly, $S \subset \gamma(p_m)$ and $S \subset \gamma(p)$, since $S \neq \gamma(p_m)$. Now, pick $q = \frac{p+p_m}{2}$. Since $\Gamma_u$ is convex, $q \in \Gamma_u$. Now, we know from Lemma 8 that $\gamma(q) \subseteq S \implies |\gamma(q)| \leq |S| < |\gamma(p_m)| \implies |\gamma(q)| < |\gamma(p_m)|$. However, since $q \in \Gamma_u$, this yields a contradiction. $\qquad\square$

**Lemma 10.** *Let $L : \mathbb{R}^d \to \mathbb{R}^n$ be a convex, differentiable surrogate. Let $\ell : \mathcal{R} \to \mathbb{R}^n$. $L$ indirectly elicits $\ell$ if and only if $\forall u \in \mathbb{R}^d$, it holds that $\gamma(p) \cap \gamma(p') \neq \emptyset$, for any $p, p' \in \Gamma_u$.*

*Proof.* We first show the $\implies$ direction. Since $L$ indirectly elicits $\ell$, it holds that $\forall u \in \mathbb{R}^d, \exists r \in \mathcal{R}$, such that $\Gamma_u \subseteq \gamma_r$. Pick any $p, p' \in \Gamma_u$. Clearly, $p, p' \in \gamma_r \implies r \in \gamma(p) \cap \gamma(p') \implies \gamma(p) \cap \gamma(p') \neq \emptyset$, for any $p, p' \in \Gamma_u$.

We now prove the $\impliedby$ direction. Suppose that for any $u \in \mathbb{R}^d$, it holds that $\gamma(p) \cap \gamma(p') \neq \emptyset$ for any $p, p' \in \Gamma_u$. Let $p_m \in \Gamma_u : |\gamma(p_m)| \leq |\gamma(p)|, \forall p \in \Gamma_u$. We know from Lemma 9, that $\gamma(p_m) \subseteq \gamma(p), \forall p \in \Gamma_u$. This implies that $\exists r \in \mathcal{R} : r \in \gamma(p_m) \implies r \in \gamma(p), \forall p \in \Gamma_u \implies \exists r \in \mathcal{R} : p \in \gamma_r, \forall p \in \Gamma_u \implies \exists r \in \mathcal{R} : \Gamma_u \subseteq \gamma_r$. $\qquad\square$

**Lemma 11.** *Let $L : \mathbb{R}^d \to \mathbb{R}^n$ be a convex, differentiable surrogate. Let $\mathcal{C}_u \subseteq \Gamma_u$ be the set of corners of $\Gamma_u$. Let $\ell : \mathcal{R} \to \mathbb{R}^n$. Suppose $r \in \mathcal{R}$. Then, $p \in \gamma_r$ for every $p \in \mathcal{C}_u \iff \Gamma_u \subseteq \gamma_r$.*

*Proof.* Recall from Lemma 6 that for any $u \in \mathbb{R}^d$, $\Gamma_u$ is a polytope. Thus, a finite set $\mathcal{C}_u \subseteq \Gamma_u$ exists such that $\mathcal{C}_u$ are the corners of $\Gamma_u$. Say $\exists r \in \mathcal{R}$, such that $p \in \gamma_r, \forall p \in \mathcal{C}_u$. Pick any $q \in \Gamma_u$. Clearly, $q \in \text{conv}(\mathcal{C}_u) \subseteq \gamma_r$, as $\gamma_r$ is convex by Lemma 7. Thus, $q \in \gamma_r \implies \Gamma_u \subseteq \gamma_r$. For the reverse direction, suppose $\Gamma_u \subseteq \gamma_r$. Let $p \in \mathcal{C}_u \subseteq \Gamma_u$. Then $p \in \gamma_r$. $\qquad\square$

**Lemma 12.** *Let $L : \mathbb{R}^d \to \mathbb{R}^n$ be a convex, differentiable surrogate. Let $\mathcal{C}_u \subseteq \Gamma_u$ be the set of corners of $\Gamma_u$ for any $u \in \mathbb{R}^d$. Let $\ell : \mathcal{R} \to \mathbb{R}^n$. If $\exists S \subseteq \mathcal{R} : \gamma(p) = S$ for every $p \in \mathcal{C}_u$ then $L$ strongly indirectly elicits $\ell$.*

*Proof.* Suppose $\gamma(p) = S \subseteq R$, for every $p \in \mathcal{C}_u$. This implies that $p \in \mathrm{relint}(\cap_{r \in S} \gamma_r)$ for each $p \in \mathcal{C}_u \implies \mathcal{C}_u \subseteq \mathrm{relint}(\cap_{r \in S} \gamma_r)$. Since $\gamma_r$ is a convex polytope for each $r \in S$, the set $\cap_{r \in S} \gamma_r$ is convex and so is the set $\mathrm{relint}(\cap_{r \in S} \gamma_r)$. Hence, $\mathrm{conv}(\mathcal{C}_u) \subseteq \mathrm{relint}(\cap_{r \in S} \gamma_r) \implies \Gamma_u \subseteq \mathrm{relint}(\cap_{r \in S} \gamma_r) \implies p \in \mathrm{relint}(\cap_{r \in S} \gamma_r), \forall p \in \Gamma_u \implies \gamma(p) = S, \forall p \in \Gamma_u \implies \gamma(p) = \gamma(p'), \forall p, p' \in \Gamma_u \implies$ strong indirect elicitation is satisfied. $\square$

**Proposition 1.** *Let $L : \mathbb{R}^d \to \mathbb{R}^n$ be a convex, differentiable surrogate and let $\ell : \mathcal{R} \to \mathbb{R}^n$ be a target. Let $u \in \mathbb{R}^d$ be a report and suppose $\mathcal{C}_u$ is the finite set of corners for $\Gamma_u$. Then the set $\{\gamma(p) | p \in \mathcal{C}_u\}$ suffices to check both indirect elicitation and strong indirect elicitation at $u$.*

*Proof.* The proof follows by Lemmas 11 and 12. $\square$

## C   Properties of convex, differentiable functions

**Lemma 13.** *Let $f : \mathbb{R}^d \to \mathbb{R}$ be a convex, differentiable function. Then $\mathrm{argmin}_{u \in \mathbb{R}^d} f(u)$ is convex.*

*Proof.* If $\mathrm{argmin}_{u \in \mathbb{R}^d} f(u) = \emptyset$, the result follows vacuously. Else, suppose $f^* = \min_{u \in \mathbb{R}^d} f(u)$ and that $x, y \in \mathrm{argmin}_{u \in \mathbb{R}^d} f(u)$. Then for any $\lambda \in [0, 1]$, $f(\lambda x + (1 - \lambda)y) \leq \lambda \cdot f(x) + (1 - \lambda) \cdot f(y) = \lambda \cdot f^* + (1 - \lambda) \cdot f^* = f^* \implies \lambda \cdot x + (1 - \lambda) \cdot y \in \mathrm{argmin}_{u \in \mathbb{R}^d} f(u), \forall \lambda \in [0, 1]$. $\square$

**Lemma 14.** *Let $f : \mathbb{R}^d \to \mathbb{R}$ be a convex finite function on $\mathbb{R}^d$. Then $\mathrm{argmin}_{u \in \mathbb{R}^d} f(u)$ is closed. If $\mathrm{argmin}_{u \in \mathbb{R}^d} f(u)$ is bounded, $\mathrm{argmin}_{u \in \mathbb{R}^d} f(u)$ is compact.*

*Proof.* By [Rockafellar, 1970, Corollary 10.1.1] $f$ is continuous. By [Hiriart-Urruty and Lemaréchal, 1996, Prop 3.2.2] every sublevel-set of $f$ is closed. $\square$

**Lemma 15.** *Let $f : \mathbb{R}^d \to \mathbb{R}$ be a convex, differentiable function. Then $f$ is continuously differentiable.*

*Proof.* See Corollary 25.5.1 of [Rockafellar, 1970]. $\square$

**Lemma 16.** *Let $f : \mathbb{R}^d \to \mathbb{R}$ be a convex, differentiable function. Suppose also that $f$ is minimizable and that the set $\mathrm{argmin}_{u \in \mathbb{R}^d} f(u)$ is bounded. Let $\mathcal{U}^* := \mathrm{argmin}_{u \in \mathbb{R}^d} f(u)$ and let $f^* := \min_{u \in \mathbb{R}^d} f(u)$. Then, for $\delta > 0$, it holds that:*

$$\inf_{u \in \mathbb{R}^d \setminus B_\delta(\mathcal{U}^*)} f(u) = \inf_{u \in \partial \bar{B}_\delta(\mathcal{U}^*)} f(u) = \min_{u \in \partial \bar{B}_\delta(\mathcal{U}^*)} f(u) > f^*$$

*Proof.* First, notice that since $\mathcal{U}^*$ is bounded, it is compact by Lemma 14. Thus, $\bar{B}_\delta(\mathcal{U}^*) \setminus B_\delta(\mathcal{U}^*) := \partial \bar{B}_\delta(\mathcal{U}^*)$ is also compact. Since $f$ is differentiable, it is continuous everywhere, and thus $f$ attains its infimum over the compact set $\partial \bar{B}_\delta(\mathcal{U}^*)$. This proves that

$$\inf_{u \in \partial \bar{B}_\delta(\mathcal{U}^*)} f(u) = \min_{u \in \partial \bar{B}_\delta(\mathcal{U}^*)} f(u) > f^*,$$

where the final inequality holds as $\partial \bar{B}_\delta(\mathcal{U}^*) \cap \mathcal{U}^* = \emptyset$. We are left to show that

$$\inf_{u \in \mathbb{R}^d \setminus B_\delta(\mathcal{U}^*)} f(u) = \min_{u \in \partial \bar{B}_\delta(\mathcal{U}^*)} f(u).$$

Clearly, $\inf_{u \in \mathbb{R}^d \setminus B_\delta(\mathcal{U}^*)} f(u) \leq \min_{u \in \partial \bar{B}_\delta(\mathcal{U}^*)} f(u)$, so for the equality to fail, we would need $\inf_{u \in \mathbb{R}^d \setminus \bar{B}_\delta(\mathcal{U}^*)} f(u) < \min_{u \in \partial \bar{B}_\delta(\mathcal{U}^*)} f(u)$. This, in turn, requires that there exist $u' \in \mathbb{R}^d \setminus \bar{B}_\delta(\mathcal{U}^*)$ such that

$$f(u') < \min_{u \in \partial \bar{B}_\delta(\mathcal{U}^*)} f(u).$$

Pick any $u^* \in \mathcal{U}^*$ and consider the line segment $\mathrm{conv}(u^*, u')$, connecting $u^*$ and $u'$. There exists some $v \in \partial \bar{B}_\delta(\mathcal{U}^*)$ such that $v \in \mathrm{conv}(u^*, u')$. It holds that

$$f(v) > f(u') > f(u^*),$$

which violates convexity of $f$ since $v \in \mathrm{conv}(u^*, u')$, completing the proof.

$\square$

**Lemma 17.** *Let $f : \mathbb{R}^d \to \mathbb{R}$ be a convex, differentiable function. Suppose also that $f$ is minimizable and that the set $\mathcal{U}^* := \operatorname{argmin}_{u \in \mathbb{R}^d} f(u)$ is bounded. Let $f^* := \min_{u \in \mathbb{R}^d} f(u)$. Let $\{u_t\}_{t \in \mathbb{N}_+}$ be a sequence in $\mathbb{R}^d$. Then:*

$$\lim_{t \to \infty} d(u_t, \mathcal{U}^*) = 0 \iff \lim_{t \to \infty} f(u_t) = f^*$$

*Proof.* We first show that:

$$\lim_{t \to \infty} d(u_t, \mathcal{U}^*) = 0 \implies \lim_{t \to \infty} f(u_t) = f^*$$

Since $\mathcal{U}^*$ is bounded, it is compact by Lemma 14. Pick $\delta > 0$. There exists $T_\delta \in \mathbb{N}_+$ such that for every $t \geq T_\delta$,

$$d(u_t, \mathcal{U}^*) < \delta.$$

In particular, for each $t \geq T_\delta$, there exists $u_t^* \in \mathcal{U}^*$ such that

$$\|u_t - u_t^*\| < \delta.$$

This implies $u_t \in B_\delta(\mathcal{U}^*) \subseteq \bar{B}_\delta(\mathcal{U}^*)$ for every $t \geq T_\delta$. Since $\mathcal{U}^*$ is compact, $\bar{B}_\delta(\mathcal{U}^*)$ is also compact.

Now, $f$ is differentiable and therefore continuous everywhere. Since $\bar{B}_\delta(\mathcal{U}^*)$ is compact, $f$ is uniformly continuous within $\bar{B}_\delta(\mathcal{U}^*)$. Pick $\epsilon > 0$. It suffices to show the existence of some $T \in \mathbb{N}_+$ such that

$$|f(u_t) - f^*| < \epsilon, \quad \forall t \geq T.$$

By uniform continuity, there exists $\delta_\epsilon > 0$ such that

$$|f(u) - f(v)| < \epsilon, \quad \text{whenever } \|u - v\| < \delta_\epsilon \text{ and } u, v \in \bar{B}_\delta(\mathcal{U}^*).$$

If $\delta_\epsilon \geq \delta$, then for any $u_t$ with $t \geq T_\delta$,

$$\|u_t - u_t^*\| < \delta \leq \delta_\epsilon,$$

where $u_t^* \in \mathcal{U}^*$, implying $u_t, u_t^* \in \bar{B}_\delta(\mathcal{U}^*)$. Thus, for any $t \geq T_\delta$,

$$|f(u_t) - f(u_t^*)| = |f(u_t) - f^*| < \epsilon.$$

Otherwise, if $\delta_\epsilon < \delta$, pick $T_{\delta_\epsilon} \in \mathbb{N}_+$ such that $d(u_t, \mathcal{U}^*) < \delta_\epsilon$ for every $t \geq T_{\delta_\epsilon}$. Then, for each $t \geq T_{\delta_\epsilon}$, there exists $u_t^* \in \mathcal{U}^*$ such that

$$\|u_t - u_t^*\| < \delta_\epsilon, \quad u_t, u_t^* \in \bar{B}_\delta(\mathcal{U}^*).$$

Thus, for all $t \geq T_{\delta_\epsilon}$,

$$|f(u_t) - f(u_t^*)| = |f(u_t) - f^*| < \epsilon.$$

We now prove the reverse direction

$$\lim_{t \to \infty} f(u_t) = f^* \implies \lim_{t \to \infty} d(u_t, \mathcal{U}^*) = 0.$$

Assume to the contrary that this implication does not hold. Then, there exists some $\delta > 0$ such that for every $T \in \mathbb{N}_+$, there exists $t \geq T$ such that

$$d(u_t, \mathcal{U}^*) \geq \delta.$$

This implies the existence of a subsequence $\{u_{t_j}\}_{j \in \mathbb{N}_+}$ such that

$$d(u_{t_j}, \mathcal{U}^*) \geq \delta, \quad \forall j \in \mathbb{N}_+.$$

For every $j \in \mathbb{N}_+$,

$$f(u_{t_j}) \geq \inf_{u \in \mathbb{R}^d \backslash B_\delta(\mathcal{U}^*)} f(u) = \inf_{u \in \partial \bar{B}_\delta(\mathcal{U}^*)} f(u).$$

By Lemma 16,

$$\inf_{u \in \partial \bar{B}_\delta(\mathcal{U}^*)} f(u) = \min_{u \in \partial \bar{B}_\delta(\mathcal{U}^*)} f(u) > f^*.$$

Thus, for every $j \in \mathbb{N}_+$,

$$f(u_{t_j}) - f^* \geq \min_{u \in \partial \bar{B}_\delta(\mathcal{U}^*)} f(u) - f^* > 0.$$

This contradicts the assumption that $f(u_t) \to f^*$ as $t \to \infty$, completing the proof. $\qquad \square$

**Lemma 18.** *Let $f_1, f_2 : \mathbb{R}^d \to \mathbb{R}$ be convex finite functions such that $S_1 = \arg\min_{x \in \mathbb{R}^d} f_1(x)$ and $S_2 = \arg\min_{x \in \mathbb{R}^d} f_2(x)$ are compact and nonempty. Let $g(x) = f_1(x) + f_2(x)$. Then, $S = \arg\min_{x \in \mathbb{R}^d} g(x)$ is also compact and nonempty.*

*Proof.* By Lemma 14, $S$ is closed, so it suffices to show it is bounded and nonempty. Fix any $x_1 \in S_1$ and $x_2 \in S_2$. Let $y_1 = f_1(x_1)$ and $y_2 = f_2(x_2)$ be the minimia achieved by $f_1$ and $f_2$. Now, choose any $x_g \in \mathbb{R}^d$ and let $y = g(x_g)$. Note that by construction we must have $y \geq y_1 + y_2$. Therefore, let $\delta := y - (y_1 + y_2) \geq 0$.

Now, let $d_1 = \text{diam}(S_1)$ be the maximum distance between any two points in $S_1$. Then $\partial B_{2d_1}(x_1)$, the set of points distance $2d_1$ from $x_1$, must be disjoint from $S_1$, so $f_1$ does not achieve its minimum on this set. However, since the set is compact, we can still minimize $f_1$ on it. Let $x_1^* = \arg\min_{x \in \partial B_{2d_1}(x_1)} f_1(x)$, and $y_1^* = f_1(x_1^*) > y_1$. By convexity, the segment between $(x_1, y_1)$ and any other point in the epigraph of $f_1$ must be entirely contained within the epigraph. In particular, for any $x$ outside the ball of radius $2d_1$, the line connecting $(x_1, y_1)$ and $(x, f(x))$ must pass through or above $(x', y_1^*)$ for some $x' \in \partial B_{2d_1}(x_1)$. Essentially, this tells us that outside the $2d_1$-ball the epigraph of $f_1$ lies above the cone of slope $\frac{y_1^* - y_1}{2d_1}$ centered at $x_1$. Algebraically, this means that for any $x \notin B_{2d_1}(x_1)$, $f_1(x) \geq \frac{y_1^* - y_1}{2d_1}\|x - x_1\| + y_1$. In particular, if we let $r_1 = \max(2d_1, \frac{\delta 2d_1}{y_1^* - y_1})$, then for any $x \notin \overline{B}_{r_1}(x_1)$,

$$f_1(x) > \frac{y_1^* - y_1}{2d_1}r_1 + y_1 \geq y_1 + \delta \ .$$

We can repeat the same process for $x_2$ and $f_2$, letting $d_2 = \text{diam}(S_2)$, $x_2^* = \arg\min_{x \in \partial B_{2d_2}(x_2)} f_2(x)$, and $y_2 = f_2(x_2^*)$, and $r_2 = \max(2d_2, \frac{\delta 2d_2}{y_2^* - y_2})$, we have for any $x \notin \overline{B}_{r_2}(x_2)$,

$$f_2(x) > y_2 + \delta \ .$$

Let $B = \overline{B}_{r_1}(x_1) \cup \overline{B}_{r_2}(x_2)$. Combining the previous two equations, we have that for any $x \notin B$,

$$g(x) = f_1(x) + f_2(x) > (y_1 + \delta) + (y_2 + \delta) \geq y \ .$$

Recall that $y = g(x_g)$ was chosen arbitrarily. In particular, we must have $\inf_{x \in \mathbb{R}^d} g(x) \leq y$. Therefore, $g$ can achieve its minimum only on $B$, so we can equivalently define $S = \arg\min_{x \in B} g(x)$. Finally, since $B$ is bounded, $S$ must be as well, and since it is closed the argmin of $g$ must be achieved somewhere, so $S$ is nonempty.

$\square$

**Theorem 8.** *Let $L : \mathbb{R}^d \to \mathbb{R}^n$. If for each $y \in [n]$, $L(\cdot)_y : \mathbb{R}^d \to \mathbb{R}$ is convex, and $\arg\min_{u \in \mathbb{R}^d} L(u)_y$ is non-empty and compact, then $\Gamma(p)$ is non-empty and compact for each $p \in \Delta_n$.*

*Proof.* Pick any $p \in \Delta_n$. Then, for each $y \in [n]$, $p_y \cdot L(\cdot)_y$ is convex. Further, notice that since $p_y \cdot L(\cdot)_y$ is just $L(\cdot)_y$ scaled by a positive scalar, it follows that $\arg\min_{u \in \mathbb{R}^d} p_y \cdot L(\cdot)_y = \arg\min_{u \in \mathbb{R}^d} L(\cdot)_y$, and thus, $\arg\min_{u \in \mathbb{R}^d} p_y \cdot L(\cdot)_y$ is non-empty and compact for each $y \in [n]$. Applying Lemma 18 inductively, it follows that $\arg\min_{u \in \mathbb{R}^d} \langle p, L(\cdot) \rangle$ is non-empty and compact. Since we picked $p$ arbitrarily, it follows that $\Gamma(p)$ is non-empty and compact for each $p \in \Delta_n$. $\square$

# D    One-Dimensional Surrogate Losses

**Definition 7. (Orderable)** *[Lambert, 2011] A finite property $\gamma : \Delta_n \rightrightarrows \mathcal{R}$ is orderable, if there is an enumeration of $\mathcal{R} = \{r_1, r_2, ..., r_k\}$ such that for all $i \leq k - 1$, we have $\gamma_{r_j} \cap \gamma_{r_{j+1}}$ is a hyperplane intersected with $\Delta_n$. We say that the ordered tuple $E_\gamma := (r_1, r_2, ..., r_k)$ is the enumeration associated with $\mathcal{R}$.*

Without loss of generality, we assume for the rest of this section that for any finite orderable property $\gamma$, it holds that, $\gamma_j \cap \gamma_{j+1}$ is a hyperplane intersected with $\Delta_n, \forall j \in [k - 1]$. In particular, the enumeration associated with $\mathcal{R}$ will always assumed to be $E_\gamma = (1, 2, ..., k - 1, k)$.

**Theorem 9.** *[Finocchiaro et al., 2020] If a convex surrogate loss $L : \mathbb{R} \to \mathbb{R}^n$ indirectly elicits a target loss $\ell : \mathcal{R} \to \mathbb{R}^n$, then the property $\gamma = \mathrm{prop}[\ell]$ is orderable.*

**Definition 8. (Intersection Graph)** *[Finocchiaro et al., 2020] Given a discrete loss $\ell : \mathcal{R} \to \mathbb{R}^n$ and associated finite property $\gamma = prop[\ell]$, the* intersection graph *has vertices $\mathcal{R}$ and edges $(r, r')$ if $\gamma_r \cap \gamma_{r'} \cap relint(\Delta_n) \neq \emptyset$.*

**Lemma 19.** *[Finocchiaro et al., 2020] A finite property $\gamma$ is orderable if and only if its intersection graph is a path, i.e., a connected graph where two nodes have degree $1$ and all other nodes have degree $2$.*

**Lemma 20.** *Let $L : \mathbb{R} \to \mathbb{R}^n$ be a convex, differentiable surrogate and suppose $\ell : \mathcal{R} \to \mathbb{R}^n$. If $L$ indirectly elicits $\ell$, then there exist disjoint sets $I^1, I^2, ..., I^{k-1} \subset \mathbb{R}^d$, where for each $j \in [k-1]$, $I^j := \{u^* \in \mathbb{R} | \Gamma_{u^*} = \gamma_j \cap \gamma_{j+1}\}$. For each $j \in [k-1]$, the set $I^j$ is either a singleton $\{u^j\}$ or a closed compact interval $[u^{j,1}, u^{j,2}]$.*

*Proof.* Since $L$ indirectly elicits $\ell$, it follows by Theorem 9 that $\gamma$ is orderable. Thus, for each $j \in [k-1]$, $\gamma_j \cap \gamma_{j+1}$ is a hyperplane intersected with $\Delta_n$. Denote $T_j := \gamma_j \cap \gamma_{j+1}$. By the non-redundancy of target reports, it holds for any $j \in [k-1]$ that $\mathrm{affdim}(\gamma_j) = n-1$, $\mathrm{affdim}(T_j) = n-2$ and that $\mathrm{relint}(T_j) \subset \mathrm{relint}(\Delta_n)$. Fix $j \in [k-1]$. Suppose $p \in \mathrm{relint}(T_j) \implies p \in \mathrm{relint}(\Delta_n)$. By minimizability of $\langle p, L(\cdot) \rangle$, there exists a $u^j \in \mathbb{R}$, such that $p \in \Gamma_{u^j}$. Since $L$ indirectly elicits $\ell$, it follows from Lemma 4 that $\mathrm{rank}(\nabla L(u^j)) = 1$. Further, since $p \in \Gamma_{u^j} \cap \mathrm{relint}(\Delta_n)$, it follows that $\mathrm{affdim}(\Gamma_{u^j}) = n-2$. Now, we claim that $\Gamma_{u^j} = T_j$. We first show that $\Gamma_{u^j} \subseteq T_j$. Assume to the contrary. Then, $\exists p' \in \Gamma_{u^j}$, such that $p' \notin T_j$. Since $\Gamma_{u^j}$ is convex, it follows that $\mathrm{conv}(p', p) \subseteq \Gamma_{u^j}$. Since $\gamma$ is orderable, we know from Lemma 19 that no 3 target cells intersect in $\mathrm{relint}(\Delta_n)$. Then, since $p$ lies on the interior of the common boundary between $(n-2$ dimensional) polytopes $\gamma_j$ and $\gamma_{j+1}$, there must be a sufficiently small $\epsilon > 0$, such that $B_\epsilon(p) \cap \Delta_n$ is fully contained in $\mathrm{relint}(\gamma_j)$ in one halfspace defined by the hyperplane $T_j$ and is fully contained in $\mathrm{relint}(\gamma_{j+1})$ in the other halfspace defined by $T_j$. It follows that $\exists q \in \mathrm{conv}(p', p)$, such that $q \in \mathrm{relint}(\gamma_j)$ or $q \in \mathrm{relint}(\gamma_{j+1})$. Suppose WLOG that $q \in \mathrm{relint}(\gamma_j)$. This means that $\exists q \in \Gamma_{u^j} : \gamma(q) = \{j\}$. So, by indirect elicitation, it must hold that $\Gamma_{u^j} \subseteq \gamma_j$ and that $\Gamma_{u^j} \nsubseteq \gamma_r$ for any $r \neq j$. However, since $\Gamma_{u^j}$ is the intersection of the subspace $\ker(\nabla L(u^j)^\top)$ with $\Delta_n$, $\Gamma_{u^j}$ must have extremal points at the boundaries of $\Delta_n$. This means that $\Gamma_{u^j}$ cannot terminate at $p$ and must extend beyond $p$ into $\gamma_{j+1}$. Thus, $\exists q' \in \mathrm{relint}(\gamma_{j+1})$, such that $q' \in \Gamma_{u^j}$. This violates $\Gamma_{u^j} \subseteq \gamma_j$ and hence violates indirect elicitation. So, $\Gamma_{u^j} \subseteq T_j$.

For the reverse inclusion, assume to the contrary that $\Gamma_{u^j} \subset T_j$. This means that $\Gamma_{u^j}$ must have an extremal point $p \in \mathrm{relint}(T_j)$, which in turn means that $p^* \in \mathrm{relint}(\Delta_n)$. However, $\Gamma_{u^j}$ is the intersection of a subspace with $\Delta_n$, so its extremal points must be on the boundary of $\Delta_n$. Thus, $\Gamma_{u^j} = T_j$.

Now, define $I^j := \{u^* \in \mathbb{R} | \Gamma_{u^*} = \gamma_j \cap \gamma_{j+1}\}$. It follows that $u^* \in I^j \implies u^* \in \Gamma(p)$ since $p \in \gamma_j \cap \gamma_{j+1}$. Thus, $I_j \subseteq \Gamma(p)$. However, choosing $u^j \in \Gamma(p)$ arbitrarily, we proved that $\Gamma_{u^j} = \gamma_j \cap \gamma_{j+1}$. Thus, $\Gamma(p) \subseteq I^j$. Therefore, $I_j = \Gamma(p)$ which always exists by minimizability of $\langle p, L(\cdot) \rangle$. Further, since $\Gamma(p)$ is compact (by assumption), it follows that $I^j$ is either a singleton, or a compact interval in $\mathbb{R}$. Similarly, by picking $j' \in [k-1] : j' \neq j$, we can establish the existence of (a singleton or compact interval set) $I^{j'}$, such that $\Gamma_{u^{j'}} = \gamma_{j'} \cap \gamma_{j'+1}$. To see that, $I^j \cap I^{j'} = \emptyset, \forall j, j' \in [k-1] : j \neq j'$, assume to the contrary that there exists some $v \in I^j \cap I^{j'}$. Then, $\Gamma_v = \gamma_j \cap \gamma_{j+1} = \gamma_{j'} \cap \gamma_{j'+1}$. However, $\gamma_j \cap \gamma_{j+1} \neq \gamma_{j'} \cap \gamma_{j'+1}$ for any $j \neq j'$ and so the sets $I_j$ and $I'_j$ must be disjoint. $\square$

Throughout the rest of this section, we will inherit from Lemma 20, the notation $I^j$ for the set $\{u^* \in \mathbb{R} | \Gamma_{u^*} = \gamma_j \cap \gamma_{j+1}\}$

**Lemma 21.** *Let $L : \mathbb{R} \to \mathbb{R}^n$ be a convex, differentiable surrogate and suppose $\ell : \mathcal{R} \to \mathbb{R}^n$. Suppose $L$ indirectly elicits $\ell$. Let $p, q \in \Delta_n : \gamma(p) = \{j\}$ and $\gamma(q) = \{j+1\}$, where $j \in [k-1]$. Let $u_p \in \Gamma(p)$ and $u_q \in \Gamma(q)$. Then, $u_p < \min(I^j) \leq \max(I^j) < u_q$, or $u_q < \min(I^j) \leq \max(I^j) < u_p$.*

*Proof.* Fix $j \in [k-1]$. Pick any $u^j \in I^j$. For $p : \gamma(p) = \{j\}, q : \gamma(q) = \{j+1\}$, and $u_p \in \Gamma(p), u_q \in \Gamma(q)$, we will show that exactly one of the following holds: $u_p < u^j < u_q$ or

$u_q < u^j < u_p$. Now, from the definition of $I^j$ and from Lemma 3, we know that $\Gamma_{u^j} = \gamma_j \cap \gamma_{j+1} = \{p' \subseteq \Delta_n | \nabla L(u^j)^\top p' = 0\}$. Since $p \in \text{relint}(\gamma_j)$ and $q \in \text{relint}(\gamma_{j+1})$, it follows that $p$ and $q$ are on different sides of the hyperplane $\{x \in \mathbb{R}^n | \nabla L(u^j)^\top x = 0\}$. So, it must hold that $\nabla L(u^j)^\top p < 0$ and $\nabla L(u^j)^\top q > 0$, or that $\nabla L(u^j)^\top p > 0$ and $\nabla L(u^j)^\top q < 0$.

Assume WLOG that $\nabla L(u^j)^\top p < 0$ and $\nabla L(u^j)^\top q > 0$. Now, notice that $\nabla \langle p, L(\cdot) \rangle = \nabla L(\cdot)^\top p$ and similarly, $\nabla \langle q, L(\cdot) \rangle = \nabla L(\cdot)^\top q$. Since the function $\langle p, L(\cdot) \rangle$ is convex and differentiable, it holds from the monotonicity of gradients of convex functions that, $\langle \nabla L(u^j)^\top p - \nabla L(u_p)^\top p, u^j - u_p \rangle \geq 0 \implies \langle \nabla L(u^j)^\top p, u^j - u_p \rangle \geq 0 \implies u^j - u_p \leq 0$, since $\nabla L(u^j)^\top p < 0$. Clearly, $u^j \neq u_p$ and so, $u^j - u_p < 0$. Similarly, we can show that $u^j - u_q > 0$. We have thus shown that $u_q < u^j < u_p$. Since $u^j$ was arbitrarily chosen from $I^j$, it holds that $u_q < \min(I^j) \leq \max(I^j) < u_p$. Now, for $j \in \mathcal{R}$, denote $U^j := \{u \in \mathbb{R} : u \in \Gamma(p), \gamma(p) = \{j\}\}$.

If instead $\nabla L(u^j)^\top p > 0$ and $\nabla L(u^j)^\top q < 0$, it would follow that $u_p < \min(I^j) \leq \max(I^j) < u_q$ by the same argument. $\qquad\square$

Denote by $U^j := \{u \in \mathbb{R} | u \in \Gamma(p), \gamma(p) = \{j\}\}$. Let $u^1 \in U^1, u^2 \in U^2, ..., u^k \in U^k$. Then by Lemma 21, we know that, $u^1 < \min(I^1) \leq \max(I^1) < u^2 < \min(I^2) \leq \max(I^2) < u^3... < u^{k-2} < \min(I^{k-2}) \leq \max(I^{k-2}) < u^{k-1} < \min(I^{k-1}) \leq \max(I^{k-1}) < u^k$, or, $u^1 > \max(I^1) \geq \min(I^1) > u^2 > \max(I^2) \geq \min(I^2) > u^3... > u^{k-2} > \max(I^{k-2}) \geq \min(I^{k-2}) > u^{k-1} > \max(I^{k-1}) \geq \min(I^{k-1}) > u^k$. This means, that either $\max(I^1) < \min(I^{k-1})$, or $\max(I^{k-1}) < \min(I^1)$. We assume WLOG from hereon, that $\max(I^1) < \min(I^{k-1})$. Thus, we have shown that $\min(I^j)$ is a uniform, strict upper bound on $U^j$ and that $\max(I^{j-1})$ is uniform, strict lower bound on $U^j$.

**Theorem 10.** *Let $L : \mathbb{R} \to \mathbb{R}^n$ be a convex, differentiable surrogate and suppose $\ell : \mathbb{R} \to \mathbb{R}^n$. Under Assumption 1, $L$ is calibrated with respect to $\ell$. Then $(L, \psi)$ is calibrated with respect to $\ell$, for any link $\psi : \mathbb{R} \to \mathcal{R}$, that satisfies*

$$\psi(u) \in \begin{cases} \{j, j+1\} & \text{if } u \in I^j \text{ for any } j \in [k-1] \\ \{1\} & \text{if } u < min(I^1) \\ \{j\} & \text{if } u \in (\max(I^{j-1}), min(I^j)), \ j \in 2, \dots, k-1 \\ \{k\} & \text{if } u > max(I^{k-1}) \end{cases}. \tag{2}$$

*Proof.* For the entirety of this proof, we define $\gamma_0 = \gamma_{k+1} = I^0 = I^{k+1} = \emptyset$. Let $\psi : \mathbb{R} \to \mathcal{R}$ be any link of the form proposed in the theorem statement. We first show calibration for $p \in \Delta_n : \gamma(p) = \{j\}$ for some $j \in \mathcal{R}$. We know from Lemma 21, that $\max(I^{j-1}) < u_p < \min(I^j)$ for any $u_p \in \Gamma(p)$. We have that $\psi(u) \neq j$ for any $u < \min(I^{j-1})$ and any $u > \max(I^j)$. Whereas, $u \in I^{j-1}$ can be linked to either $j-1$ or $j$, and $u \in I^j$ can be linked to either $j$ or $j+1$. The remaining reports always link to $j$.

$$\inf_{u \in \mathbb{R} : \psi(u) \notin \gamma(p)} \langle p, L(u) \rangle = \inf_{u \in \mathbb{R} : \psi(u) \neq j} \langle p, L(u) \rangle$$
$$\geq \inf_{u \in \mathbb{R} : u \leq \max(I^{j-1}) \text{ or } u \geq \min(I^j)} \langle p, L(u) \rangle$$
$$= \min\{\inf_{u \in \mathbb{R} : u \leq \max(I^{j-1})} \langle p, L(u) \rangle, \inf_{u \in \mathbb{R} : u \geq \min(I^j)} \langle p, L(u) \rangle\}$$
$$= \min\{\langle p, L(\max(I^{j-1})) \rangle, \langle p, L(\min(I^j)) \rangle\} > \inf_{u \in \mathbb{R}} \langle p, L(u) \rangle$$

The final equality follows from convexity of the function $\langle p, L(\cdot) \rangle$. The final strict inequality follows from the fact that $\max(I^{j-1}) < u_p < \min(I^j)$ for any $u_p \in \Gamma(p)$. The same argument holds for any other distribution $p' : \gamma(p') = \{j\}$. In fact, since $j$ was picked arbitrarily from $\mathcal{R}$, the argument extends to any $q : \gamma(q) = \{j'\}$, for any $j' \in \mathcal{R}$. So, we have established calibration at all distribution lying in the relative interiors of target cells. We still need to prove calibration for distributions lying on target boundaries.

Again, start by fixing some $j \in [k-1]$. Suppose $p \in \text{relint}(\gamma_j \cap \gamma_{j+1})$. Then, since $\gamma$ is orderable, we know from Lemma 19 that no 3 target cells can intersect in the relative interior of the simplex. Thus, $\gamma(p) = \{j, j+1\}$. We have that $\psi(u) \notin \{j, j+1\}$ for any $u < \min(I^{j-1})$ and any $u > \max(I^{j+1})$. Whereas, $u \in I^{j-1}$ can be linked to either $j-1$ or $j$ and $u \in \cup I^{j+1}$ can be linked to either $j$ or

$j + 1$. The remaining reports always link to one of $j - 1$ or $j$.

$$\inf_{u \in \mathbb{R}: \psi(u) \notin \gamma(p)} \langle p, L(u) \rangle = \inf_{u \in \mathbb{R}: \psi(u) \notin \{j, j+1\}} \langle p, L(u) \rangle$$

$$\geq \inf_{u \in \mathbb{R}: u \leq \max(I^{j-1}) \text{ or } u \geq \min(I^{j+1})} \langle p, L(u) \rangle$$

$$= \min\{\inf_{u \in \mathbb{R}: u \leq \max(I^{j-1})} \langle p, L(u) \rangle, \inf_{u \in \mathbb{R}: u \geq \min(I^{j+1})} \langle p, L(u) \rangle\}$$

$$= \min\{\langle p, L(\max(I^{j-1})) \rangle, \langle p, L(\min(I^{j+1})) \rangle\} > \inf_{u \in \mathbb{R}} \langle p, L(u) \rangle$$

The final equality follows from convexity of the function $\langle p, L(\cdot) \rangle$. The final strict inequality follows by noting that $\gamma_{j-1} \cap \gamma_j$ and $\gamma_{j+1} \cap \gamma_{j+2}$ are disjoint from $\text{relint}(\gamma_j \cap \gamma_{j+1})$ and hence $\max(I^{j-1})$ and $\min(I^{j+1})$ are suboptimal for $p \in \text{relint}(\gamma_j \cap \gamma_{j+1})$. The same argument extends to all distributions lying in the relative interiors of target boundaries. Thus, we have established calibration for all distributions, barring distributions on target boundaries that are not inside the relative interiors of the boundaries.

Suppose $p \in \Delta_n$, such that $p$ lies on some target boundary, however, $p$ does not lie in the relative interior of the target boundary. Such points can lie within the intersection of 2 or more target cells. In case, $p \in \cap_{j \in \mathcal{R}} \gamma_j$, $\gamma(p) = \mathcal{R}$ and calibration follows trivially, since the set $u \in \mathbb{R} : \psi(u) \notin \gamma(p)$ is empty. Otherwise, suppose, $\gamma(p) = S$, for some $S \subset R$. It must be that discrete reports within $S$ are consecutive integers. That is, suppose $j \in S$ and $j' \in S$. Then, either $|j - j'| \leq 1$, or else, if $|j - j'| > 1$, then $j^* \in S$ for any $j^* : \min\{j, j'\} < j^* < \max\{j, j'\}$. This follows from the fact that $\gamma$ is orderable, and we are assuming the enumeration associated with $\mathcal{R}$ is $E_\gamma = (1, 2, ..., k - 1, k)$. Thus, suppose that $\gamma(p) = S \subset R : S = \{j, j + 1, ..., j + t\}$, where $t \geq 1$. We have that $\psi(u) \notin S$ for any $u < \min(I^{j-1})$ and for any $u > \max(I^{j+t})$. Whereas, $u \in I^{j-1}$ may be linked to either $j - 1 \notin S$ or $j \in S$, and $u \in I^{j+t}$ may be linked to either $j + t \in S$ or $j + t + 1 \notin S$. The remaining surrogate reports always link to some discrete report in $S$.

$$\inf_{u \in \mathbb{R}: \psi(u) \notin \gamma(p)} \langle p, L(u) \rangle = \inf_{u \in \mathbb{R}: \psi(u) \notin S} \langle p, L(u) \rangle$$

$$\geq \inf_{u \in \mathbb{R}: u \leq \max(I^{j-1}) \text{ or } u \geq \min(I^{j+t})} \langle p, L(u) \rangle$$

$$= \min\{\inf_{u \in \mathbb{R}: u \leq \max(I^{j-1})} \langle p, L(u) \rangle, \inf_{u \in \mathbb{R}: u \geq \min(I^{j+t})} \langle p, L(u) \rangle\}$$

$$= \min\{\langle p, L(\max(I^{j-1})) \rangle, \langle p, L(\min(I^{j+t})) \rangle\} > \inf_{u \in \mathbb{R}} \langle p, L(u) \rangle$$

The final equality follows from convexity of the function $\langle p, L(\cdot) \rangle$. The final strict inequality follows by noting that $p \notin \gamma_{j-1} \cap \gamma_j$ and $p \notin \gamma_{j+t} \cap \gamma_{j+t+1}$ by construction. Thus, $(L, \psi)$ is calibrated at $p$. The same argument extends to any distribution that lies on some target boundary, but not its relative interior. $\square$

## E    Correspondences

We consolidate basic definitions and results about correspondences in this section. Note that different authors can have slightly differing terminology and conventions related to correspondences. We direct the reader to Border [2013] and references therein for a more detailed discussion on different conventions. We adopt the conventions, definitions and terminology used in Border [2013].

**Definition 9.  (Correspondence, graph, image, domain)** *[Border, 2013] A* correspondence *$\varphi$ from $X$ to $Y$ associates to each point in $X$ a subset $\varphi(x)$ of $Y$. We write this as $\varphi : X \rightrightarrows Y$. For a correspondence $\varphi : X \rightrightarrows Y$, let $\text{gr}\,\varphi$ denote the* graph *of $\varphi$, which we define to be*

$$\text{gr}\,\varphi = \{(x, y) \in X \times Y : y \in \varphi(x)\}.$$

*Let $\varphi : X \rightrightarrows Y$, and let $F \subset X$. The* image *$\varphi(F)$ of $F$ under $\varphi$ is defined to be*

$$\varphi(F) = \bigcup_{x \in F} \varphi(x).$$

*The value $\varphi(x)$ is allowed to be the empty set, but we call $\{x \in X : \varphi(x) \neq \emptyset\}$ the* domain *of $\varphi$, denoted $\text{dom}\,\varphi$.*

*The terms* multifunction, point-to-set mapping, *and* set-valued function *are also used for a correspondence.*

**Definition 10. (Metric upper hemicontinuity)** *[Border, 2013] Let $X$ be a metric space equipped with the metric $d_X : X \times X \to \mathbb{R}$. A correspondence $\varphi : X \rightrightarrows Y$ is said to satisfy* metric upper hemicontinuity *at a point $x \in X$ if for every $\epsilon > 0$, $\exists \delta > 0$, such that*

$$d(x, z) < \delta \implies \varphi(z) \subseteq B_\epsilon(\varphi(x))$$

Metric upper hemicontinuity is a special case of the more general, topological notion of upper hemicontinuity which requires that the pre-image of open neighborhoods of $\varphi(x)$ be open sets (see [Border, 2013] for a formal definition). However, metric upper hemicontinuity at $x$ and the topological notion of upper hemicontinuity at $x$ are equivalent when $\varphi(x)$ is compact (see Proposition 11 of [Border, 2013]). For our purposes, this will always be the case. So, we simply work with the simpler, metric based notion of upper hemicontinuity.

**Definition 11. (Closed at $x$, Closed correspondence)***[Border, 2013] The correspondence $\varphi : X \rightrightarrows Y$ is* closed at $x \in X$ *if whenever $x_n \to x$, $y_n \in \varphi(x_n)$, and $y_n \to y$, then $y \in \varphi(x)$. A correspondence is* closed *if it is closed at every point of its domain, that is, if its graph is closed.*

**Lemma 22.** *[Border, 2013] If the correspondence $\varphi : X \rightrightarrows Y$ is closed at $x \in X$, then $\varphi(x)$ is a closed set.*

**Lemma 23.** *[Border, 2013] Suppose $Y$ is compact and $\varphi : X \rightrightarrows Y$ is closed at $x \in X$, then $\varphi$ is upper hemicontinuous at $x$.*

We say a correspondence $\varphi : X \rightrightarrows Y$ is compact-valued if $\varphi(x)$ is compact for every $x \in X$.

**Lemma 24.** *[Border, 2013] Let $K \subset X$ be a compact set and suppose $\varphi : X \rightrightarrows Y$ is upper hemicontinuous and compact-valued. Then $\varphi(K)$ is compact.*

# F   Sufficiency of Strong Indirect Elicitation

We start by presenting a couple of helper results, that we will leverage in our main proofs.

**Lemma 25. Lebesgue's Number Lemma**: *(see Thm. IV.5.4 of Hu [1966]) Let $(\mathcal{X}, d)$ be a compact metric space. Let $\mathcal{A}$ be an arbitrary index set, and $\mathcal{U} = \cup_{\alpha \in \mathcal{A}} \mathcal{U}_\alpha$ be an open cover of $\mathcal{X}$. Then, there exists a number $\delta_{\mathcal{L}} > 0$, such that for any $x \in \mathcal{X}$, $B_{\delta_{\mathcal{L}}}(x) \subset U_{\alpha_x}$, where $\alpha_x \in \mathcal{A}$*

Given an open cover $\mathcal{U}$ compact set $\mathcal{X}$, a constant $\delta_{\mathcal{L}} > 0$ that satisfies the condition of Lemma 25 is known as a *Lebesgue Number* for the cover [Hu, 1966]. We state and prove a simply corollary of Lemma 25, which we will make use of later.

**Corollary 1.** *Let $m \in \mathbb{Z}_+$, $\delta_{\mathcal{L}} > 0$ be a Lebesgue number for some open cover $\mathcal{U} = \cup_{\alpha \in \mathcal{A}} \mathcal{U}_\alpha$ of a compact set $\mathcal{X} \subseteq \mathbb{R}^m$. Then, $B_{\delta_{\mathcal{L}}/2}(\mathcal{X}) \subseteq \mathcal{U}$*

*Proof.* Let $v \in B_{\delta_{\mathcal{L}}/2}(\mathcal{X})$. This means, there exist $u \in \mathcal{X}$ and $b \in B_{\delta_{\mathcal{L}}/2}(\mathbf{0_m})$, such that, $v = u + b$, i.e., $v \in B_{\delta_{\mathcal{L}}/2}(u) \subset B_{\delta_{\mathcal{L}}}(u)$. Now, by Lemma 25, $\exists \alpha \in \mathcal{A} : B_{\delta_{\mathcal{L}}}(u) \subset \mathcal{U}_\alpha \subseteq \mathcal{U}$. So, $v \in \mathcal{U}$. Since $v$ was arbitrarily chosen from $B_{\delta_{\mathcal{L}}/2}(\mathcal{X})$, it follows that $B_{\delta_{\mathcal{L}}/2}(\mathcal{X}) \subseteq \mathcal{U}$ $\qquad \square$

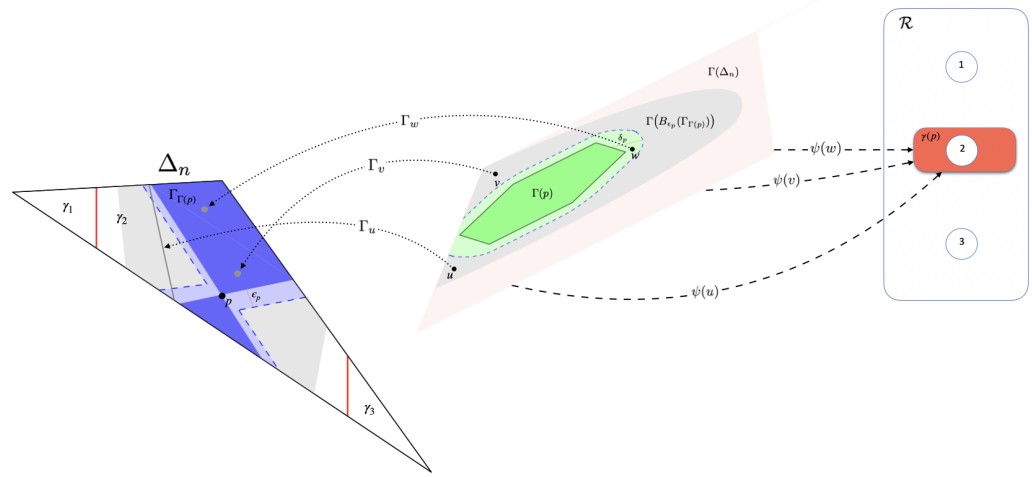

Figure 7: Visual intuition for the proof of sufficiency of strong IE. Let $n = 3, \mathcal{Y} = \{1, 2, 3\}$ and $\mathcal{R} = \{1, 2, 3\}$ (*white block with blue border - right*). Let $p \in \Delta_n$ (*black point within triangle - left*). $\Gamma_{\Gamma(p)}$ is the level-set bundle at $p$ (*dark purple region - left*). Here, $\gamma(p) = \{2\}$. Lemma 29 ensures the existence of $\epsilon_p > 0$: The image of $\Gamma(B_{\epsilon_p}(\Gamma_{\Gamma(p)}))$ (*gray region - center*) under $\Gamma_{(\cdot)}$ is fully contained within $\gamma_2$ (*gray region - left*). By Lemma 27, $\exists \delta_p > 0 : \Gamma(\Delta_n) \cap B_{\delta_p}(\Gamma(p))$ (*light green region - center*) is contained within $\Gamma(B_{\epsilon_p}(\Gamma_{\Gamma(p)}))$ (gray region - center). Thus, every report within the light green region necessarily links to $\gamma(p)$ (*red block - right*). So any sequence of reports $\{u_t\}_{t \in \mathbb{N}_+} \subseteq \Gamma(\Delta_n)$ (*light pink region - center*), for which $\lim_{t \to \infty} u_t \to \Gamma(p)$, must eventually link to $\gamma(p)$. Further, Theorem 11 shows how to link nowhere optimal reports without violating calibration. Thus, $\Gamma(p)$ is 'protected' in the calibration sense.

Suppose $L : \mathbb{R}^d \to \mathbb{R}^n$ is a convex, differentiable surrogate loss. We assume throughout that for every $p \in \Delta_n, \Gamma(p) := \arg\min_{u \in \mathbb{R}^d} \langle p, L(u) \rangle$ exists and that $\Gamma(p)$ is compact. Note that, since the function $\langle p, L(\cdot) \rangle$ is convex, $\Gamma(p)$ is closed by Lemma 14. Thus, it suffices to say $\Gamma(p)$ is bounded, though we will usually say compact for clarity.

To prove the sufficiency of strong indirect elicitation, we first establish basic properties about the surrogate level sets $\{\Gamma_u | u \in \mathbb{R}^d\}$. To do so, we adjust our lens slightly, and view $\Gamma_u$ as the image of a correspondence at a point $u$ in its domain. In particular, we denote $\Gamma_{(\cdot)} : \mathbb{R}^d \rightrightarrows \Delta_n$ as the correspondence that maps surrogate reports to the set of distributions they optimize.

**Lemma 26.** *The correspondence $\Gamma_{(\cdot)} : \mathbb{R}^d \rightrightarrows \Delta_n$ is closed at every $u \in \Gamma(\Delta_n)$. In fact, $\Gamma_{(\cdot)}$ is upper hemicontinuous at every $u \in \Gamma(\Delta_n)$.*

*Proof.* Suppose $u \in \Gamma(\Delta_n) \implies \Gamma_u \neq \emptyset$. Let $\{u_i\}_{i \in \mathbb{N}}$ be a sequence in $\mathbb{R}^d$ and let $\{p_i\}_{i \in \mathbb{N}}$ be a sequence in $\Delta_n$ such that $p_i \in \Gamma_{u_i} \forall i \in \mathbb{N}$ and suppose $p_i \to p \in \Delta_n$ and $u_i \to u \in \mathbb{R}^d$. To prove $\Gamma_{(\cdot)}$ is closed, we need to show $p \in \Gamma_u$ (see Definition 11). Let $i \in \mathbb{N}$. Since $p_i \in \Gamma_{u_i}$, $\nabla L(u_i)^T p_i = \mathbf{0_d}$. Now, notice that:

$$\|\nabla L(u_i)^T p_i - \nabla L(u)^T p\| \leq \|\nabla L(u_i)^T p_i - \nabla L(u_i)^T p\| + \|\nabla L(u_i)^T p - \nabla L(u)^T p\|$$
$$\leq \|\nabla L(u_i)\| \cdot \|p_i - p\| + \|\nabla L(u_i) - \nabla L(u)\| \cdot \|p\|$$

Observe that $\lim_{i \to \infty} p_i = p$ (by construction), and $\lim_{i \to \infty} \nabla L(u_i) = \nabla L(u)$ (since $u_i \to u$ by construction, and since $\nabla L(\cdot)$ is continuous). Thus,

$$\lim_{i \to \infty} \|\nabla L(u_i)\| \cdot \|p_i - p\| + \|\nabla L(u_i) - \nabla L(u)\| \cdot \|p\| = 0$$
$$\implies \lim_{i \to \infty} \nabla L(u_i)^T p_i = \nabla L(u)^T p = \mathbf{0_d} \implies p \in \Gamma_u$$

We have thus shown that $\Gamma_{(\cdot)}$ is closed at $u$. Since the target space, $\Delta_n$ is compact, the fact that $\Gamma_{(\cdot)}$ is closed at $u$ implies that $\Gamma_{(\cdot)}$ is upper hemicontinuous at $u$ (by Lemma 23). $\square$

**Lemma 27.** *Let $p \in \Delta_n$. For every $\epsilon > 0$, there exists $\delta > 0$, such that:*

$$\Gamma(\Delta_n) \cap B_\delta(\Gamma(p)) \subseteq \Gamma(B_\epsilon(\Gamma_{\Gamma(p)}))$$

*Proof.* Pick any $p \in \Delta_n$. Since $\Gamma_{(\cdot)}$ is upper hemicontinuous at $u \in \Gamma(p)$ by Lemma 26, we have that for every $\epsilon > 0$, $\exists \delta_u > 0$ for each $u \in \Gamma(p)$ such that:

$$
\begin{aligned}
&\Gamma_{u'} \subseteq B_\epsilon(\Gamma_u), \forall u' \in B_{\delta_u}(u) \\
\implies &\cup_{u' \in B_{\delta_u}} \Gamma_{u'} = \Gamma_{B_{\delta_u}(u)} \subseteq B_\epsilon(\Gamma_u) \\
\implies &\cup_{u \in \Gamma(p)} \Gamma_{B_{\delta_u}(u)} \subseteq \cup_{u \in \Gamma(p)} B_\epsilon(\Gamma_u) = B_\epsilon(\cup_{u \in \Gamma(p)} \Gamma_u) = B_\epsilon(\Gamma_{\Gamma(p)})
\end{aligned}
$$

So, we have shown that $\cup_{u \in \Gamma(p)} \Gamma_{B_{\delta_u}(u)} \subseteq B_\epsilon(\Gamma_{\Gamma(p)})$. It follows by Lemma 2 that:

$$\Gamma(\cup_{u \in \Gamma(p)} \Gamma_{B_{\delta_u}(u)}) \subseteq \Gamma(B_\epsilon(\Gamma_{\Gamma(p)})) \tag{3}$$

Now, suppose $u' \in \Gamma(\Delta_n) \cap B_{\delta_u}(u)$. Let $p' \in \Gamma_{u'}$. Then, $p' \in \Gamma_{u'} \subseteq \Gamma_{B_{\delta_u}(u)}$. So $u' \in \Gamma(p') \subseteq \Gamma(\Gamma_{B_{\delta_u}(u)})$. Since $u'$ was picked arbitrarily in $\Gamma(\Delta_n) \cap B_{\delta_u}(u)$, it follows that:

$$
\begin{aligned}
&\Gamma(\Delta_n) \cap B_{\delta_u}(u) \subseteq \Gamma(\Gamma_{B_{\delta_u}(u)}) \\
\implies &\Gamma(\Delta_n) \cap \cup_{u \in \Gamma(p)} B_{\delta_u}(u) \subseteq \cup_{u \in \Gamma(p)} \Gamma(\Gamma_{B_{\delta_u}(u)}) \subseteq \Gamma(\cup_{u \in \Gamma(p)} \Gamma_{B_{\delta_u}(u)})
\end{aligned}
$$

The final inclusion, i.e., $\cup_{u \in \Gamma(p)} \Gamma(\Gamma_{B_{\delta_u}(u)}) \subseteq \Gamma(\cup_{u \in \Gamma(p)} \Gamma_{B_{\delta_u}(u)})$, holds by the following rationale: Suppose $v \in \cup_{u \in \Gamma(p)} \Gamma(\Gamma_{B_{\delta_u}(u)})$. This means $\exists u' \in \Gamma(p) : v \in \Gamma(\Gamma_{B_{\delta_{u'}}(u')}) \implies \exists p_v \in \Gamma_{B_{\delta_{u'}}(u')} : v \in \Gamma(p_v)$. Now, since $p_v \in \Gamma_{B_{\delta_{u'}}(u')}$, $p_v \in \cup_{u \in \Gamma(p)} \Gamma_{B_{\delta_u}(u)}$ as $u' \in \Gamma(p)$. Thus, $v \in \Gamma(\cup_{u \in \Gamma(p)} \Gamma_{B_{\delta_u}(u)})$. So, we have shown that:

$$\Gamma(\Delta_n) \cap \cup_{u \in \Gamma(p)} B_{\delta_u}(u) \subseteq \Gamma(\cup_{u \in \Gamma(p)} \Gamma_{B_{\delta_u}(u)}) \tag{4}$$

Observing that the RHS of inclusion (4) and the LHS of inclusion (3) are the same, we combine the inclusions to get that:

$$\Gamma(\Delta_n) \cap \cup_{u \in \Gamma(p)} B_{\delta_u}(u) \subseteq \Gamma(B_\epsilon(\Gamma_{\Gamma(p)})) \tag{5}$$

Let us denote $\cup_{u \in \Gamma(p)} B_{\delta_u}(u)$ by $\mathcal{U}$. Now, notice that $\Gamma(p) \subseteq \mathcal{U}$. Further, $\mathcal{U}$ is a union of open sets and is thus open itself. This means, $\mathcal{U}$ is an open cover of $\Gamma(p)$. Since $\Gamma(p)$ is compact, due to Lemma 25, there exists a Lebesgue number $\delta_{\mathcal{L}} > 0$ for $\mathcal{U}$. Set $\delta := \delta_{\mathcal{L}}/2$. Then, by Corollary 1, $B_\delta(\Gamma(p)) \subseteq \mathcal{U}$.

$$
\begin{aligned}
\implies \Gamma(\Delta_n) \cap B_\delta(\Gamma(p)) &\subseteq \Gamma(\Delta_n) \cap \mathcal{U} \\
&= \Gamma(\Delta_n) \cap \left( \cup_{u \in \Gamma(p)} B_{\delta_u}(u) \right) \\
&\subseteq \Gamma(B_\epsilon(\Gamma_{\Gamma(p)}))
\end{aligned}
$$

where the final inclusion follows from inclusion (5), thus concluding our proof. $\square$

**Lemma 28.** *For any $p \in \Delta_n$, suppose $\Gamma(p)$ is a non-empty, compact set. Then, the set $\Gamma_{\Gamma(p)} = \cup_{u \in \Gamma(p)} \Gamma_u$ is closed.*

*Proof.* Fix $p \in \Delta_n$. We know from Lemma 26 that $\Gamma_{(\cdot)}$ is closed at any $u \in \Gamma(p)$. This implies that for any $u \in \Gamma(p)$, $\Gamma_u$ is a closed set. Since the target space $\Delta_n$ is compact, $\Gamma_u$ must be bounded, and thus $\Gamma_u$ is compact for each $u \in \Gamma(p)$. So the restriction of $\Gamma_{(\cdot)}$ to $\Gamma(p)$, i.e., $\Gamma_{(\cdot)|\Gamma(p)} : \Gamma(p) \rightrightarrows \Delta_n$ is upper hemicontinuous and compact-valued, and so by Lemma 24, $\Gamma_{\Gamma(p)}$ is compact since $\Gamma(p)$ is compact by assumption. Thus, $\Gamma_{\Gamma(p)}$ is closed. $\square$

**Lemma 29.** *Let $L : \mathbb{R}^d \to \mathbb{R}^n$ be a convex, differentiable surrogate. Suppose Assumption 1 holds . Let $\ell : \mathcal{R} \to \mathbb{R}^n$ be a discrete target loss, with finite property $\gamma := \text{prop}[L]$. If $L$ strongly indirectly elicits $\ell$, then the following holds: For each $p \in \Delta_n$:*

$$\exists \, \epsilon_p > 0 : \text{ for any } v \in \Gamma(B_{\epsilon_p}(\Gamma_{\Gamma(p)})), \text{ it holds that } \gamma(q) \subseteq \gamma(p), \forall q \in \Gamma_v$$

*Proof.* Let $p \in \Delta_n$. Define $S := \cup_{r' \in \mathcal{R} \setminus \gamma(p)} \gamma_{r'}$. If $S$ is empty, then $\gamma(p) = \mathcal{R}$ and $\gamma(p') \subseteq \gamma(p), \forall p \in \Delta_n$ and then the result follows trivially. Otherwise, notice that $\gamma(q) = \gamma(p)$ for any $q \in \Gamma_{\Gamma(p)}$, since $\exists v \in \Gamma(p) : q \in \Gamma_v$ and then $\gamma(p) = \gamma(q)$ by strong indirect elicitation. Thus, $\gamma(q) \cap S = \emptyset, \forall q \in \Gamma_{\Gamma(p)}$, and so $S \cap \Gamma_{\Gamma(p)} = \emptyset$. Recall that the target cells, i.e., $\gamma_r : r \in \mathcal{R}$ are closed by virtue of being convex polytopes as shown in Lemma 7. So, $S$ is a finite union of closed sets, and is thus closed. In fact $S$ is compact since $S \subseteq \Delta_n$. Also, $\Gamma_{\Gamma(p)}$ is closed by Lemma 28. Similarly, $\Gamma_{\Gamma(p)}$ is compact since $\Gamma_{\Gamma(p)} \subseteq \Delta_n$. Since $S \cap \Gamma_{\Gamma(p)} = \emptyset$, and both $S$ and $\Gamma_{\Gamma(p)}$ are compact and , it holds that $d(S, \Gamma_{\Gamma(p)}) > 0$. Thus, $\exists \epsilon_p > 0 : \forall p' \in B_{\epsilon_p}(\Gamma_{\Gamma(p)}), p' \notin S$ and so:

$$\exists \epsilon_p > 0 : \forall p' \in B_{\epsilon_p}(\Gamma_{\Gamma(p)}), \gamma(p') \subseteq \gamma(p) \tag{6}$$

Now, suppose $v \in \Gamma(B_{\epsilon_p}(\Gamma_{\Gamma(p)}))$. So, there exists $p' \in \Gamma_v$, such that $p' \in B_{\epsilon_p}(\Gamma_{\Gamma(p)}) \implies \gamma(p') \subseteq \gamma(p)$ due to (6). Now, for any $q \in \Gamma_v$, it must hold that $\gamma(q) = \gamma(p')$ by strong indirect elicitation. Therefore, $\gamma(q) \subseteq \gamma(p), \forall q \in \Gamma_v$. The result follows since $v$ was arbitrarily chosen from $\Gamma(B_{\epsilon_p}(\Gamma_{\Gamma(p)}))$. $\qquad \square$

**Lemma 30.** *Let $L : \mathbb{R}^d \to \mathbb{R}^n$ be a convex, differentiable surrogate. Suppose Assumption 1 holds. Then, $\Gamma(\Delta_n)$ is closed.*

*Proof.* We show that for any sequence $\{u_t\}_{t \in \mathbb{N}_+}$ such that $u_t \in \Gamma(\Delta_n)$ for each $t \in \mathbb{N}_+$, if $u_t \to u$, then it holds that $u \in \Gamma(\Delta_n)$. Since $u_t \in \Gamma(\Delta_n)$, $\exists p_t \in \Delta_n$, such that $p_t \in \Gamma_{u_t}$ for each $t \in \mathbb{N}_+$. Now, since $\Delta_n$ is compact, we can extract a subsequence $p_{t_j}$, such that $p_{t_j} \to p \in \Delta_n$ and that $u_{t_j} \to u$. We will show that $u \in \Gamma(p)$. Consider:

$$\|\nabla L(u_{t_j})^\top p_{t_j} - \nabla L(u)^\top p\| \leq \|\nabla L(u_{t_j})^\top p_{t_j} - \nabla L(u)^\top p_{t_j}\| + \|\nabla L(u)^\top p_{t_j} - \nabla L(u)^\top p\|$$
$$\leq 1 \cdot \|\nabla L(u_{t_j}) - L(u)\| + \|\nabla L(u)\| \cdot \|p_{t_j} - p\|$$

Now, as $j \to \infty$, $\|\nabla L(u_{t_j}) - \nabla L(u)\| \to 0$ as $u_{t_j} \to u$ (by the continuity of $\nabla L(\cdot)$). Also, $\|p_{t_j} - p\| \to 0$ by construction. Hence, $\nabla L(u_{t_j})^\top p_{t_j} \to \nabla L(u)^\top p$. However, since $p_{t_j} \in \Gamma_{u_{t_j}}$, it holds that $\nabla L(u_{t_j})^\top p_{t_j} = \mathbf{0}_d \implies \nabla L(u)^\top p = \mathbf{0}_d \implies u \in \Gamma(p) \implies u \in \Gamma(\Delta_n)$, thus concluding our proof. $\qquad \square$

We are now ready to state and prove our main link construction, which in turn proves the sufficiency of strong IE for calibration under our assumptions. Throughout the proof, let $\text{dist}(a, B) := \inf_{b \in B} \|a - b\|_2$, and when the infimum is attained, let $\text{proj}_B(a) := \{b \in B | \|a - b\|_2 = \text{dist}(a, B)\}$, for any point $a$ and any set $B$.

**Theorem 11.** *Let $L : \mathbb{R}^d \to \mathbb{R}^n$ be a convex, differentiable surrogate. Let $\ell : \mathcal{R} \to \mathbb{R}^n$ be a discrete target loss, with finite property $\gamma := \text{prop}[L]$. Suppose $L$ strongly indirectly elicits $\ell$. Under Assumption 1, $(L, \psi)$ is calibrated with respect to $\ell$, for any link $\psi : \mathbb{R}^d \to \mathcal{R}$, that satisfies the following:*

$$\psi(u) \in \gamma(p), \text{ for any } p \in \Gamma_v \text{ where } v \in \text{proj}_{\Gamma(\Delta_n)}(u)$$

*Proof.* First observe that a link of form $\psi$ exists since for any $u \in \mathbb{R}^d$, $\text{proj}_{\Gamma(\Delta_n)}(u)$ is non-empty and well-defined as $\Gamma(\Delta_n)$ is closed by Lemma 30. Now, we show that for any such link $\psi$, $(L, \psi)$ satisfies calibration with respect to $\ell$.

For any $p \in \Delta_n$, we know from Lemma 29 that there exists some $\epsilon_p > 0$, such that for any $v \in \Gamma(B_{\epsilon_p}(\Gamma_{\Gamma(p)}))$, it holds that $\gamma(q) \subseteq \gamma(p), \forall q \in \Gamma_v$. Then, we know from Lemma 27, that there exists $\delta_p > 0$ such that $\Gamma(\Delta_n) \cap B_{\delta_p}(\Gamma(p)) \subseteq \Gamma(B_{\epsilon_p}(\Gamma_{\Gamma(p)}))$. So,

$$\forall v \in \Gamma(\Delta_n) \cap B_{\delta_p}(\Gamma(p)), \gamma(q) \subseteq \gamma(p), \forall q \in \Gamma_v . \tag{7}$$

Now, suppose there exists a link $\psi : \mathbb{R}^d \to \mathcal{R}$ of the form proposed in the theorem statement, such that $(L, \psi)$ is not calibrated. This means, there exists $p \in \Delta_n$, such that

$$\inf_{u \in \mathbb{R}^d : \psi(u) \notin \gamma(p)} \langle p, L(u) \rangle = \inf_{u \in \mathbb{R}^d} \langle p, L(u) \rangle .$$

Thus, there exists some sequence $\{u_t\}_{t \in \mathbb{N}_+}$ such that $\psi(u_t) \notin \gamma(p)$ but $\lim_{t \to \infty} \langle p, L(u_t) \rangle \to \inf_{u \in \mathbb{R}^d} \langle p, L(u) \rangle$. It follows from Lemma 17 in Appendix C, that $\lim_{t \to \infty} \text{dist}(u_t, \Gamma(p)) \to 0$.

Thus, for some $t \in \mathbb{N}_+$, it holds that $\text{dist}(u_t, \Gamma(p)) < \delta_p/4$. Since $\Gamma(p)$ is compact, there must be some $u^* \in \Gamma(p)$, such that $\|u_t - u^*\|_2 < \delta_p/4$. However, since $\psi(u_t) \notin \gamma(p)$, there exists some $v \in \text{proj}_{\Gamma(\Delta_n)}(u_t)$, such that $\|u_t - v\|_2 \le \|u_t - u^*\|_2 < \delta_p/4$. Further, by the link definition, $\psi(u_t) \in \gamma(q)$, and since $\psi(u_t) \notin \gamma(p)$, it holds that $\gamma(q) \not\subseteq \gamma(p)$, for $q \in \Gamma_v$. However, this contradicts condition (7) as $\|u^* - v\|_2 \le \|u^* - u_t\|_2 + \|u_t - v\|_2 < \delta_p/2$. Thus, no such sequence exists and so

$$\inf_{u \in \mathbb{R}^d : \psi(u) \notin \gamma(p)} \langle p, L(u) \rangle > \inf_{u \in \mathbb{R}^d} \langle p, L(u) \rangle .$$

Hence $(L, \psi)$ is calibrated with respect to $\ell$. $\qquad\square$

## G  Equivalence of Strong Indirect Elicitation and Calibration under Strong Convexity

**Lemma 31.** *Let* $L : \mathbb{R}^d \to \mathbb{R}^n$ *be a convex, differentiable surrogate. Consider the function,* $F_L : \Delta_n \times \mathbb{R}^d \to \mathbb{R}$, *where* $F_L(p, u) = \langle p, L(u) \rangle$. $F_L$ *is continuous.*

*Proof.* Let $\{(p_t, u_t)\}_{t \in \mathbb{N}_+}$ be a sequence in $\Delta_n \times \mathbb{R}^d$, such that $\lim_{t \to \infty} (p_t, u_t) \to (p, u)$, for some $p \in \Delta_n, u \in \mathbb{R}^d$. We need to show that $\lim_{t \to \infty} \langle p_t, L(u_t) \rangle \to \langle p, L(u) \rangle$.

$$\begin{aligned} |\langle p_t, L(u_t) \rangle - \langle p, L(u) \rangle| &\le |\langle p_t, L(u_t) \rangle - \langle p_t, L(u) \rangle| + |\langle p_t, L(u) \rangle - \langle p, L(u) \rangle| \\ &\le \|p_t\| \cdot \|L(u_t) - L(u)\| + \|p_t - p\| \cdot \|L(u)\| \\ &\le 1 \cdot \|L(u_t) - L(u)\| + \|L(u)\| \cdot \|p_t - p\| \end{aligned}$$

Taking the limit as $t \to \infty$, $\|L(u_t) - L(u)\| + \|L(u)\| \cdot \|p_t - p\| \to 0$ since $\|p_t - p\| \to 0$ by construction, and $\|L(u_t) - L(u)\| \to 0$ as $u_t \to u$ and $L_y(\cdot)$ is continuous for each $y \in [n]$, and thus $L_y(u_t) \to L_y(u), \forall y \in [n] \implies \|L(u_t) - L(u)\| \to 0$. Thus, $\lim_{t \to \infty} \langle p_t, L(u_t) \rangle = \langle p, L(u) \rangle$, and so $\lim_{t \to \infty} F_L(p_t, u_t) \to F_L(p, u)$, whenever $\lim_{t \to \infty}(p_t, u_t) \to (p, u)$. Hence, $F_L$ is continuous. $\qquad\square$

**Lemma 32.** *Let* $f : \mathbb{R}^d \to \mathbb{R}$ *be a differentiable, strongly convex function. Then* $\arg\min_{u \in \mathbb{R}^d} f(u)$ *exists and is a singleton.*

**Lemma 33.** *[Boyd, 2004] Let* $f : \mathbb{R}^d \to \mathbb{R}$ *be* $\mu_f$-*strongly convex and let* $g : \mathbb{R}^d \to \mathbb{R}$ *be* $\mu_g$-*strongly convex. Then* $f + g$ *is* $(\mu_f + \mu_g)$-*strongly convex. For any* $\alpha > 0$, *the function* $\alpha \cdot f$ *is* $\alpha \cdot \mu_f$-*strongly convex. Also,* $f$ *is* $\mu$-*strongly convex for every* $0 < \mu \le \mu_f$.

**Lemma 34.** *Let* $L : \mathbb{R}^d \to \mathbb{R}^n$ *be a convex, differentiable surrogate. Suppose for each* $y \in [n]$ $L_y : \mathbb{R}^d \to \mathbb{R}$ *is* $\mu_y$-*strongly convex, where* $\mu_y > 0$. *Let* $\mu_m := \min\{\mu_i\}_{i=1}^n$. *Then* $\langle p, L(\cdot) \rangle$ *is* $\mu_m$-*strongly convex, for every* $p \in \Delta_n$.

*Proof.* Let $p \in \Delta_n$. For any $y \in [n]$, $p_y \cdot L_y$ is $p_y \cdot \mu_y$-strongly convex by Lemma 33. Also, $\langle p, L(\cdot) \rangle = \Sigma_{y \in [n]} p_y \cdot L_y(\cdot)$ is $\Sigma_{y \in [n]} p_y \cdot \mu_y$- strongly convex by Lemma 33. Notice that, $\Sigma_{y \in [n]} p_y \cdot \mu_y \le \Sigma_{y \in [n]} p_y \cdot \mu_m = \mu_m$. Thus, $\langle p, L(\cdot) \rangle$ is $\mu_m$-strongly convex by Lemma 33. $\qquad\square$

**Lemma 35.** *Let* $L : \mathbb{R}^d \to \mathbb{R}^n$ *be a surrogate loss, with strongly convex, differentiable components. Then,* $\Gamma : \Delta_n \rightrightarrows \mathbb{R}^d$ *is single-valued and continuous.*

*Proof.* Suppose for each $y \in [n]$ $L_y : \mathbb{R}^d \to \mathbb{R}$ is $\mu_y$-strongly convex, where $\mu_y > 0$. Let $\mu_m := \min\{\mu_i\}_{i=1}^n$. Then we know by Lemma 34, that $\langle p, L(\cdot) \rangle$ is $\mu_m$-strongly convex for every $p \in \Delta_n$. Fix some $p \in \Delta_n$. Let $\{p_t\}_{t \in \mathbb{N}_+}$ be a sequence of distributions in $\Delta_n$, such that $\lim_{t \to \infty} p_t \to p$. We know from Lemma 32 that $\Gamma(p_t)$ exists and is single-valued for each $p_t$ since $\langle p_t, L(\cdot) \rangle$ is differentiable and strongly convex. Suppose $u_t \in \mathbb{R}^d$ such that $u_t = \Gamma(p_t)$, for each $t \in \mathbb{N}_+$. We need to show that $\lim_{t \to \infty} u_t \to u^*$ where $u^* = \Gamma(p)$ (again, $\Gamma(p)$ exists and is single-valued).

Suppose $v \in \mathbb{R}^d$. We know from Lemma 31 that the function $F : \Delta_n \times \mathbb{R}^d$, where $F_L(\cdot, \cdot) = \langle \cdot, L(\cdot) \rangle$ is continuous. Then define $m_v := \min_{u \in \partial B_1(v), q \in \Delta_n} \langle q, L(u) \rangle$ and $M_v := \max_{q \in \Delta_n} \langle q, L(v) \rangle$. Both $m_v, M_v$ exist since $F_L$ is continuous, $\partial B_1(v) \times \Delta_n$ and $\Delta_n \times \{v\}$ are compact. Now, pick any

$w \in \mathbb{R}^d : \|w\| = 1$. Let $\beta > \max\{1, 1 + \frac{2 \cdot (M_v - m_v)}{\mu_m}\}$. Let $v_1 := v + w$ and $v_2 := v + \beta \cdot w$. Notice, $v_1 = \frac{\beta - 1}{\beta} v + \frac{1}{\beta} v_2$. Next, for any $q \in \Delta_n$, we have that:

$$
\begin{aligned}
m_v &\leq \langle q, L(v_1) \rangle \\
&\leq \frac{\beta - 1}{\beta} \langle q, L(v) \rangle + \frac{1}{\beta} \langle q, L(v_2) \rangle - \frac{1}{2} \cdot \mu_m \cdot \frac{\beta - 1}{\beta} \cdot \frac{1}{\beta} \cdot \|v - v_2\|^2 \\
&= \frac{\beta - 1}{\beta} \langle q, L(v) \rangle + \frac{1}{\beta} \langle q, L(v_2) \rangle - \frac{1}{2} \cdot \mu_m \cdot (\beta - 1)
\end{aligned}
$$

So we have established that,

$$
m_v \leq \frac{\beta - 1}{\beta} \langle q, L(v) \rangle + \frac{1}{\beta} \langle q, L(v_2) \rangle - \frac{1}{2} \cdot \mu_m \cdot (\beta - 1) \tag{8}
$$

The first inequality holds due to the definition of $m_v$. The second inequality follows by $\mu_m$-strong convexity and the final equality follows from the definition of $v_2$. Now, we claim that, $\langle q, L(v_2) \rangle > \langle q, L(v) \rangle$. Assume to the contrary that $\langle q, L(v_2) \rangle \leq \langle q, L(v) \rangle$. Since, $M_v \geq \langle q, L(v) \rangle$, we get by (8) that, $m_v \leq M_v - \frac{1}{2} \cdot \mu_m \cdot (\beta - 1) \implies m_v - M_v \leq -\frac{1}{2} \cdot \mu_v \cdot (\beta - 1) \implies \beta \leq 1 + \frac{2 \cdot (M_v - m_v)}{\mu_m}$, which violates the condition for choosing $\beta > \max\{1, 1 + \frac{2 \cdot (M_v - m_v)}{\mu_m}\}$. Thus, $\langle q, L(u) \rangle > \langle q, L(v) \rangle$ for any $u : \|v - u\| > \max\{1, 1 + \frac{2 \cdot (M_v - m_v)}{\mu_m}\}$ and any $q \in \Delta_n$, since $v_2 = v + \beta \cdot w$ for arbitrary $w \in \mathbb{R}^d : \|w\| = 1$ and $\beta > \max\{1, 1 + \frac{2 \cdot (M_v - m_v)}{\mu_m}\}$ was chosen arbitrarily, followed by which $q \in \Delta_n$ was also arbitrarily picked. Thus, for any $q \in \Delta_n$, $\Gamma(q)$ must be such that $\|v - \Gamma(q)\| \leq \max\{1, 1 + \frac{2 \cdot (M_v - m_v)}{\mu_m}\}$ since $\langle q, L(\Gamma(q)) \rangle \leq \langle q, L(v) \rangle$. Thus, $\Gamma(\Delta_n)$ is uniformly bounded in a ball around $v$. And so, the sequence $\{u_t\}_{t \in \mathbb{N}_+} = \Gamma(p_t)_{t \in \mathbb{N}_+}$ must be bounded as well. Thus, there exists a $u' \in \mathbb{R}^d$ and a subsequence $\{u_{t_j}\}_{j \in \mathbb{N}_+}$ such that $\lim_{j \to \infty} u_{t_j} \to u'$. By definition, $\langle p_{t_j}, L(u_{t_j}) \rangle \leq \langle p_{t_j}, L(u) \rangle$ for every $u \in \mathbb{R}^d$. Thus, by continuity of $F_L$, we have that $\langle p, L(u') \rangle \leq \langle p, L(u) \rangle, \forall u \in \mathbb{R}^d$. Since $\langle p, L(\cdot) \rangle$ admits a unique minimizer, it follows that $u' = u^* = \Gamma(p)$. Thus, every convergent subsequence of $\{u_t\}_{t \in \mathbb{N}_+}$ must converge to the same limit $u^*$, and as $\{u_t\}_{t \in \mathbb{N}_+}$ is bounded, $\lim_{t \to \infty} u_t = u^* = \Gamma(p)$.

Thus, we have shown that for any $p \in \Delta_n$ and any sequence of distributions $\{p_t\}_{t \in \mathbb{N}_+}$, such that $\lim_{t \to \infty} p_t \to p$, it follows that $\Gamma(p_t) \to \Gamma(p)$. Thus, $\Gamma$ is continuous. $\qquad \square$

**Lemma 36.** *Let $L : \mathbb{R}^d \to \mathbb{R}^n$ be a surrogate loss with strongly convex, differentiable components. Let $\ell : \mathcal{R} \to \mathbb{R}^n$. If $L$ indirectly elicits $\ell$, but does not strongly indirectly elicit $\ell$, there exists some report $u \in \mathbb{R}^d$, such that $\gamma(p_m) \subset \gamma(p)$, for some $p_m, p \in \Gamma_u$.*

*Proof.* We claim that $\exists p', q' \in \Gamma_u$, such that $|\gamma(p')| \neq |\gamma(q')|$. Assume to the contrary that for every $p, q \in \Gamma_u$, $|\gamma(p)| = |\gamma(q)|$. So, we have that $\exists u \in \mathbb{R}^d$, such that $\gamma(p^*) \neq \gamma(q^*)$, but $|\gamma(p^*)| = |\gamma(q^*)|$. Sine $L$ indirectly elicits $\ell$, we have by Lemma 10 that $\gamma(p^*) \cap \gamma(q^*) \neq \emptyset$. So $\exists S \subseteq R : S \neq \emptyset$ and $S = \gamma(p^*) \cap \gamma(q^*)$, while $S \subset \gamma(p^*)$ and $S \subset \gamma(q^*)$. In particular, this means $|S| < |\gamma(p^*)|$. Now, we know by Lemma 8 that $\frac{p^* + q^*}{2}$ is such that $\gamma(\frac{p^* + q^*}{2}) \subseteq S$. This means $|\gamma(\frac{p^* + q^*}{2})| \leq |S| < |\gamma(p^*)| \implies |\gamma(\frac{p^* + q^*}{2})| < \gamma(p)$ which contradicts our assumption since $\frac{p^* + q^*}{2} \in \Gamma_u$ by convexity of $\Gamma_u$. Thus, $\exists p', q' \in \Gamma_u$, such that $|\gamma(p')| \neq |\gamma(q')|$.

Let $p_m \in \Gamma_u : |\gamma(p_m)| \leq |\gamma(p)|, \forall p \in \Gamma_u$. We know by Lemma 9 that $\gamma(p_m) \subseteq \gamma(p), \forall p \in \Gamma_u$. In fact, $\exists p \in \Gamma_u : |\gamma(p)| \neq |\gamma(p_m)| \implies |\gamma(p_m)| < |\gamma(p)|$ and so $\gamma(p_m) \subset \gamma(p)$. $\qquad \square$

**Theorem 12.** *Let $L : \mathbb{R}^d \to \mathbb{R}^n$ be a surrogate loss, with strongly convex, differentiable components. Let $\ell : \mathcal{R} \to \mathbb{R}^n$. If $L$ does not strongly indirectly elicit $\ell$, then there is no link function $\psi : \mathbb{R}^d \to \mathcal{R}$, such that $(L, \psi)$ satisfies calibration with respect to $\ell$.*

*Proof.* First, suppose $L$ does not indirectly elicit $\ell$. Then straight away, calibration fails since calibration implies indirect elicitation by Theorem 6. Now, suppose $L$ indirectly elicits $\ell$, but does not strongly indirectly elicit $\ell$. We know from Lemma 36, that for some $u \in \mathbb{R}^d$, $\gamma(p_m) \subset \gamma(p)$, where $p_m, p \in \Gamma_u$. Assume WLOG that $\gamma(p_m) = \{1, 2, ..., t\}$ and that $\gamma(p) = \{1, 2, ..., t, ..., t + j\}$

where $j \geq 1$. In particular, $p$ lies on the boundary of the cell $\gamma_{t+1}$, which is a convex polytope. Thus, we can pick a sequence $\{p_i\}_{i \in \mathbb{N}_+}$, such that $\gamma(p_i) = \{t+1\}, \forall i \in \mathbb{N}_+$, and $\lim_{i \to \infty} p_i \to p$. Define $v_i := \Gamma(p_i)$, for every $i \in \mathbb{N}_+$. Then, we have by Lemma 35 that since each of the components of $L$ are differentiable and strongly convex, it holds that $\Gamma$ is continuous and hence $\lim_{i \to \infty} v_i = u$. To ensure calibration at $p_m$, it is necessary $\psi(u) \in \gamma(p_m)$, since $u = \Gamma(p_m)$ and $\gamma(p_m) \subseteq \gamma(p), \forall p \in \Gamma_u$. Also, to ensure calibration at $p_i$ it is necessary that $\psi(v_i) = t+1$, since $\gamma(v_i) = \{t+1\}$ for every $i \in \mathbb{N}_+$. However, despite this, we show calibration fails:

$$
\begin{aligned}
\inf_{v \in \mathbb{R}^d : \psi(v) \notin \gamma(p_m)} \langle p_m, L(v) \rangle &\leq \inf_{v \in \mathbb{R}^d : \psi(v) = t+1} \langle p_m, L(v) \rangle \\
&\leq \inf_{v \in \mathbb{R}^d : v \in \{v_i\}_{i \in \mathbb{N}_+}} \langle p_m, L(v) \rangle \\
&\leq \lim_{i \to \infty} \langle p_m, L(v_i) \rangle \\
&= \langle p, L(u) \rangle = \inf_{v \in \mathbb{R}^d} \langle p, L(v) \rangle
\end{aligned}
$$

Hence, we have shown that $\inf_{v \in \mathbb{R}^d : \psi(v) \notin \gamma(p_m)} \langle p_m, L(v) \rangle = \inf_{v \in \mathbb{R}^d} \langle p, L(v) \rangle$, thus violating calibration at $p_m$. $\qquad \square$

## H  Constructing 1d surrogates for orderable properties

In this section, we provide an explicit construction of a consistent, convex, differentiable surrogate with domain dimension 1 for a given orderable target loss. Formally, given an orderable target, $\ell : \mathcal{R} \to \mathbb{R}^n$, we prove constructively the existence of a convex, differentiable surrogate $L : \mathbb{R} \to \mathbb{R}^n$ that is consistent with respect to $\ell$.

Our construction hinges on a subroutine which we will call `LinIntGrad`$(X)$. Given a vector $X \in \mathbb{R}^{k-1}$, for some integer $k \geq 2$, `LinIntGrad` constructs a function $f : \mathbb{R} \to \mathbb{R}$ that is convex, differentiable and whose gradients match $X$ at inputs $\{1, 2, .., k-1\}$, i.e., $f'(i) = X[i], \forall i \in [k-1]$. See Subroutine 1 for the detailed construction.

---

**Subroutine 1** `LinIntGrad`$(X)$

---

1: **Input:** $X[1], \ldots, X[k-1]$ (with $k \geq 2$)
2: **Goal:** Define a gradient map $g : \mathbb{R} \to \mathbb{R}$ and $f(x) = \int_1^x g(t)\, dt$

3: **(A) Define** $g$ **on** $(1, k-1)$ **by linear interpolation**
4: **for** $j = 1, 2, \ldots, k-2$ **do**
5:     For $x \in (j, j+1] \setminus \{k-1\}$, set

$$
g(x) \leftarrow X[j] + (x - j)\big(X[j+1] - X[j]\big).
$$

6: **(B) Left extrapolation on** $(-\infty, 1]$
7: For $x \leq 1$, set
$$
g(x) \leftarrow X[1] + (x - 1)
$$

8: **(C) Right extrapolation on** $[k-1, \infty)$
9: For $x \geq k-1$, set
$$
g(x) \leftarrow X[k-1] + \big(x - (k-1)\big)
$$

10: **(D) Define** $f$
11: $f(x) \leftarrow \int_1^x g(t)\, dt$
12: **return** $f$

---

**Lemma 37.** *Given some $X \in \mathbb{R}^{k-1}$, where $k \geq 2$ is an integer. Following the notation of Subroutine 1, let $g$ be the gradient-map constructed and let $f = $ `LinIntGrad`$(X)$. Then:*

*(i) $g$ is continuous and nondecreasing on $\mathcal{R}$.*

*(ii) $f$ is convex and $C^1$ on $\mathcal{R}$, with $f'(x) = g(x)$ for all $x$.*

*(iii)* $f'(i) = X[i]$ *for each* $i \in \{1, \ldots, k-1\}$.

*(iv) Minimizer location:*

- *In Case 1 ($X[1] \leq 0 \leq X[k-1]$), one has* $\arg\min f \subseteq [1, k-1]$. *If* $X \equiv 0$ *on* $\{1, \ldots, k-1\}$ *then* $\arg\min f = [1, k-1]$; *otherwise g crosses 0 inside* $[1, k-1]$ *and the minimizer lies there (unique if the crossing is strict).*
- *In Case 2, when $X[1] > 0$,* $\arg\min f = \{1 - X[1]\}$
- *In Case 3, when $X[k-1] < 0$,* $\arg\min f = \{k - 1 - X[k-1]\}$.

*(v)* $\arg\min f$ *is nonempty and compact.*

*Proof. (i) Continuity and monotonicity of g.* First, consider the behavior of $g$ on the interval $(1, k-1)$ – see **(A)** in Subroutine 1. On each sub-interval, i.e., on each element of the set $\{(j, j+1] \setminus \{k-1\} | j \in [k-2]\}$, $g$ is either affine with nonnegative slope or constant. Thus, $g$ is continuous on $(j, j+1)$, and nondecreasing on $(j, j+1] \setminus \{k-1\}, \forall j \in [k-2]$. Let $j \in \{2, 3, \ldots, k-2\}$. We have that $g(j^-) := \lim_{x \to j^-} X[j-1] + (x - (j-1))(X[j] - X[j-1]) = X[j]$, while $g(j^+) := \lim_{x \to j^+} X[j] + (x - j)(X[j+1] - X[j]) = X[j]$. Thus, $g(j^-) = g(j^+) = X[j] = g(j)$. Hence $g$ is continuous on $(1, j-1)$. We already know $g$ is non-decreasing on each subinterval. Notice also that, $g(j) = X[j] \leq X[j+1] = g(j+1)$. Hence, $g$ is non-decreasing and continuous on $(1, j-1)$.

We now analyze continuity at $x = 1$; see **(B)** in Subroutine 1: for $x \leq 1$, we set $g(x) = X[1] + (x-1)$. For $x \in (1, 2)$, we have $g(x) = X[1] + (x-1)(X[2] - X[1])$. Clearly, $\lim_{x \to 1^-} g(x) = X[1] = \lim_{x \to 1^+} g(x)$. Thus $g$ is continuous at $x = 1$.

A similar check at $x = k - 1$; see **(C)** in Subroutine 1: For $x \in (k-2, k-1)$, $g(x) = X[k-2] + (x - (k-2))(X[k-1] - X[k-2])$. So, $\lim_{x \to (k-1)^-} g(x) = X[k-1]$. Whereas, for $x \geq k - 1$, $g(x) = X[k-1] + (x - (k-1))$. So, $\lim_{x \to (k-1)^+} g(x) = X[k-1]$. Hence $g$ is continuous at 1 and $k - 1$, which means $g$ is continuous on $[1, k-1]$. On $(-\infty, 1)$, as well as $(k-1, \infty)$, $g$ is affine with non-negative slope. Thus, $g$ is continuous and non-decreasing on $\mathbb{R}$.

*(ii) Convexity and differentiability of f.* Since $f(x) = \int_1^x g(t) \, dt$, $f$ is $C^1(\mathbb{R})$ with $f' = g$ everywhere. So $f'$ is monotone (since $g$ is nondecreasing), and hence $f$ is convex.

*(iii) Gradients match X in $[k-1]$.* For each $j \in \{1, \ldots, k-1\}$, the interpolation $g(x) = X[j] + (X[j+1] - X[j])(x - j)$ gives $g(j) = X[j]$. Hence $f'(j) = g(j) = X[j]$.

*(iv) Minimizer location.* Recall that for differentiable convex $f$, any minimizer $x^\star$ satisfies $f'(x^\star) = 0$, and conversely if $f'$ changes sign from negative to positive at $x^\star$, then $x^\star$ is the unique minimizer.

Case 1 ($X[1] \leq 0 \leq X[k-1]$): If $X[1] < 0$, then $g(x) = X[1] < 0, \forall x \leq 1$. Whereas, if $X[1] = 0$, then $g(x) = x - 1 < 0, \forall x < 1$. Either way, $g(x) \neq 0$ for any $x < 0$. Similarly, if $X[k-1] > 0$, then $g(x) = X[k-1] > 0, \forall x \geq k - 1$. Whereas, if $X[k-1] = 0$, $g(x) = x - (k-1) > 0, \forall x > k - 1$. Either way, $g(x) \neq 0$ for any $x > k - 1$. Thus, $g \neq 0$ on $(-\infty, 1) \cup (1, \infty)$.

On $[1, k-1]$, $g$ is continuous and nondecreasing with $g(1) = X[1] \leq 0$ and $g(k-1) = X[k-1] \geq 0$, hence any zero of $g$ lies in $[1, k-1]$. If $X \equiv 0$ on $\{1, \ldots, k-1\}$, we have $g \equiv 0$ on $[1, k-1]$. If not, $g$ must cross 0 somewhere in $[1, k-1]$ due to continuity. Thus, $\arg\min f \subseteq [1, k-1]$ and $\arg\min f \neq \emptyset$. So $\arg\min f$ is non-empty and bounded. Since $f$ is convex, $\arg\min f$ is closed, and hence compact.

Case 2 ($X[1] > 0$): First notice that in this case, since $X[1] > 0$, $g(1) > 0$ and since $g$ is nondecreasing it follows that $g(x) > 0$ for every $x \geq 1$. So any zero $g$ attains must be in $(-\infty, 1)$. For $x < 1$, $g(x) = X[1] + (x - 1) = 0 \iff x = 1 - X[1] < 1$. Since $g$ is strictly increasing on $(-\infty, 1)$, it crosses 0 exactly once at $1 - X[1]$. Thus, $\arg\min f = \{1 - X[1]\}$ which is non-empty and compact.

Case 2 ($X[k-1] < 0$): First notice that in this case, since $X[k-1] < 0$, $g(k-1) < 0$ and since $g$ is nondecreasing it follows that $g(x) < 0$ for every $x \leq k - 1$. So any zero $g$ attains must be in $(k-1, \infty)$. For $x > k-1$, $g(x) = X[k-1] + (x - (k-1)) = 0 \iff x = k-1-X[k-1] > k-1$. Since $g$ is strictly increasing on $(k-1, \infty)$, it crosses 0 exactly once at $k - 1 - X[k-1]$. Thus, $\arg\min f = \{k - 1 - X[k-1]\}$ which is non-empty and compact.

*(v)* $\arg\min f$ *is non-empty and compact.* Follows directly from part *(iv)*.  □

---

**Construction 2** `SURROGATE CONSTRUCTION`

---
1: **Inputs:** $V^1, \ldots, V^n \in \mathbb{R}^{k-1}$  (rows of matrix $\mathbf{V}$)
2: **Output:** Consistent, convex and differentiable surrogate loss $L : \mathbb{R} \to \mathbb{R}^n$
3: **for all** $j \in [n]$ **do**
4:    $L(\cdot)_j \leftarrow$ `LinIntGrad`$(V^j)$
5: **Define** $L : L(u) \leftarrow [L(u)_1, L(u)_2, ..., L(u)_n]$
6: **return** $L$

---

**Theorem 13.** *Given an orderable target $\ell : \mathcal{R} \to \mathbb{R}^n$, there exists a convex, differentiable surrogate $L : \mathbb{R} \to \mathbb{R}^n$ satisfying Assumption 1, which is calibrated with respect to $\ell$. In particular, Construction 2 yields such a surrogate loss $L$.*

*Proof.* Since $\gamma = \text{prop}[\ell]$ is orderable, there exists an orderable enumeration of $\mathcal{R}$, i.e., $E_\gamma = (r_1, r_2, ..., r_k)$ (see Definition 7). We know from Theorem 11 of Finocchiaro et al. [2020] that there exists a set $\{v_1, v_2, ..., v_{k-1}\} \subset \mathbb{R}^n$ such that:

1. The set satisfies coordinate-wise monotonicity. That is, $\forall i \in [k-2], y \in [n]$ it holds that, $v_{i,y} \leq v_{i+,y}$.

2. The set of vectors are normal to target boundaries, i.e., $\forall p \in \gamma_{r_i} \cap \gamma_{r_{i+1}}, \langle p, v_i \rangle = 0$.

Let us denote by $\mathbf{V} \in \mathbb{R}^{n \times k-1}$ the matrix with column vectors $v_1, v_2, ..., v_{k-1}$ in that order. Let $V^j, j \in [n]$ denote the $j^{\text{th}}$ row of $\mathbf{V}$. These row-vectors are set as inputs to the surrogate construction described in Construction 2. We now show that the output of $L = $ `SURROGATE CONSTRUCTION`$(V^1, V^2, ..., V^n)$ is convex, differentiable, satisfies Assumption 1 and is consistent w.r.t. $\ell$.

Each component $L(\cdot)_j, j \in [n]$ defined in Construction 2, is obtained via Subroutine 1. Thus, for each $j \in [n]$, $L(\cdot)_j$ is convex, differentiable and satisfies Assumption 1 by Lemma 37. We show that $L$ indirectly elicits $\ell$ and the result then follows by Theorem 1.

Assume $L$ does not indirectly elicit $\ell$. This means, there exists some $u \in \mathbb{R}$, such that $\Gamma_u \nsubseteq \gamma_r, \forall r \in \mathcal{R}$. In particular, this means that the level-set $\Gamma_u$ crosses from one target cell's relative interior into another target cell's relative interior. By convexity of $\Gamma_u$, there exists some $j \in [k-1]$ and some $p \in \gamma_{r_j} \cap \gamma_{r_{j+1}}$ such that $p \in \Gamma_u$. By construction, $\nabla L(j) = v_j$. Clearly, $u \neq j$ as $\nabla L(u) \neq v_j$. Assume WLOG that $u < j$. This means $\nabla L(u) = \nabla L(j) - \delta$, where $\delta_i \geq 0, \forall i \in [n]$ and $\delta_{i^*} > 0$ for some $i^* \in [n]$. So, $\langle p, \nabla L(u) \rangle = \langle p, \nabla L(j) - \delta \rangle = -\langle p, \delta \rangle < 0$ by the condition on $\delta$. Thus, $p \notin \Gamma_u$ yielding a contradiction. Hence $L$ must indirectly elicit $\ell$.  □

