# OpenReview forum: "Consistency Conditions for Differentiable Surrogate Losses"
_NeurIPS.cc/2025/Conference — NeurIPS 2025 poster_

### Official Review · Reviewer_3bYS · 2025-06-08

**Clarity:** 2
**Significance:** 3
**Originality:** 3
**Rating:** 5
**Confidence:** 3

**Summary:**

With a multiclass output, verifying the consistency of a surrogate loss is a challenging task. The authors argue that examining the properties "indirect elicitation" and "strong indirect elicitation" provides insight on consistency. These two properties are easier to verify than consistency. Specifically, they prove that with when the input is one dimensional, consistency is equivalent to a property called "Indirect Elicitation", which essentially requires that distributions minimize surrogate are always included also minimize the target loss. However, this equivalence is false when the input is higher dimensional. In this case, a property called "Strong Indirect Elicitation" implies consistency. This property essentially requires that the set of minimizers of the surrogate is included in some intersection of sets of minimizers of the target loss. For strongly convex losses, strong indirect elicitation is equivalent to consistency.

**Questions:**

**Questions:**
- Verifying consistency is simple in the two class case. See for instance Theorem 2 of [1], Theorem 3.1 of [2], or Theorem 2 of [3]. Can you give an intuitive explanation of why this is difficult for multiple classes?
- Could you provide an intuitive explanation of what indirect elicitation and strong indirect elicitation entail?
- I tried to working through Example 1 indepentently and found different behavior than what was described in the text. Specifically:
$$ \langle p, L(u)\rangle=p_1( (1-u)^2+|u|)+p_2( (1+u)^2+|u|) ,$$ which is always minimized at zero. Could you clarify this calculation and provide further details?
- Can you explain the intuition why the proof strategy of theorem 1 does not extend to the multi-input case?
- Similarly, it would be helpful to emphasize exactly where Theorem 2 uses the property $\gamma_S^*=\{p:-\gamma(p)=S\}$, rather than $\gamma_r=\{p:r\in \gamma(p)\}$.



**Other Comments:**
- line 63 should specify that this result is restricted to convex surrogates
-Line 79: it would be helpful to discuss an example when $\mathcal R\neq \mathcal Y$, such as classification with a reject option referred to later in the paper.
-line 223: it would be helpful to explicitly state what $p$ is here



[1] Peter L. Bartlett and Michael I. Jordan and Jon D. McAuliffe. "Convexity, Classification, and risk bounds."
[2] Yi Lin. "A note on margin-based loss functions in classification"
[3] Natalie Frank. "Adversarial Surrogate Risk Bounds for Binary Classification".

**Ethical Concerns:**

["NO or VERY MINOR ethics concerns only"]

**Final Justification:**

I maintain my score. As stated in my review, I believe this work makes a good contribution to the theory of surrogate risks. I think the biggest problem with this submission is its clarity. Through their comments, the authors have convinced me that they will work to improve the clarity of their paper prior to publication.

**Limitations:**

Some losses don't have minimizers for all vectors $p$ in the simplex, see [1]. Some examples are the exponential loss for binary classification and the cross entropy loss on the simplex. How do your results apply to such losses?/ Is this solved via a link function?

[1] Ambuj Tewari and Peter Bartlett. *On the Consistency of Multiclass Classification Methods*. JMLR. 2007.

**Quality:**

3

**Strengths And Weaknesses:**

**Strengths**:
**Quality and Significance:** This paper addresses a compelling problem-- understanding when studying minimizers of losses suffices to understand consistency. The results are convincing.

**Weaknesses**:
**Clarity:** There is a lot of technical notation and technical jargon in this paper that makes the results difficult to parse. I understand that the nature of this result is highly technical, but the authors could make this paper more patable by discussing the intuition surrounding the main objects of this paper.

---

> ### Author Rebuttal · Authors · 2025-07-31
>
> Thank you for your time and helpful feedback. We will improve and clarify the technical notation in the paper. We address specific concerns and questions below.
>
> $$\textbf{“Verifying consistency is simple in the two class case. … Can you give an intuitive explanation of why this is difficult for multiple classes?”}$$
> In the binary case, a scalar-score suffices for classification-calibration: it’s enough to determine whether $\eta(x) = \mathbb{P}(Y = 1 \mid X = x)$ is above or below 0.5. For example, the derivative-rule from Bartlett et al. (2006) [1] states that a convex margin loss $\phi$ is consistent w.r.t. $0-1$ loss iff it’s differentiable at $0$ with $\phi'(0) < 0$. This condition ensures that the expected risk $\eta(x)\phi(t) + (1 - \eta(x))\phi(-t)$ decreases when $t$ moves in the correct direction away from $0$—negative if $\eta(x) < 0.5$, positive otherwise. Thus, if $\eta(x) < 0.5$, the minimizer $t^{\ast}(x) < 0$, and the link $\psi(t^{\ast}(x)) = sign(t^{\ast}(x)) = -1$ yields the correct prediction.
>
> In contrast, no such scalar-derivative condition exists in the multiclass case. Even for $n = 3$, predictions depend on the full vector $\eta(x) = [\eta_1(x), \eta_2(x), \eta_3(x)]$, where $\eta_i(x) = \mathbb{P}(Y = i \mid X = x)$. Suppose $\arg\max_i \eta_i(x) = {3}$—then predicting class 3 is optimal for the $3$-class $0-1$ loss. At the uniform distribution $p = (1/3, 1/3, 1/3)$, we would want the surrogate loss’s gradient to decrease in all directions where coordinate 3 dominates and increase otherwise. This requires controlling an entire gradient field, not just a scalar.
>
> The absence of a simple derivative-rule motivated the formal notion of calibration in Tewari and Bartlett (2007) [2]. But checking calibration remains difficult: as noted as well on page 5 of the seminal paper [2]. Doing so involves analyzing minimizing-sequences of the surrogate loss, which is hard in $\geq$ 2D due to the uncountably many sequences surrounding the minimizer (see also Figure 1, pg. 2 of our paper for visual intuition).
>
> $$\textbf{“Could you provide an intuitive explanation of what indirect elicitation and strong indirect elicitation entail?”}$$
> The key advantage of IE and strong IE is that they depend only on surrogate minimizers, unlike calibration, which also requires reasoning about $\textit{all}$ sequences converging to those minimizers (see Figure 1, pg. 2). Specifically, for any distribution $p$, calibration demands not only that the surrogate minimizer $\Gamma(p)$ links to the target minimizer $\gamma(p)$, but also that $\textit{every sequence}$ converging to $\Gamma(p)$ eventually links to $\gamma(p)$. IE only requires ensuring that $\Gamma(p)$ links to $\gamma(p)$.
>
> For geometric intuition: IE holds if each level set $\Gamma_u$ lies entirely within a target-cell. As illustrated in Figure 3 (pg. 6 of our paper), the level set $\Gamma_{0,0}$ (blue line) violates IE by crossing from $\gamma_{1}$ into $\gamma_{2}$ (left figure), and satisfies IE by lying entirely within $\gamma_2$ (middle figure). However, because it $\textit{touches}$ the red boundary between $\gamma_1$ and $\gamma_2$, strong IE is violated. Strong IE strengthens IE by requiring that level sets be fully contained and $\textit{also bounded away}$ from target boundaries (right figure).
>
> $$\textbf{“I tried to working through Example 1 indepentently … Specifically: $\langle p, L(u) \rangle = p_{1}((1-u)^{2} + |u|) + p_{2}((1+u)^{2} + |u|)$, which is always minimized at zero.”}$$
> Thank you for cross-checking. The expected loss $p_1((1 - u)^2 + |u|) + p_2((1 + u)^2 + |u|)$ is only minimized by $u=0$ for distributions in the set $\\{p \in \Delta_2: \max{p_1, p_2} \leq 0.75 \\}$. For instance, if $p_1 = 1, p_2 = 0$, the expression becomes $(1 - u)^2 + |u|$, which is minimized at $u = 0.5$.
>
> $$\textbf{“Can you explain the intuition why the proof strategy of theorem 1 does not extend to the multi-input case?”}$$
> The proof of Theorem 1 assumes the existence of a 1D convex surrogate that indirectly elicits the target—an assumption that holds $\textit{only}$ for orderable targets (see Proposition 27, pg. 17 of [3]). It therefore leverages structure specific to orderable losses. In contrast, Theorem 2 addresses general, non-orderable targets, requiring a fundamentally different proof technique to handle full generality.
>
> $$\textbf{“It would be helpful to emphasize exactly where Theorem 2 uses the property… ”}$$
> The property you're referring to is strong IE (Definition 6, pg. 7) and is used in Lemma 29 (pgs. 24–25) to prove Theorem 2. Intuitively, strong IE, combined with properties of the set-valued map $\Gamma_{(\cdot)}$, ensures that minimizers of distributions near $\Gamma_u$ don't link to reports incompatible with $u$ (for any surrogate report $u$). We will make this explicit.
>
> $$\textbf{Other Comments, regarding lines  63, 79 and 223}$$
> Thank you for pointing these out. We will be sure to incorporate your feedback.
>
> $$\textbf{“Some losses don’t have minimizers for all vectors in the simplex … How do your results apply to such losses?/ Is this solved via a link function?”}$$
> The losses you mention—e.g., cross-entropy—are minimizable in the interior of the simplex but not on its boundary. Our results apply wherever minimizers exist, i.e., in the interior. At boundary distributions IE/strong IE conditions don’t apply. In such cases, one must consider minimizing sequences, and we conjecture that addressing these cases in full-generality may require tools from the emerging theory of astral spaces ([4]).
>
> $$\textbf{References}$$
> [1] P. L. Bartlett, M. I. Jordan, and J. D. McAuliffe. "Convexity, classification, and risk bounds". Journal of the American Statistical Association, 2006.
>
> [2] A. Tewari and P. L. Bartlett, “On the Consistency of Multiclass Classification Methods.” JMLR. 2008
>
> [3] J. Finocchiaro, R. Frongillo, and B. Waggoner. "Embedding dimension of polyhedral losses" COLT 2020
>
> [4] M. Dudík, R. E. Schapire, and M. Telgarsky. “Convex analysis at infinity: An introduction to astral space.” 2022

---

> > ### Comment · Reviewer_3bYS · 2025-08-01
> >
> > ## (2)
> > This is helpful. I think you should include this intuition in the paper, and explain why the strong indirect elicitation but not IE requires that level sets are bounded away from each other.
> >
> >
> > ## (4)
> > It would be nice to understand how the orderable property is used in this proof/ why it's so important for this result.

---

> > > ### Author Response · Authors · 2025-08-02
> > >
> > > Thank you for your feedback!
> > >
> > > $\textbf{Regarding (2)}$: We are glad you found the intuition helpful, and will certainly include this in the paper.
> > >
> > > $\textbf{Regarding (4)}$: Absolutely – we clarify the role of orderability and its induced geometry here:
> > >
> > > $\textit{Note}$: A visual comparison between orderable and non-orderable properties (see Figures 1 and 2 in [1]) may be helpful while following our explanation.
> > >
> > > Let $L: \mathbb{R} \to \mathbb{R}^{n}$ be a convex, differentiable surrogate that IEs an orderable target property $\gamma$. A property is orderable if and only if the target cells can be effectively “lined-up” one after another. In other words, there exists a line-segment that intersects each of the target cells elicited by $\gamma$. Fix any such line-segment (there are multiple for $n \geq 3$), and denote it by $\lambda$. Our proof shows that the surrogate minimizers trace a line that faithfully mirrors any such $\lambda$ in the following sense: for any pair of distributions $p, q \in \lambda$, if $p < q$ (i.e., when viewed from the origin, $q$ lies to the right of $p$), then $\Gamma(p) < \Gamma(q)$. Thus, the ordering of the surrogate minimizers reflects the ordering of the target cells. This enables the construction of a natural link function $\psi$ under which $(L, \psi)$ is calibrated with respect to the target.
> > >
> > > [1] J. Finocchiaro, R. Frongillo, and B. Waggoner. "Embedding dimension of polyhedral losses" COLT 2020

---

### Official Review · Reviewer_zx7Y · 2025-06-24

**Clarity:** 3
**Significance:** 3
**Originality:** 3
**Rating:** 4
**Confidence:** 3

**Summary:**

The paper tackles the long-standing challenge of certifying when a convex, differentiable surrogate loss is statistically consistent with a desired discrete target loss.  Building on the simpler condition of indirect elicitation (IE), the authors show that for any one-dimensional convex differentiable surrogate, IE is equivalent to the standard calibration criterion that underpins consistency.  They then construct a concrete two-dimensional counter-example demonstrating that this equivalence breaks in higher dimensions, motivating a new, equally easy-to-check notion called strong IE.  Strong IE is proved to imply calibration for all convex differentiable surrogates and to be both necessary and sufficient when the surrogate is also strongly convex.

**Questions:**

* All your guarantees assume the learner reaches an *exact* minimiser of the surrogate risk. In practice we obtain only $\epsilon$-optimal solutions. How does this gap affect the IE / strong-IE ⇒ calibration implications?

* Can the new surrogates certified by strong IE be trained effectively, and do they improve accuracy or calibration error on real data versus standard baselines?

**Ethical Concerns:**

["NO or VERY MINOR ethics concerns only"]

**Final Justification:**

This paper gives a clear, rigorous criterion—strong IE—for when smooth convex surrogates are consistent with a discrete target loss. In 1D it shows IE and calibration coincide; a 2D counterexample shows where this breaks, motivating strong IE, which implies calibration for all convex differentiable surrogates and is necessary and sufficient under strong convexity. The results extend beyond the usual polyhedral setting and are backed by careful proofs; there’s also a constructive path for 1D orderable targets (used in an ordinal-regression example).

The downsides are practical: no experiments, no optimization heuristics, and dense notation that hurts accessibility. The rebuttal helps—consistency already covers ε-optimal solutions, and the authors will streamline the exposition—but there is still no empirical check or rate-style bound. Accordingly, I cannot fully endorse acceptance, but I recognize the paper’s merits and maintain a borderline accept (4).

**Limitations:**

Yes

**Quality:**

3

**Strengths And Weaknesses:**

Strengths

This paper

* Presents rigorous, fully detailed proofs that are internally consistent and build on well-established convex-analysis tools.

* Moves the theory of consistent surrogates beyond polyhedral losses, which dominate prior work, to the far larger class of smooth losses common in deep learning.

* Strong IE itself is a novel concept that generalises earlier “flat-cell” ideas from the polyhedral setting but is tailored to smooth objectives.


## Weaknesses

* Empirical validation is minimal: there is no experiment showing that strong-IE-certified surrogates actually improve learning outcomes compared to uncertified baselines.

* Contributions are purely theoretical; no new algorithmic techniques or optimisation heuristics are proposed, limiting novelty on the applied side.

* Dense notation and frequent switching between set-valued and point-wise views ($\Gamma(p)$, $\Gamma_u$, $\gamma_r$) can overwhelm readers;


I carefully bid for papers that match my areas of expertise, but this paper lies outside my core research interests. Although I work on the theory of machine learning, I found the paper difficult to follow. If the authors were writing for a journal without page constraints, they could devote more space to intuitive explanations, clearer notation, and motivating examples. A thorough rewrite with a simpler narrative flow would greatly improve accessibility for readers who are not already experts in this field.

---

> ### Author Rebuttal · Authors · 2025-07-31
>
> Thank you for your time and helpful feedback. We will improve and clarify the technical notation in the paper. We address specific concerns and questions below.
>
> $$\textbf{“Empirical validation is minimal: there is no experiment showing that strong-IE-certified surrogates actually improve learning outcomes compared to uncertified baselines.”}$$
> Our results show that strong IE guarantees calibration for convex, differentiable surrogates. In this setting, certifying strong IE certifies calibration—and thus consistency. Statistical consistency is a fundamental requirement for surrogate-target alignment going back to at least Bartlett et al. (2006) [1].
>
> $$\textbf{“Contributions are purely theoretical; no new algorithmic techniques or optimisation heuristics are proposed, limiting novelty on the applied side.”}$$
> Our focus is indeed on theory, which is badly needed in the design of surrogate losses. That said, we are motivated by a practical challenge: calibration is often hard to verify. We provide simpler, verifiable conditions—like strong IE—that imply calibration for convex, differentiable surrogates. Prior IE-calibration equivalence for polyhedral surrogates [2] spurred the design of new consistent losses [3–6]; we expect similar impact here. As an initial application of our theory, Theorem 4 provides a constructive method for building consistent 1D differentiable surrogates for orderable targets, which we use in Example 5 to design a novel surrogate for ordinal regression.
>
> $$\textbf{“All your guarantees assume the learner reaches an exact minimiser of the surrogate risk. In practice we obtain only $\epsilon$-optimal solutions. How does this gap affect the IE / strong-IE ⇒ calibration implications?”}$$
> While IE and strong IE allow us to $\textit{reason}$ only about exact minimizers, the $\textit{guarantees}$ we prove (consistency) do handle approximate minimizers by definition. In contrast, proving them via calibration requires analyzing both exact and $\epsilon$-optimal solutions (see Figure 1, pg. 2 of our paper for visual intuition).
>
> $$\textbf{“Can the new surrogates certified by strong IE be trained effectively, and do they improve accuracy or calibration error on real data versus standard baselines?”}$$
> Our results apply to a broad class of convex, differentiable surrogates. These surrogates are well-supported by established training protocols with strong theoretical and empirical performance. For reliable accuracy/ target metric, a principled ML pipeline must ensure the surrogate is statistically aligned with the target loss—precisely what our IE and strong IE results guarantee.
>
> $$\textbf{References}$$
> [1] P. L. Bartlett, M. I. Jordan, and J. D. McAuliffe. "Convexity, classification, and risk bounds". Journal of the American Statistical Association, 2006
>
> [2] J. J. Finocchiaro, R. Frongillo, B. Waggoner. "An Embedding Framework for Consistent Polyhedral Surrogates". NeurIPS 2019
>
> [3] Y. Wang and C. Scott. "Weston-watkins hinge loss and ordered partitions". NeurIPS 2020
>
> [4] J. J. Finocchiaro, R. Frongillo, E. Goodwill, and A. Thilagar. "Consistent polyhedral surrogates
> for top-k classification and variants." ICML 2022
>
> [5] J. J. Finocchiaro, R. Frongillo, and E. B. Nueve. "The structured abstain problem and the lovász hinge". COLT 2022
>
> [6] E. Nueve, D. Kimpara, B. Waggoner, and J. Finocchiaro. "Trading off consistency and dimensionality
> of convex surrogates for multiclass classification". NeurIPS 2024

---

> > ### Comment · Reviewer_zx7Y · 2025-08-01
> >
> > The reviewer thanks the authors for their response and is pleased that they plan to improve the paper’s clarity. Your work is clearly deep, and the effort invested in establishing this theory is appreciated. The reviewer recognizes the paper’s value as foundational theoretical research.
> >
> > Nonetheless, one substantive concern remains:
> >
> > - **Empirical validation:** The most significant weakness is still the lack of experiments. Without empirical evidence demonstrating that the strong-IE–certified surrogates improve accuracy or calibration on real data, the practical benefit of the theory is unproven.

---

> > ### Comment · Reviewer_zx7Y · 2025-08-01
> >
> > The reviewer thanks the authors for their response and is pleased that they plan to improve the paper’s clarity. Your work is clearly deep, and the effort invested in establishing this theory is appreciated. The reviewer recognizes the paper’s value as foundational theoretical research.
> >
> > Nonetheless, one substantive concern remains:
> >
> > - **Empirical validation:** The most significant weakness is still the lack of experiments. Without empirical evidence demonstrating that the strong-IE–certified surrogates improve accuracy or calibration on real data, the practical benefit of the theory is unproven.

---

> > > ### Author Response · Authors · 2025-08-02
> > >
> > > Thank you for your feedback. We appreciate that you value the paper’s theoretical results.
> > > $$\textbf{Regarding empirical validation}$$
> > >
> > > Our paper does not provide experiments for the same reason the vast majority of theoretical work on consistency does not (see [1]-[15]): there is broad consensus that consistency is a fundamental and practically-valuable criterion for aligning surrogates with target losses ([16]-[20]). Our results are all about easing the task of proving consistency, and as such, the practical benefits of strong IE-certified surrogates derive directly from the benefits of consistency-certified surrogates. In other words, strong-IE improves accuracy/calibration on real data, because consistency improves accuracy/calibration on real data.
> > >
> > > That said, we would be happy to add some experiments to empirically demonstrate the benefits. The first one that comes to mind is an ordinal regression experiment that compares our proposed surrogate with another reasonable 1d surrogate which does not satisfy IE; we would be open to other suggestions however.
> > >
> > > $$\textbf{References}$$
> > > [1] T. Zhang. "Statistical behavior and consistency of classification methods based on convex risk minimization", Annals of Statistics, 2004.
> > >
> > > [2] P. Bartlett, M. Jordan, and J. McAuliffe. "Convexity, classification, and risk bounds", Journal of the American Statistical Association, 2006.
> > >
> > > [3] A. Tewari and P. Bartlett. "On the consistency of multiclass classification methods", JMLR, 2007.
> > >
> > > [4] I. Steinwart. "How to compare different loss functions and their risks", Constructive Approximation, 2007.
> > >
> > > [5] D. Cossock, T. Zhang. “Statistical Analysis of Bayes Optimal Subset Ranking”, IEEE Transactions on Information Theory, 2008
> > >
> > > [6] MD Reid and RC Williamson. “Composite Binary Losses”, JMLR, 2010
> > >
> > > [7] W. Gao, Z.H. Zhao. “On the Consistency of Multi-Label Learning”, COLT, 2011
> > >
> > > [8] C. Scott. “Calibrated asymmetric surrogate losses”, EJS, 2012
> > >
> > > [9] A. Agarwal and S. Agarwal. On consistent surrogate risk minimization and property elicitation. COLT, 2015
> > >
> > > [10] H. G. Ramaswamy and S. Agarwal. "Convex calibration dimension for multiclass loss matrices." JMLR 2016
> > >
> > >
> > > [11] A Osokin, F Bach, S Lacoste-Julien. “On structured prediction theory with calibrated convex surrogate losses.” NeurIPS 2017
> > >
> > >
> > > [12] J. J. Finocchiaro, R. Frongillo, B. Waggoner. "An Embedding Framework for Consistent Polyhedral Surrogates". NeurIPS 2019
> > >
> > >
> > > [13] J. J. Finocchiaro, R. Frongillo, and E. B. Nueve. "The structured abstain problem and the lovász hinge". COLT 2022
> > >
> > >
> > > [14] H. Bao, “Proper Losses, Moduli of Convexity and Surrogate Regret Bounds”, COLT 2023
> > >
> > >
> > > [15] N. Frank, J Niles-Weed. “The adversarial consistency of surrogate risks for binary classification”, NeurIPS 2023
> > >
> > > [16] P. Ravikumar, A. Tewari, E. Yang. “On NDCG Consistency of Listwise Ranking Methods”, AISTATS, 2011
> > >
> > > [17] Kotlowski, W., Dembczynski, K. J., & Huellermeier, E. “Bipartite ranking through minimization of univariate loss”, ICML, 2011
> > >
> > > [18] HG, A Tewari, S Agarwal. “Convex calibrated surrogates for hierarchical classification.”, ICML 2015
> > >
> > > [19] HG Ramaswamy, A Tewari, S Agarwal. “Consistent algorithms for multiclass classification with an abstain option”, EJS, 2018
> > >
> > > [20] M Blondel, AFT Martins, V Niculae. “Learning with Fenchel-Young Losses”, JMLR 2020

---

### Official Review · Reviewer_fLKV · 2025-07-03

**Clarity:** 3
**Significance:** 2
**Originality:** 3
**Rating:** 4
**Confidence:** 3

**Summary:**

This work establishes general calibration conditions on convex differentiable surrogate losses for domains with discrete targets. Previous work had shown that calibration and Indirect Elicitation (IE) are equivalent for polyhedral surrogates. This work extends the results and introduces the notion of strong IE, which implies calibration for differentiable functions. Strong IE is also shown to be a necessary and sufficient condition for calibration for the case of strongly convex differentiable functions. The paper also shows that for 1-d convex losses, IE and calibration are equivalent. But this equivalence breaks in higher dimensions, requiring the use of strong IE for calibration. This is made clear with a number of counter examples in the paper.

The theoretical results in the paper makes it possible to more easily analyse and prove calibration for previously studied surrogate losses. Examples 3 and 4 in the paper are proven using the results of Theorems 2 and 3, resulting in significantly shortened proof of calibration. The paper also introduces a Huber-like surrogate for ordinal regression (Example 5), which is shown to be calibrated as a 1-dimensional convex and differentiable function. There are no empirical studies present in the paper.

**Questions:**

- Is it possible to design higher dimensional surrogates using the insights of Theorem 3? Are there any examples that could be added to the paper?
- Line 106: IIUC this looks incorrect. "For any r" instead of "For any u".

**Ethical Concerns:**

["NO or VERY MINOR ethics concerns only"]

**Final Justification:**

The rebuttal answered some of the concerns and questions raised in my review. I'm updating the overall score based on the updated version with expanded discussions and added citations.

**Limitations:**

The paper has sufficiently discussed its limitations and potentials for future work in section 5.

**Paper Formatting Concerns:**

I did not find any formatting issues with the paper.

**Quality:**

3

**Strengths And Weaknesses:**

Quality, Clarity: The paper is overall well written, but the specific terminology and overloaded notation can be cumbersome to deal with. The work is motivated mostly in theory, without much attention to empirical implications. The paper can greatly benefit from examples of misbehaved training due to lack of calibration for the chosen surrogate loss.

Significance, Originality: The study of calibration for non-polyhedral losses is new to this work (to the best of my knowledge). Theorems 2, 3 and 4 are novel and can be used (as demonstrated in 4.1) to easily prove calibration for a large class of surrogate losses. However, IIUC, there is only one novel Huber-like surrogate loss presented in this work (Example 5), the design of which is rooted in Theorem 4. The paper can greatly benefit from other constructions that might be possible with the insights and theorems presented in the work.

---

> ### Author Rebuttal · Authors · 2025-07-31
>
> Thank you for your time and helpful feedback. We will improve and clarify the technical notation in the paper. We address specific concerns and questions below.
>
> $$\textbf{“The paper can greatly benefit from examples of misbehaved training due to lack of calibration for the chosen surrogate loss."}$$
> Statistical consistency (and hence calibration) is a fundamental requirement for ensuring compatibility between a surrogate and target loss. This long-standing literature goes back to at least Bartlett et al. (2006) [1] and has since been an active area of research.
>
> $$\textbf{“However, IIUC, there is only one novel Huber-like surrogate loss presented in this work (Example 5), the design of which is rooted in Theorem 4. The paper can greatly benefit from other constructions that might be possible with the insights and theorems presented in the work"}$$
> We also provide a general-purpose construction for designing convex, differentiable surrogates for orderable targets. Our results can enable new convex, differentiable surrogate designs—just as the IE-calibration equivalence in [3] led to progress on consistent polyhedral surrogates [4–7].
>
> $$\textbf{“Is it possible to design higher dimensional surrogates using the insights of Theorem 3? Are there any examples that could be added to the paper?"}$$
> High-dimensional surrogates—e.g., $(n-1)$-dimensional consistent surrogates for $n$ outcomes—are already known (see Lemma 11, p.19 of [2]). Low-dimensionality is desirable as it can yield significant computational savings during optimization.
>
> $$\textbf{References}$$
> [1] P. L. Bartlett, M. I. Jordan, and J. D. McAuliffe. "Convexity, classification, and risk bounds". Journal of the American Statistical Association, 2006
>
> [2] H. G. Ramaswamy and S. Agarwal. "Convex calibration dimension for multiclass loss matrices." JMLR 2016
>
> [3] J. J. Finocchiaro, R. Frongillo, B. Waggoner. "An Embedding Framework for Consistent Polyhedral Surrogates". NeurIPS 2019
>
> [4] Y. Wang and C. Scott. "Weston-watkins hinge loss and ordered partitions". NeurIPS 2020
>
> [5] J. J. Finocchiaro, R. Frongillo, E. Goodwill, and A. Thilagar. "Consistent polyhedral surrogates
> for top-k classification and variants." ICML 2022
>
> [6] J. J. Finocchiaro, R. Frongillo, and E. B. Nueve. "The structured abstain problem and the lovász hinge". COLT 2022
>
> [7] E. Nueve, D. Kimpara, B. Waggoner, and J. Finocchiaro. "Trading off consistency and dimensionality
> of convex surrogates for multiclass classification". NeurIPS 2024

---

> > ### Author Response · Authors · 2025-08-06
> >
> > We hope you have had a chance to go through our response, and that it has helped clarify things. Please let us know if you have any further questions.

---

> > > ### Comment · Reviewer_fLKV · 2025-08-08
> > >
> > > The authors have answered most of my questions in the review. I would encourage the authors to expand on these points in the main body of the paper by citing the studies that were discussed in the rebuttal. I am updating my score accordingly.

---

> > > > ### Author Response · Authors · 2025-08-08
> > > >
> > > > Thank you very much for your response. We will be sure to incorporate the key points and relevant citations from our discussion into the final version of our paper.

---

### Official Review · Reviewer_u7xK · 2025-07-09

**Clarity:** 3
**Significance:** 3
**Originality:** 3
**Rating:** 4
**Confidence:** 3

**Summary:**

The paper introduces a new approach for proving consistency of differentiable surrogate losses. Their approach studies how the indirect elicitation can be used to prove calibration and thus statistical consistency. Within the paper they highlight examples of when indirect elicitation and calibration simultaneously hold and when they do not simultaneously hold. They then prove that calibration holds for 1-dimensional continuous and differentiable surrogates that indirectly elicit the target loss. To handle higher dimensional surrogates, they introduce the strong indirect elicitation condition to prove the same calibration result. Finally, they show for strongly convex surrogates that strong indirect elicitation is equivalent to calibration. The remainder of the paper highlights how their approach simplifies existing proofs proving calibration for the statistical consistency of surrogates.

**Questions:**

1. Can more detailed be provided why Example 6 fixes the issues of Example 1? In particular, the last argument in Example 6 seems to hold for Example 1 since $L$ is strictly convex for Example 1 as well.
2. Are there some simple examples or intuition for when $\gamma$ is order able?

**Ethical Concerns:**

["NO or VERY MINOR ethics concerns only"]

**Final Justification:**

The paper seems to provide novel theoretical contribution towards understanding the consistency of surrogates losses. My issues with the paper mainly arise from the exposition as it is somewhat hard to follow and feels written primarily for researchers who have extensive background knowledge in the area. The authors' rebuttals to my and other reviewers' questions were helpful, however, it is hard for me to increase the score since it is less in my area of expertise and thus am more willing to defer to the other reviewers.

**Limitations:**

Yes

**Quality:**

3

**Strengths And Weaknesses:**

#Strengths
1. The paper introduces a useful and simpler technique for proving the statistical consistency of surrogate losses.
2. The paper provides useful examples highlighting how using indirect elicitation simplifies complicated calibration proofs.

#Weaknesses
1. From a broader perspective, it's hard to understand if there's any drawbacks of only focusing on surrogate losses which it is easy to prove indirect elicitation. Or in the opposite perspective, there's limited intuition on what settings naturally lead to creating a surrogate with indirect elicitation. As a result it's hard to measure how general this approach is in developing new surrogate losses.
2. The exposition and notation could be improved in some sections of the paper. For example for notation, $\gamma_{\gamma(p)}$ in example 6 is some what confusing based on its definition in Definition 1. Similarly, the authors use $p = (1/2, 0, 1/2)$ even though $p$ is used more often as a generic input for $\gamma$ and $\Gamma$.  Exposition-wise, it was hard to follow the intuition for why Example 1 shows a surrogate loss can satisfy IE but not calibration especially when compared to Example 6. In particular, it was hard to see why the last argument in Example 6 doesn't hold for Example 1.

---

> ### Author Rebuttal · Authors · 2025-07-31
>
> Thank you for your time and helpful feedback. We will improve and clarify the technical notation in the paper. We address specific concerns and questions below.
>
> $$\textbf{‘‘...  As a result it's hard to measure how general this approach is in developing new surrogate losses."}$$
> While our paper's primary focus is not on designing novel surrogates, we have taken important first steps towards discovering novel consistent surrogates satisfying convexity and differentiability. Specifically:
>
> 1) The conditions of IE and strong IE are simpler to verify than calibration, offering practical tools to more efficiently evaluate and prototype candidate surrogates. See Theorem 4 and Example 5.
>
> 2) Our results mirror prior work on polyhedral surrogates ([1]), where an analogous IE-calibration equivalence led to rapid progress in designing novel surrogates in the polyhedral setting ([2]–[5]). Notably, this equivalence enabled the resolution of open problems such as the construction of consistent polyhedral surrogates for top-k classification [3], despite earlier negative conjectures [6]. The fact that such conjectures were overturned using simpler IE-based reasoning underscores how opaque and non-constructive calibration alone can be.
>
> $$\textbf{‘‘... the last argument in Example 6 seems to hold for Example 1 since $L$
>  is strictly convex for Example 1 as well."}$$
> The two examples use different links. We will make this more obvious. The inequality from Example 6 that you reference makes implicit use of the link definition $\textit{before}$ invoking strict-convexity.
>
> To elaborate: the set of distributions where abstaining is target-optimal is $\gamma_{\bot} = \\{p \in \Delta_2 : \max{p_1, p_2} \leq 0.75\\}$. Consider $q = [0.5, 0.5]$, so $q \in \gamma_{\bot}$. Calibration holds for example 6 but fails for example 1 at $q$.
>
> $\textbf{Example 6}$: $\psi_{smooth}(u) = \bot$ iff $u \in [-0.5,0.5]$. So, $\\{u: \psi_{smooth}(u) \notin \bot\\} = \\{u: u \notin [-0.5,0.5]\\}$. Recall also that $\Gamma_{smooth}(q) = \\{0\\}$. Thus, $inf_{u: \psi_{smooth}(u) \notin \bot} \langle q,  L_{smooth}(u) \rangle = inf_{u: u \notin [-0.5, 0.5] } \langle q, L_{smooth}(u) \rangle >  inf_{u: u \in [-\epsilon, \epsilon] } \langle q, L_{smooth}(u) \rangle = \langle q, L_{smooth}(0) \rangle$, where the inequality holds for a small enough $\epsilon$, say $\epsilon = 0.1$ by strict-convexity (the function value grows away from the minimum).
>
> $\textbf{Example 1}$: Here, we $\textit{only}$ link $0$ to $\bot$, i.e., $\psi_{cusp}(u) = \bot \iff u = 0$. Hence $inf_{u: \psi_{cusp}(u) \notin \bot} \langle q, L_{cusp}(u) \rangle = inf_{u: u \notin \\{0\\}} \langle q, L_{cusp}(u) \rangle$ and there is no $\epsilon > 0$ satisfying, $inf_{u: u \notin \\{0\\}} \langle q, L_{cusp}(u) \rangle >  inf_{u: u \in [-\epsilon, \epsilon] } \langle q, L_{cusp}(u) \rangle$. Here too, $\Gamma_{cusp}(q) = \\{0\\}$ and hence any non-zero sequence converging to $0$ minimizes $\langle q, L_{cusp}(\cdot) \rangle$ despite that sequence never linking to $\bot$, which violates calibration.
>
> $$\textbf{“Are there some simple examples or intuition for when $\gamma$ is orderable?"}$$
> Yes, see our Example 5 (ordinal regression loss), or Example 9 in [7] ($\epsilon$-insensitive loss for robust classification). Other orderable properties include the median and the ratio-of-expectations [9]. Intuitively, a target loss is orderable if it is possible to draw a line segment through the simplex such that it intersects $\textit{every}$ target-cell. See [8, Figures 1 and 2] for a visual comparison between orderable and non-orderable properties.
>
> $\textbf{References}$
>
> [1] J. J. Finocchiaro, R. Frongillo, B. Waggoner. "An Embedding Framework for Consistent Polyhedral Surrogates". NeurIPS 2019
>
> [2] Y. Wang and C. Scott. "Weston-watkins hinge loss and ordered partitions". NeurIPS 2020
>
> [3] J. J. Finocchiaro, R. Frongillo, E. Goodwill, and A. Thilagar. "Consistent polyhedral surrogates
> for top-k classification and variants." ICML 2022
>
> [4] J. J. Finocchiaro, R. Frongillo, and E. B. Nueve. "The structured abstain problem and the lovász hinge". COLT 2022
>
> [5] E. Nueve, D. Kimpara, B. Waggoner, and J. Finocchiaro. "Trading off consistency and dimensionality
> of convex surrogates for multiclass classification". NeurIPS 2024
>
> [6] F. Yang and S. Koyejo. "On the consistency of top-k
> surrogate losses." ICML 2020
>
> [7] H. G. Ramaswamy and S. Agarwal. "Convex calibration dimension for multiclass loss matrices." JMLR 2016
>
> [8] J. Finocchiaro, R. Frongillo, and B. Waggoner. "Embedding dimension of polyhedral losses" COLT 2020
>
> [9] N. S. Lambert, D. M. Pennock, and Y. Shoham. “Eliciting properties of probability distributions.” In Proceedings of the 9th ACM Conference on Electronic Commerce, 2008

---

> > ### Author Response · Authors · 2025-08-06
> >
> > We hope you have had a chance to go through our response, and that it has helped clarify things. Please let us know if you have any further questions.

---

### Decision · Program_Chairs · 2025-09-17

**Decision:**

Accept (poster)

**Comment:**

I am recommending this paper for publication. The reviewers all agree that the technical contribution is strong. Specifically that IE is equivalent to calibration is a strong result, I suggest authors work on improving the presentation as mentioned by the reviewers.